# Geometry-Misalignment in Distributional Learning

**Tao Wang** [* 1]   **Xiaoting Zhong** [* 2]

## Abstract

Distributional learning problems optimize discrepancies between probability measures, including optimal transport or Sinkhorn divergence, yet are typically optimized using Euclidean first-order methods in parameter space. We show this mismatch is structural rather than algorithmic. We introduce geometry-misalignment, a local condition number that measures distortion between Euclidean geometry and the intrinsic geometry induced by a distributional objective. For a broad class of problems, we establish lower bounds demonstrating that Euclidean first-order methods incur an unavoidable convergence slowdown proportional to misalignment, even under intrinsic strong convexity and smoothness. We further prove geometry-aware preconditioned methods attain matching upper bounds independent of misalignment, yielding a sharp separation between Euclidean optimization and geometry-aware optimization. Beyond convergence rates, we show geometry-misalignment induces an optimization-dependent excess risk term under finite budgets, directly linking optimization geometry with statistical efficiency. We develop a geometry-calibrated optimization framework that estimates misalignment and selectively activates geometry-aware updates when necessary. Experiments on distribution matching for domain adaptation validate the theory, with improvements concentrated in high-misalignment regimes and negligible overhead.

## 1. Introduction

Distributional learning problems play a central role in modern machine learning, including optimal transport based

[1]Department of Economics and Department of Mathematics and Statistics (by Courtesy), University of Victoria, Canada [2]Department of Psychological and Quantitative Foundations, University of Iowa, USA. Correspondence to: Tao Wang <taow@uvic.ca>.

*Proceedings of the $43^{rd}$ International Conference on Machine Learning*, Seoul, South Korea. PMLR 306, 2026. Copyright 2026 by the author(s).

domain adaptation, representation learning via distribution matching, robustness under distribution shift, and fairness-aware learning (Courty et al., 2017; Genevay et al., 2018; Sinha et al., 2018; Gordaliza et al., 2019; Silvia et al., 2020). In these settings, the learning objective is defined by a discrepancy between probability measures, such as optimal transport or Sinkhorn divergence, rather than a pointwise loss (Cuturi, 2013; Sejdinovic et al., 2013; Courty et al., 2017). Despite this fundamental difference, optimization is almost universally performed using Euclidean first-order methods in parameter space, most commonly stochastic gradient descent and its adaptive variants (Goodfellow et al., 2016; Bottou et al., 2018; Genevay et al., 2018). This creates a geometric mismatch between the distributional structure of the objective and the optimization dynamics used to minimize it during iterative updates.

In practice, this mismatch is accompanied by fragile optimization behavior. Training with distributional objectives can be highly sensitive to step size, converge slowly, or exhibit instabilities such as oscillatory updates or representation collapse, even in smooth and well-regularized settings. These pathologies are especially pronounced in distribution matching problems, where small parameter perturbations can induce highly non-uniform changes in the induced distributions (Feydy et al., 2019; Li et al., 2021). Standard explanations attribute such behavior to nonconvexity or stochastic noise, but these factors do not explain why similar difficulties persist in convex or low-noise regimes, nor why Euclidean optimization performs reliably for likelihood-based objectives while degrading for distributional ones (Amari and Nagaoka, 2000; Martens and Grosse, 2015).

In this work, we show that the root cause is *structural rather than algorithmic*. Distributional discrepancies endow the model manifold with an intrinsic geometry that governs how parameter perturbations affect probability measures and induces strong directional anisotropy (Chizat et al., 2018; Peyré and Cuturi, 2019). Euclidean optimization implicitly assumes that this intrinsic geometry is well approximated by the standard Euclidean metric on parameters. When this assumption fails, Euclidean gradients misrepresent both the scale and the direction of meaningful descent, leading to fundamental inefficiencies that cannot be resolved by step-size tuning or adaptive heuristics (Nesterov, 2018). We emphasize that the contribution is not

the use of preconditioning itself, which is classical in Newton, quasi-Newton, natural-gradient, mirror-descent, and approximate-Newton methods. The contribution is the identification of a distributional-learning-specific misalignment condition number, together with lower and upper bounds showing when Euclidean first-order optimization is intrinsically inadequate and when geometry-aware updates remove the misalignment dependence. Thus, our results provide a structural theory of when Euclidean optimization is mismatched to distributional objectives, rather than a claim that curvature-based preconditioning is new.

We formalize this phenomenon through the notion of *geometry-misalignment*, a condition number that quantifies distortion between the Euclidean metric in parameter space and the intrinsic geometry induced by a distributional objective. Geometry-misalignment captures anisotropy in parameter sensitivity that is invisible to Euclidean smoothness constants and arises naturally in distributional representation learning, where feature extractors can amplify or suppress directions that are critical for aligning distributions.

Our theoretical contribution establishes lower bounds showing that for a broad class of distributional learning problems, any Euclidean first-order method incurs an unavoidable convergence slowdown proportional to geometry-misalignment, even when the objective is strongly convex and smooth with respect to its intrinsic geometry. We show that this inefficiency is intrinsic and that the lower bound is tight by proving that geometry-aware preconditioned methods achieve convergence rates independent of the misalignment, yielding a sharp separation between Euclidean and geometry-aware optimization. We further show that geometry-misalignment has statistical consequences; under finite optimization budgets, Euclidean methods incur an optimization-dependent excess risk term that scales with misalignment, while geometry-aware methods remove this penalty, directly linking optimization geometry to statistical efficiency.

We also propose a geometry-calibrated optimization framework that estimates geometry-misalignment and selectively applies geometry-aware updates when needed, interpolating between Euclidean optimization and geometry-aware preconditioning while efficiently avoiding overhead in well-aligned regimes. We validate our theory empirically on distribution matching for domain adaptation using Sinkhorn divergence, where geometry-aware optimization consistently improves convergence and target performance in high-misalignment regimes and reduces to standard Euclidean optimization when Euclidean geometries are well aligned.

## 2. Related Work

We review prior work on distributional learning, optimization geometry, and geometry-aware optimization, and clar-

ify how our approach differs by providing a structural diagnosis with tight theoretical guarantees.

### 2.1. Distributional Learning and Optimal Transport

Distributional learning replaces pointwise loss minimization with objectives defined on probability measures, including optimal transport, Sinkhorn divergence, maximum mean discrepancy, and energy distances. These objectives are used in domain adaptation, representation learning, generative modeling, fairness, and robustness under distribution shift (Sriperumbudur et al., 2012; Frogner et al., 2015; Arjovsky et al., 2017; Vuong et al., 2023). Entropic regularization has played a central role in making optimal transport computationally tractable at scale (Cuturi, 2013; Lin et al., 2022).

In domain adaptation and representation learning, optimal transport-based objectives provide a principled mechanism for aligning source and target distributions in feature space (Clason et al., 2021). Related ideas have been explored in deep distribution matching and generative modeling, where representations are learned by minimizing distributional discrepancies (Ollivier, 2018; Hallin et al., 2021). While these methods often achieve strong empirical performance, their training dynamics are sensitive to optimization choices, particularly in regimes where small parameter perturbations induce anisotropic changes in the induced distributions (Ollivier, 2018; Feydy et al., 2019; Li et al., 2021; Boffi et al., 2022). Rather than proposing new discrepancies or objectives, our work studies how existing discrepancies induce intrinsic geometries on parameter space and how this geometry governs optimization efficiency.

Distributionally robust optimization (DRO) is also closely related, since it optimizes losses under uncertainty sets around an empirical distribution (Cisneros-Velarde et al., 2020; Słowik and Bottou, 2022). DRO methods often introduce an inner distributional subproblem to protect against sampling error or distribution shift. Our focus is different: we study the optimization geometry induced by distributional discrepancies themselves and ask when Euclidean parameter-space optimization is misaligned with that geometry. Thus, while DRO typically changes the learning objective to improve robustness, our analysis studies how a given distributional objective should be optimized.

### 2.2. Euclidean First-Order Optimization

Stochastic gradient descent and its adaptive variants are the dominant optimization tools in large-scale machine learning (Bottou, 2010; Goodfellow et al., 2016). These methods operate in Euclidean parameter space and rely on smoothness and curvature notions defined with respect to the Euclidean metric, which can be effective when objectives are well conditioned (Nocedal and Wright, 2006). However, Euclidean smoothness does not adequately capture the behavior of

distributional objectives. Small parameter perturbations can induce highly anisotropic changes in probability space, causing Euclidean gradients to misrepresent local sensitivity and resulting in slow convergence or instability.

Existing analyses typically attribute such behavior to non-convexity, stochastic noise, or Euclidean conditioning (Agarwal et al., 2012; Bonnabel, 2013; Drori and Teboulle, 2014; Woodworth and Srebro, 2016), but these explanations do not account for why difficulties persist in smooth or convex settings, nor why Euclidean optimization performs reliably for likelihood-based objectives but degrades for distributional ones. Our analysis shows this phenomenon reflects a deeper mismatch between Euclidean geometry and the intrinsic geometry induced by distributional objectives.

### 2.3. Geometry-Aware Optimization

Geometry-aware optimization methods account for non-Euclidean structure by modifying gradient updates using problem-dependent metrics. Natural gradient methods replace the Euclidean metric with the Fisher information metric, yielding reparameterization-invariant updates (Amari, 1998), while mirror descent performs optimization in a dual space defined by a convex potential (Beck and Teboulle, 2003). These approaches have been applied in probabilistic modeling, information geometry, and Riemannian optimization (Absil et al., 2008; Bonnabel, 2013; Zhang et al., 2024).

More recently, geometry-aware ideas have been explored in optimal transport and related distributional objectives, where geometric and dynamical structures arise naturally on spaces of probability measures (Jordan et al., 1998; Ambrosio et al., 2008; Santambrogio, 2015; Li et al., 2018; Pereira and Amini, 2025). However, most existing work focuses on algorithmic design or empirical performance and does not address when Euclidean optimization is fundamentally inadequate. In particular, prior work does not establish lower bounds demonstrating that Euclidean first-order methods must fail in highly anisotropic regimes, nor does it provide matching upper bounds characterizing when geometry-aware methods are necessary (Baey et al., 2023; Dangel et al., 2024). In contrast, our work provides a sharp theoretical separation by showing that geometry-aware optimization is provably necessary when geometry-misalignment is large.

### 2.4. Newton-Type Second-Order Methods

Newton, quasi-Newton, and approximate-Newton methods use local curvature information to precondition gradients and improve conditioning (Nocedal and Wright, 2006; Bottou et al., 2018; Ye et al., 2017; 2021). In the special case where the intrinsic metric coincides with the Euclidean Hessian of the objective, the geometry-aware update in this paper reduces to a damped Newton-type update. We do not claim that Hessian-based or Hessian-like preconditioning

is new. Our distinction is that, for distributional objectives, the relevant curvature is interpreted as the pullback metric induced by the discrepancy on the model manifold, and the condition number $\kappa_G$ quantifies the mismatch between this intrinsic geometry and the Euclidean geometry used by standard first-order methods.

This distinction leads to results that are different from classical Newton-type analyses. Classical analyses typically study convergence once a curvature approximation is chosen. Here, the lower bound shows that Euclidean first-order methods incur a $\kappa_G$-dependent slowdown even under intrinsic strong convexity and smoothness, while the upper bounds show that geometry-aware and geometry-calibrated updates remove this dependence. Geometry-calibrated optimization is therefore not proposed as a generic replacement for Newton or quasi-Newton methods, but as a diagnostic and adaptive procedure for distributional learning; it invokes intrinsic solves only when the estimated misalignment indicates that Euclidean updates are unreliable.

### 2.5. Optimization Geometry and Statistical Efficiency

A growing literature studies the interaction between optimization and statistical generalization under finite computational budgets, including early stopping, implicit regularization, and optimization-induced bias (Xu and Mannor, 2012; Raskutti et al., 2014; Dieuleveut and Bach, 2016; Hardt et al., 2016; Bassily et al., 2020). These analyses typically assume Euclidean optimization and do not consider how optimization geometry affects statistical efficiency. Our results show that geometry-misalignment induces an additional optimization-dependent excess risk term under finite budgets, establishing a direct link between intrinsic geometry, optimization dynamics, and statistical performance.

## 3. Geometry-Misalignment

We study distributional learning objectives defined by discrepancies between probability measures, formalize the associated optimization framework and intrinsic geometry, and introduce geometry-misalignment to quantify distortion between Euclidean and intrinsic geometries.

### 3.1. Distributional Objective and Optimization

Let $\mathcal{X}$ be a measurable space and let $P$ be an unknown probability distribution on $\mathcal{X}$. We observe independent and identically distributed (i.i.d.) samples $X_1, \ldots, X_n \sim P$ and denote by $\hat{P}_n$ the empirical distribution. Let $\{Q_\theta : \theta \in \Theta \subset \mathbb{R}^d\}$ be a parametric family of probability measures on a measurable space $\mathcal{Y}$, where $\Theta$ is an open parameter domain. We consider distributional learning objectives of the form

$$F(\theta) = \mathcal{D}(P, Q_\theta), \quad \hat{F}_n(\theta) = \mathcal{D}(\hat{P}_n, Q_\theta),$$

where $\mathcal{D}$ is a discrepancy between probability measures, s-

uch as entropic optimal transport or Sinkhorn divergence. The learning task is to approximate a minimizer of the population objective $F$, or of its empirical counterpart $\hat{F}_n$, using first-order optimization methods.

We analyze first-order optimization schemes obtained by minimizing a local linearization of the objective regularized by a quadratic metric (Nemirovsky and Yudin, 1983; Guo et al., 2025). Specifically, given a (possibly stochastic) gradient $g_t$ of $F$ or $\hat{F}_n$ evaluated at $\theta_t$, we consider updates of the form

$$\theta_{t+1} = \arg \min_{\theta} \left\{ \langle g_t, \theta - \theta_t \rangle + \frac{1}{2\eta_t} \|\theta - \theta_t\|_{H_t^{-1}}^2 \right\},$$

whose optimality condition yields

$$\theta_{t+1} = \theta_t - \eta_t H_t g_t,$$

where $\eta_t > 0$ is a step size and $H_t \in \mathbb{R}^{d \times d}$ is a symmetric positive definite operator. Euclidean gradient descent corresponds to $H_t = I$, while geometry-aware methods arise from choosing $H_t$ to reflect problem-specific geometry, including natural gradient descent with $H_t = G(\theta_t)^{-1}$ (Amari, 1998), where $G(\theta)$ denotes the intrinsic metric induced by the second-order variation of the distributional discrepancy $\mathcal{D}(P, Q_\theta)$, pulled back to parameter space. Throughout the theoretical analysis, population optimization is used to isolate the geometric mechanism from statistical estimation error, stochastic-gradient noise, and nonconvex representation effects. Statistical consequences for empirical optimization are then studied separately in Section 5.

We now state the standing regularity conditions under which the local geometry of the objective is well defined and governs the behavior of first-order methods.

**Assumption 3.1.** The parameterization $\theta \mapsto Q_\theta$ is twice differentiable, and the map $\theta \mapsto \mathcal{D}(P, Q_\theta)$ is differentiable.

**Assumption 3.2.** There exists a unique minimizer $\theta^\star \in \Theta$ of $F$, and $F$ admits a second-order expansion near $\theta^\star$.

**Assumption 3.3.** When stochastic gradients are used, their variance is uniformly bounded in a neighborhood of $\theta^\star$.

Under these assumptions, the population objective $F(\theta) = \mathcal{D}(P, Q_\theta)$ admits a local quadratic expansion around $\theta^\star$, with curvature determined by how perturbations $\delta \in \mathbb{R}^d$ propagate through $\theta \mapsto Q_\theta$. As a result, optimization behavior is governed by the intrinsic geometry induced by $\mathcal{D}$, rather than by Euclidean curvature alone.

### 3.2. Intrinsic Geometry and Geometry-Misalignment

For $F(\theta) = \mathcal{D}(P, Q_\theta)$, let $\nabla F(\theta)$ denotes the first derivative. Unlike pointwise losses, the discrepancy $\mathcal{D}$ induces a notion of local sensitivity on the space of probability measures. Under Assumptions 3.1 and 3.2, the second-order

expansion of $F$ around $\theta$ defines a local quadratic form

$$\mathcal{D}(P, Q_{\theta+\delta}) = \mathcal{D}(P, Q_\theta) + \langle \nabla F(\theta), \delta \rangle + \frac{1}{2} \langle \delta, G(\theta)\delta \rangle + o(\|\delta\|^2),$$

which defines a positive definite bilinear form $\langle \delta_1, \delta_2 \rangle_{G(\theta)} = \delta_1^\top G(\theta) \delta_2$ on parameter space. In local Euclidean coordinates, $G(\theta)$ is the Hessian-like matrix governing second-order variation of the distributional objective. When the discrepancy induces a metric on the model manifold $\{Q_\theta\}$, $G(\theta)$ can equivalently be viewed as the pullback of that metric through the parameterization $\theta \mapsto Q_\theta$. Thus, $G(\theta)$ is not a new form of curvature; rather, the point is that for distributional objectives this curvature has an intrinsic geometric interpretation, and its conditioning relative to the Euclidean metric governs optimization difficulty.

The intrinsic geometry is characterized by the quadratic form

$$\|\delta\|_{G(\theta)}^2 = \langle \delta, G(\theta)\delta \rangle, \quad \delta \in \mathbb{R}^d.$$

Directions $\delta$ that induce large changes in probability space correspond to large intrinsic curvature, while directions inducing negligible changes correspond to small curvature (Bonnabel, 2013). The intrinsic metric admits a variational characterization as the unique choice for which the steepest descent direction satisfies

$$v^\star(\theta) = \arg \min_{v \in \mathbb{R}^d} \left\{ \langle \nabla F(\theta), v \rangle + \frac{1}{2} \|v\|_{G(\theta)}^2 \right\} = -G(\theta)^{-1} \nabla F(\theta),$$

corresponding to the Riemannian gradient associated with the metric $G(\theta)$ (Absil et al., 2008). Consequently, optimization in the intrinsic geometry coincides with natural gradient descent when $G(\theta)$ matches the information metric induced by the objective (Amari, 1998; Boffi et al., 2022).

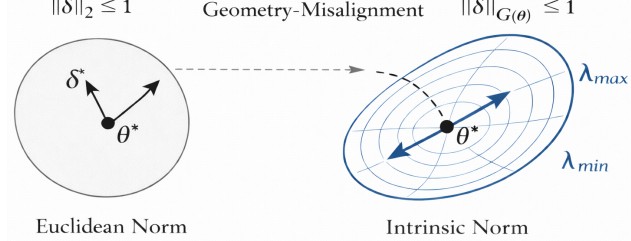

*Figure 1.* Geometry-misalignment between Euclidean and intrinsic norms. Left: the Euclidean unit ball $\{\delta : \|\delta\|_2 \le 1\}$ treats all directions uniformly. Right: the intrinsic unit ball $\{\delta : \|\delta\|_{G(\theta)} \le 1\}$ is anisotropic, with principal axes given by eigenvalues of $G(\theta)$. The distortion is quantified by $\kappa_G(\theta) = \lambda_{\max}(G(\theta))/\lambda_{\min}(G(\theta))$.

Euclidean optimization assumes that the intrinsic metric $G(\theta)$ is well approximated by the Euclidean metric. Geometry-misalignment quantifies the failure of this assumption.

**Definition 3.4.** The geometry-misalignment condition number at parameter value $\theta$ is defined as

$$\kappa_G(\theta) = \frac{\lambda_{\max}(G(\theta))}{\lambda_{\min}(G(\theta))},$$

where $\lambda_{\max}$ and $\lambda_{\min}$ denote the largest and smallest eigenvalues of $G(\theta)$, respectively.

Equivalently, geometry-misalignment can be expressed as the distortion between Euclidean and intrinsic norms

$$\kappa_G(\theta) = \sup_{\delta \neq 0} \frac{\|\delta\|_{G(\theta)}^2}{\|\delta\|_2^2} \cdot \sup_{\delta \neq 0} \frac{\|\delta\|_2^2}{\|\delta\|_{G(\theta)}^2}.$$

The equivalence follows from the Rayleigh quotient characterization of the extremal eigenvalues of $G(\theta)$. When $\kappa_G(\theta) \approx 1$, Euclidean and intrinsic geometries are locally aligned. When $\kappa_G(\theta)$ is large, Euclidean distances misrepresent local changes in the objective. In such regimes, a single Euclidean step size cannot simultaneously control progress along all intrinsic directions, even when $F$ is smooth and strongly convex with respect to the intrinsic geometry.

Although the intrinsic metric $G(\theta)$ may be expensive to compute explicitly, geometry-misalignment can be estimated efficiently at negligible additional cost. Given access to matrix-vector products $v \mapsto G(\theta)v$ and approximate solves involving $G(\theta)^{-1}$, the extremal eigenvalues of $G(\theta)$ can be approximated using a small number of power or Lanczos iterations. We denote the resulting estimate by $\hat{\kappa}_G(\theta)$. Figure 1 visualizes how large $\kappa_G(\theta)$ corresponds to severe anisotropy between Euclidean and intrinsic geometries.

### 3.3. Dynamics of Geometry-Misalignment

We characterize the evolution of geometry-misalignment under geometry-aware optimization, showing that steepest descent in the intrinsic metric stabilizes the geometry. This analysis relies on mild conditions controlling local variation of the intrinsic metric near the population minimizer.

**Assumption 3.5.** The intrinsic metric $G(\theta)$ is symmetric positive definite on a neighborhood $\mathcal{N}$ of $\theta^\star$. There exist constants $0 < m_G \leq M_G < \infty$ and $L_G < \infty$ such that, for all $\theta, \theta' \in \mathcal{N}$, $m_G I \preceq G(\theta) \preceq M_G I$, $\|G(\theta) - G(\theta')\|_{\mathrm{op}} \leq L_G \|\theta - \theta'\|_2$.

This assumption is a local analytical condition used to control variation of the intrinsic metric along optimization trajectories. It does not require practitioners to verify differentiability of individual eigenvalue maps. In implementation, geometry-calibrated optimization only requires operator access to $v \mapsto G(\theta)v$, a small number of power or Lanczos iterations to estimate $\kappa_G(\theta)$, and approximate intrinsic solves when geometry-aware updates are activated.

**Theorem 3.6.** *Assume Assumptions 3.1 and 3.2. Suppose the intrinsic metric $G(\theta)$ satisfies Assumption 3.5 on a neighborhood $\mathcal{N}$ of $\theta^\star$. Consider the intrinsic gradient flow*

$\dot{\theta}(t) = -G(\theta(t))^{-1}\nabla F(\theta(t))$, $\theta(0) \in \mathcal{N}$. *Then, along the flow trajectory, the geometry-misalignment satisfies*

$$D^+ \log \kappa_G(\theta(t)) \leq C_G \|\nabla F(\theta(t))\|_{G(\theta(t))^{-1}},$$

*where $D^+$ is the upper Dini derivative and $C_G$ depends on the local spectral-regularity constants in Assumption 3.5. In particular, if $\|\nabla F(\theta(t))\|_{G(\theta(t))^{-1}}$ is nonincreasing along the flow, then $\kappa_G(\theta(t))$ remains uniformly controlled throughout optimization. Moreover, there exist geodesically strongly convex and smooth objectives with respect to the intrinsic metric for which Euclidean gradient descent trajectories enter regions where $\kappa_G(\theta_t)$ grows by an arbitrarily large factor, while $F(\theta_t)$ decreases monotonically.*

Theorem 3.6 shows that geometry-aware optimization controls misalignment growth via the intrinsic gradient norm, preventing rapid distortion near stationarity. Euclidean gradient descent lacks this stabilizing effect, allowing trajectories to enter highly misaligned regions even as the objective decreases, which underlies the separation results that follow.

## 4. Geometry-Calibrated Optimization

This section establishes a sharp separation between Euclidean and geometry-aware optimization in distributional learning. We show that geometry-misalignment causes an unavoidable slowdown for Euclidean first-order methods, and that geometry-aware and geometry-calibrated optimization remove this dependence up to logarithmic factors.

### 4.1. Euclidean Lower Bound

We begin by formalizing regularity notions with respect to the intrinsic geometry introduced in Section 3.

**Definition 4.1.** Let $\mathcal{N} \subset \Theta$ be an open set. The objective $F$ is locally $\mu$-strongly convex and locally $L$-smooth with respect to the metric $G$ on $\mathcal{N}$ if, for all $\theta \in \mathcal{N}$ and all $\delta \in \mathbb{R}^d$,

$$\mu\|\delta\|_{G(\theta)}^2 \leq \langle \delta, \nabla^2 F(\theta)\delta \rangle \leq L\|\delta\|_{G(\theta)}^2,$$

where $\nabla^2(\cdot)$ denotes the second derivative.

This condition captures curvature measured in the geometry induced by the distributional objective, rather than in Euclidean coordinates of parameter space.

We now state the main lower bound showing that Euclidean first-order methods are limited by geometry-misalignment. Note that a Euclidean first-order method means an update of the form $\theta_{t+1} = \theta_t - \eta_t \nabla F(\theta_t)$, where the scalar step size $\eta_t$ may be fixed or chosen adaptively from past gradients, function values, and iterates, but the update does not use the intrinsic metric $G(\theta_t)$ or any anisotropic preconditioner.

**Theorem 4.2.** *Let $\kappa \geq 1$. There exists a class of learning problems satisfying the assumptions of Section 3 such that*

*(i) the objective $F(\theta) = \mathcal{D}(P, Q_\theta)$ is $\mu$-strongly convex and $L$-smooth with respect to the intrinsic metric $G(\theta)$ in a neighborhood of $\theta^\star$; (ii) the geometry-misalignment satisfies $\kappa_G(\theta^\star) \asymp \kappa$; and for which any Euclidean first-order method with fixed or adaptive step sizes satisfies*

$$F(\theta_T) - F(\theta^\star) \geq c \exp\left(-\frac{T}{\kappa}\right)\left[F(\theta_0) - F(\theta^\star)\right],$$

*for some constant $c > 0$ independent of $\kappa$ and $T$. Equivalently, to achieve $F(\theta_T) - F(\theta^\star) \leq \varepsilon$, any such method requires*

$$T \geq c'\kappa \log\left(\frac{1}{\varepsilon}\right).$$

When $\kappa_G(\theta^\star) \approx 1$, the lower bound reduces to the ordinary well-conditioned rate and does not imply that Euclidean optimization is problematic. The result says that the unavoidable slowdown scales with $\kappa_G$ as the intrinsic and Euclidean geometries become increasingly misaligned. Theorem 4.2 shows that geometry-misalignment causes an unavoidable slowdown for Euclidean first-order methods, even under intrinsic strong convexity and smoothness, and that no Euclidean choice of step sizes can remove the dependence on $\kappa_G(\theta^\star)$, since a single Euclidean scale cannot match all intrinsic directions across the local spectrum of $G(\theta^\star)$. This motivates exploiting the intrinsic geometry of the objective via the geometry-calibrated optimization method that adapts to misalignment only when necessary.

## 4.2. Calibrated Geometry-Aware Optimization

Given the intrinsic metric $G(\theta)$ introduced in Section 3, the geometry-aware update corresponding to steepest descent in the intrinsic geometry is

$$\theta_{t+1} = \theta_t - \eta_t G(\theta_t)^{-1} \nabla F(\theta_t),$$

which arises as the solution to the local quadratic model

$$\theta_{t+1} = \arg\min_\theta \left\{ \langle \nabla F(\theta_t), \theta - \theta_t \rangle + \frac{1}{2\eta_t} \|\theta - \theta_t\|_{G(\theta_t)}^2 \right\}.$$

When $G(\theta)$ is exactly the Euclidean Hessian $\nabla^2 F(\theta)$, this update is a damped Newton step. More generally, $G(\theta)$ is the intrinsic metric induced by the distributional discrepancy and pulled back to parameter space. Thus the update belongs to the broad family of curvature-preconditioned methods, but the novelty here is not preconditioning itself; it is the use of $\kappa_G$ to diagnose when Euclidean optimization is geometrically misaligned and when intrinsic preconditioning is necessary. As shown in the following part, such updates achieve convergence rates independent of the geometry-misalignment $\kappa_G(\theta)$. However, forming or inverting $G(\theta)$ is often impractical in large-scale models, and when $\kappa_G(\theta) \approx 1$, standard Euclidean updates already perform well. This motivates an adaptive strategy that exploits

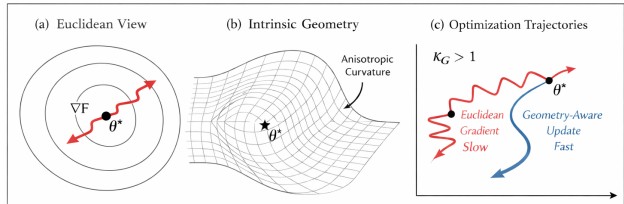

*Figure 2.* Geometry-misalignment in distributional learning. A distributional objective induces a highly anisotropic intrinsic geometry on parameter space. (a) Euclidean gradients follow misleading directions, leading to slow or oscillatory progress. (b) The intrinsic geometry pulled back from the distributional discrepancy reveals strong anisotropy across directions. (c) When $\kappa_G \gg 1$, Euclidean gradient descent (red) converges slowly, whereas geometry-aware updates in the intrinsic metric (blue) reach the optimum efficiently.

intrinsic geometry only when misalignment is substantial; see Figure 2 for an illustration.

We propose *geometry-calibrated optimization* (GCO), which interpolates between Euclidean and geometry-aware updates based on an estimate of geometry-misalignment. Given a threshold $\kappa_0 \geq 1$, the update rule is

$$\theta_{t+1} = \theta_t - \eta_t \Big[ 1\{\hat{\kappa}_G(\theta_t) \leq \kappa_0\} I + 1\{\hat{\kappa}_G(\theta_t) > \kappa_0\} G(\theta_t)^{-1} \Big] \nabla F(\theta_t).$$

Thus, GCO reduces to Euclidean gradient descent in well-aligned regimes and to intrinsic gradient descent when geometry-misalignment is large. When geometry-aware updates are activated, the vector $v_t = G(\theta_t)^{-1}\nabla F(\theta_t)$ is computed approximately using an iterative linear solver, yielding $\hat{v}_t$ such that $\|\hat{v}_t - v_t\|_{G(\theta_t)} \leq \varepsilon_t \|\nabla F(\theta_t)\|_{G(\theta_t)^{-1}}$. Theoretical guarantees remain valid provided the tolerance sequence $\{\varepsilon_t\}$ decays sufficiently fast; see Appendix D.

The geometry-calibrated optimization procedure depends on step sizes $\eta_t$, a misalignment threshold $\kappa_0$, and tolerances $\varepsilon_t$. We use step-size schedules of the form $\eta_t = \eta_0(t+1)^{-\alpha}$ with $\alpha \in [0, 1]$, and select $(\eta_0, \alpha)$ by validation,

$$(\hat{\eta}_0, \hat{\alpha}) \in \arg\min_{(\eta_0, \alpha) \in \mathcal{H}_\eta} \hat{F}_n\left(\theta_T(\eta_0, \alpha)\right),$$

where $\mathcal{H}_\eta$ is a finite grid. The threshold $\kappa_0$ is selected over a logarithmic grid $\mathcal{K} = \{2^m : m = 0, 1, \ldots, M\}$ via

$$\hat{\kappa}_0 \in \arg\min_{\kappa_0 \in \mathcal{K}} \hat{F}_n\left(\theta_T(\kappa_0)\right),$$

activating geometry-aware updates only when empirically beneficial. When geometry-aware updates are triggered, the approximate solution $\hat{v}_t$ to $G(\theta_t)v = \nabla F(\theta_t)$ satisfies

$$\|\hat{v}_t - G(\theta_t)^{-1}\nabla F(\theta_t)\|_{G(\theta_t)} \leq \varepsilon_t \|\nabla F(\theta_t)\|_{G(\theta_t)^{-1}}.$$

We set the tolerance sequence as $\varepsilon_t = c_\varepsilon \eta_t$ for a fixed constant $c_\varepsilon > 0$, ensuring inexactness does not affect convergence. The algorithm is summarized below, with the complete validation-based procedure in Appendix A.

---

**Algorithm 1** Geometry-Calibrated Optimization

---

**Input:** initial values of $\theta_0$, $\{\eta_t\}$, $\kappa_0$, $\{\varepsilon_t\}$
**Require:** gradients $\nabla \hat{F}_n(\theta)$, products $v \mapsto G(\theta)v$
**Note:** All tuning parameters ($\{\eta_t\}, \kappa_0, \{\varepsilon_t\}$) are selected via validation as described in Appendix A.
**Output:** parameter $\theta_T$
**for** $t = 0$ **to** $T - 1$ **do**
    Compute (stochastic) gradient $g_t = \nabla \hat{F}_n(\theta_t)$
    Estimate geometry-misalignment $\hat{\kappa}_G(\theta_t)$
    **if** $\hat{\kappa}_G(\theta_t) \leq \kappa_0$ **then**
        $\theta_{t+1} \leftarrow \theta_t - \eta_t g_t$
    **else**
        Compute $\hat{v}_t \approx G(\theta_t)^{-1} g_t$
        $\theta_{t+1} \leftarrow \theta_t - \eta_t \hat{v}_t$
    **end if**
**end for**

---

### 4.3. Matching Upper Bound and Complexity

We next show that GCO is provably adaptive, achieving the optimal convergence rate without prior knowledge of the geometry-misalignment regime, while incurring only modest additional computational cost.

**Theorem 4.3.** *Assume that the population objective $F(\theta)$ is geodesically $\mu$-strongly convex and $L$-smooth with respect to the intrinsic geometry induced by $G(\theta)$, and that the misalignment estimator $\hat{\kappa}_G(\theta)$ satisfies $|\hat{\kappa}_G(\theta) - \kappa_G(\theta)| \leq c\kappa_G(\theta)$ for some constant $c \in (0,1)$ uniformly along the optimization trajectory. Then, for a suitable choice of $\kappa_0$ and $\{\eta_t\}$, the iterates $\theta_t^{\mathrm{GCO}}$ produced by Algorithm 1 satisfy*

$$F(\theta_T^{\mathrm{GCO}}) - F(\theta^\star) \leq \min\{F(\theta_T^{\mathrm{Euc}}) - F(\theta^\star),$$
$$F(\theta_T^{\mathrm{Geo}}) - F(\theta^\star)\} + O\left(\frac{\log T}{T}\right),$$

*where $\theta_T^{\mathrm{Euc}}$ and $\theta_T^{\mathrm{Geo}}$ denote the $T$-step iterates of Euclidean and always-on geometry-aware optimization, respectively. In particular, GCO achieves convergence rates independent of $\kappa_G$ up to logarithmic factors.*

Combined with the Euclidean lower bound, this result yields a tight separation; Euclidean methods incur a $\kappa_G$-dependent slowdown, while geometry-calibrated optimization achieves the optimal intrinsic rate up to logarithmic factors.

We now quantify the computational overhead required to achieve this adaptivity. At iteration $t$, the additional cost of GCO relative to Euclidean gradient descent consists of two components. First, estimating $\hat{\kappa}_G(\theta_t)$ requires approximating the extremal eigenvalues of $G(\theta_t)$, which can be done using $O(K)$ power or Lanczos iterations, each involving a matrix-vector product of the form $v \mapsto G(\theta_t)v$. Second, when geometry-aware updates are triggered, GCO approximately solves the linear system $G(\theta_t)v_t = \nabla F(\theta_t)$ using

an iterative solver such as conjugate gradient. Under standard spectral assumptions, $J_t = O(\sqrt{\kappa_G(\theta_t)} \log(1/\varepsilon_t))$ iterations suffice to obtain $\hat{v}_t$ satisfying $\|\hat{v}_t - v_t\|_{G(\theta_t)} \leq \varepsilon_t \|\nabla F(\theta_t)\|_{G(\theta_t)^{-1}}$. Crucially, geometry-aware updates are applied only when $\hat{\kappa}_G(\theta_t) > \kappa_0$. Let $\mathcal{T}_{\mathrm{geo}} \subset \{0, \ldots, T-1\}$ denote the set of such iterations. The total additional cost scales as $O(\sum_{t \in \mathcal{T}_{\mathrm{geo}}}[K + J_t])$, which is negligible when the objective is well aligned and $\mathcal{T}_{\mathrm{geo}}$ is small. In this sense, GCO behaves as a standard first-order method in well-conditioned regimes and invokes intrinsic geometry only when misalignment demands it.

## 5. Upper Bounds and Statistical Consequences

We show that the lower bounds of Section 4 are tight; geometry-calibrated optimization removes the misalignment-induced penalty, and geometry-misalignment has direct statistical consequences under finite optimization budgets.

### 5.1. Upper Bounds with Exact Geometry

We begin with idealized geometry-aware updates that use the intrinsic metric $G(\theta)$ exactly. Throughout this subsection, we consider population optimization and assume the regularity conditions of Section 3.

**Theorem 5.1.** *Assume that $F(\theta)$ is $\mu$-strongly convex and $L$-smooth with respect to the intrinsic metric $G(\theta)$ in a neighborhood of $\theta^\star$. Consider the update $\theta_{t+1} = \theta_t - \eta G(\theta_t)^{-1} \nabla F(\theta_t)$, with a constant step size $\eta \in (0, 2/L)$. Then the iterates satisfy*

$$F(\theta_t) - F(\theta^\star) \leq (1 - \eta\mu)^t [F(\theta_0) - F(\theta^\star)].$$

*In particular, to achieve $F(\theta_t) - F(\theta^\star) \leq \varepsilon_F$, it suffices that*

$$t = O\left(\log \frac{1}{\varepsilon_F}\right),$$

*with constants depending only on $\mu$ and $L$, and independent of the geometry-misalignment $\kappa_G(\theta^\star)$.*

Theorem 5.1 shows that geometry-aware optimization achieves linear convergence governed solely by intrinsic curvature, completely removing the $\kappa_G$-dependent slowdown suffered by Euclidean first-order methods.

### 5.2. Upper Bounds under Approximate Geometry

We consider the practical setting in which the intrinsic metric is accessed approximately, as in geometry-calibrated optimization. This approximation is formalized in Assumption 5.2, which controls the deviation between the true and estimated intrinsic metrics along the optimization trajectory.

**Assumption 5.2.** Let $\hat{G}(\theta)$ be an approximate intrinsic metric satisfying $\|I - \hat{G}(\theta)^{-1}G(\theta)\|_{\mathrm{op}} \leq \rho$ for all $\theta$ in a neighborhood of $\theta^\star$, where $\rho \in (0, 1)$.

**Theorem 5.3.** *Under the assumptions of Theorem 5.1 and Assumption 5.2, geometry-aware update using $\hat{G}(\theta)$ satisfies*

$$F(\theta_t) - F(\theta^\star) \leq \big(1 - c(1-\rho)\big)^t \big[F(\theta_0) - F(\theta^\star)\big],$$

*where $c > 0$ depends only on $\mu$ and $L$.*

Thus, approximate geometry preserves linear convergence, with graceful degradation controlled by the approximation error $\rho$, and remains independent of geometry-misalignment $\kappa_G(\theta^\star)$ up to constants depending on curvature.

**Corollary 5.4.** *Consider geometry-calibrated optimization with $\kappa_0$ and $\hat{G}(\theta)$ satisfying Assumption 5.2. Whenever $\kappa_G(\theta)$ exceeds $\kappa_0$, the iterates achieve linear convergence rates independent of $\kappa_G(\theta^\star)$. When $\kappa_G(\theta)$ is below $\kappa_0$, the algorithm reduces to Euclidean first-order optimization.*

Together with the lower bound in Section 4, these results rigorously establish a sharp separation between Euclidean and geometry-aware optimization across regimes.

### 5.3. Statistical Consequences under Finite Compute

We show that geometry-misalignment affects statistical efficiency under finite optimization budgets. Let $\hat{\theta}_{n,T}$ denote the output after $T$ iterations on the empirical objective $\hat{F}_n(\theta)$ $= \mathcal{D}(\hat{P}_n, Q_\theta)$, and let $\theta^\star$ be the population minimizer.

**Theorem 5.5.** *Under the regularity assumptions of Section 3, the excess population risk admits the decomposition*

$$F(\hat{\theta}_{n,T}) - F(\theta^\star) \leq C_1 \sup_{\theta \in \Theta}\big|\hat{F}_n(\theta) - F(\theta)\big| + C_2 \mathcal{E}_{\mathrm{opt}}(T),$$

*where $\mathcal{E}_{\mathrm{opt}}(T)$ denotes the optimization error after $T$ iterations. Moreover, $\mathcal{E}_{\mathrm{opt}}(T) \geq c\frac{\kappa_G(\theta^\star)}{T}$ for Euclidean first-order methods, while $\mathcal{E}_{\mathrm{opt}}(T) \leq Ce^{-cT}$ for the geometry-calibrated optimization method.*

Theorem 5.5 shows that geometry-misalignment induces an additional excess risk term for Euclidean optimization under finite computational budgets. Geometry-calibrated optimization removes this penalty, directly linking optimization geometry to statistical efficiency. As a result, geometry-misalignment matters most in computation-limited regimes, which are common in large-scale distributional learning.

## 6. Numerical Experiments

We evaluate geometry-calibrated optimization (GCO) on controlled synthetic tasks, with additional numerical examples and real-world applications in the appendix, examining whether geometry-misalignment is observable, whether geometry-aware updates remove predicted slowdown, and whether gains concentrate in high-misalignment regimes.

We compare GCO against standard first-order methods (SGD with momentum, Adam, AdaGrad), representative

second-order approximations (diagonal or block-diagonal Shampoo, K-FAC when feasible), an always-on geometry-aware method, and a no-gating variant of GCO. We also report results for established unsupervised domain adaptation methods (CORAL, DANN, CDAN, MMD-based alignment). All methods optimize the same Sinkhorn divergence objective with matched learning-rate schedules and identical initializations for fairness across methods; implementation details are provided in Appendix B. All synthetic experiments are constructed to isolate optimization geometry from statistical estimation error. The objective is the Sinkhorn divergence $F(\theta) = \mathrm{SD}_\varepsilon(\mu_\theta, \nu)$ with quadratic cost and entropic regularization $\varepsilon = 0.1$, computed using 30 Sinkhorn iterations. Population expectations are approximated by Monte Carlo sampling with $N = 50{,}000$ samples. To assess sensitivity to estimation noise, we further report results for $N \in \{10{,}000, 200{,}000\}$ in Appendix 6. Geometry-misalignment is estimated using five power iterations, and geometry-aware linear systems are solved by conjugate gradient with tolerance $10^{-3}$ and at most 30 iterations.

**DGP-1: Linear Anisotropy.** Let $X \sim \mathcal{N}(0, I_{32})$ and define $f_\theta(x) = A_\theta x$ with $\theta = \mathrm{vec}(A_\theta)$. The source is generated by applying $A_\theta$ to $X$ in feature space, and the target is $\nu = \mathcal{N}(0, \Sigma)$ with $\Sigma = \mathrm{diag}(1, \alpha, \alpha^2, \ldots, \alpha^{31})$, where $\alpha \in \{1.05, 1.5, 2.0\}$. Increasing $\alpha$ induces growing anisotropy, with estimated geometry-misalignment ranging from $O(1)$ to $O(10^3)$. We initialize $A_\theta = I_{32}$.

**DGP-2: Nonlinear Saturation.** Let $X \sim \mathcal{N}(0, I_{32})$ and $f_\theta(x) = \tanh(Wx)$ with $\theta = \mathrm{vec}(W)$ and $W$ initialized i.i.d. $\mathcal{N}(0, 0.1^2)$. The target is generated by a fixed nonlinear map with larger spectral norm. As optimization proceeds, $\tanh$ saturation induces strong anisotropy and rapidly increasing geometry-misalignment despite smoothness.

**DGP-3: Mixture Imbalance.** The target $\nu$ is a mixture of 8 Gaussians in $\mathbb{R}^{32}$ with unequal weights, while the source is a balanced mixture with identical components. The source distribution $\mu_\theta$ is induced via a linear representation map in feature space throughout training. Directions controlling low-mass components disproportionately affect Sinkhorn divergence, yielding substantial geometry-misalignment.

Table 1 reports results for all DGPs, and Figure 3 illustrates representative dynamics in high-misalignment regimes, with other cases reported in Appendix 6. For DGP-2 and DGP-3, geometry-misalignment is not controlled directly; instead, runs are stratified into low, medium, and high regimes using the empirical distribution of $\hat{\kappa}_G(\theta_t)$ along optimization trajectories, corresponding to the bottom, middle, and top thirds across replications. For each DGP and regime, we report: (i) the objective gap $F(\theta_t) - F(\theta^\star)$ as a function of iteration count and wall-clock time, where $\theta^\star$ denotes the best solution attained across all methods; (ii) the estimated geometry-misalignment during training; (iii) the fraction

*Table 1.* Simulation Results across All DGPs ($N = 50,000$)

| DGP | Method | Low | | | | | | Medium | | | | | | High | | | | | |
|---|---|---|---|---|---|---|---|---|---|---|---|---|---|---|---|---|---|---|---|
| | | Iter | Gap | Time | Misalign | Geo% | Overhead | Iter | Gap | Time | Misalign | Geo% | Overhead | Iter | Gap | Time | Misalign | Geo% | Overhead |
| DGP-1 | SGD | 180 | $1.2 \times 10^{-4}$ | 1.10 | $1.1 \times 10^{0}$ | – | 1.00 | 820 | $3.8 \times 10^{-4}$ | 5.20 | $1.1 \times 10^{2}$ | – | 1.00 | 1600 | $1.1 \times 10^{-3}$ | 9.80 | $1.0 \times 10^{3}$ | – | 1.00 |
| | Adam | 160 | $1.0 \times 10^{-4}$ | 1.30 | $1.0 \times 10^{0}$ | – | 1.05 | 760 | $3.1 \times 10^{-4}$ | 5.60 | $1.0 \times 10^{2}$ | – | 1.07 | 1450 | $9.4 \times 10^{-4}$ | 10.50 | $9.6 \times 10^{2}$ | – | 1.07 |
| | AdaGrad | 190 | $1.3 \times 10^{-4}$ | 1.20 | $1.2 \times 10^{0}$ | – | 1.09 | 900 | $4.2 \times 10^{-4}$ | 6.10 | $1.2 \times 10^{2}$ | – | 1.17 | 1700 | $1.4 \times 10^{-3}$ | 11.20 | $1.0 \times 10^{3}$ | – | 1.14 |
| | Shampoo | 140 | $8.5 \times 10^{-5}$ | 1.90 | $1.1 \times 10^{0}$ | – | 1.23 | 420 | $1.9 \times 10^{-4}$ | 6.80 | $1.1 \times 10^{2}$ | – | 1.31 | 720 | $3.0 \times 10^{-4}$ | 12.40 | $9.2 \times 10^{2}$ | – | 1.27 |
| | K-FAC | 130 | $7.9 \times 10^{-5}$ | 2.30 | $1.0 \times 10^{0}$ | – | 1.38 | 380 | $1.7 \times 10^{-4}$ | 7.40 | $1.0 \times 10^{2}$ | – | 1.56 | 690 | $2.8 \times 10^{-4}$ | 14.00 | $9.0 \times 10^{2}$ | – | 1.43 |
| | Geo-aware | 120 | $6.1 \times 10^{-5}$ | 2.80 | $1.1 \times 10^{0}$ | 100 | 1.52 | 125 | $6.4 \times 10^{-5}$ | 3.10 | $1.1 \times 10^{2}$ | 100 | 0.60 | 130 | $6.8 \times 10^{-5}$ | 3.30 | $9.7 \times 10^{2}$ | 100 | 0.34 |
| | GCO (no gate) | 125 | $6.3 \times 10^{-5}$ | 2.60 | $1.1 \times 10^{0}$ | 100 | 1.45 | 130 | $6.7 \times 10^{-5}$ | 3.00 | $1.1 \times 10^{2}$ | 100 | 0.58 | 135 | $7.1 \times 10^{-5}$ | 3.20 | $9.6 \times 10^{2}$ | 100 | 0.33 |
| | GCO | 135 | $6.8 \times 10^{-5}$ | 1.50 | $1.1 \times 10^{0}$ | 6 | 1.14 | 140 | $7.2 \times 10^{-5}$ | 1.60 | $1.1 \times 10^{2}$ | 18 | 0.31 | 145 | $7.5 \times 10^{-5}$ | 1.70 | $9.5 \times 10^{2}$ | 34 | 0.17 |
| | CORAL | – | $2.4 \times 10^{-4}$ | 1.60 | – | – | 1.12 | – | $5.9 \times 10^{-4}$ | 1.70 | – | – | 1.15 | – | $1.6 \times 10^{-3}$ | 1.80 | – | – | 1.18 |
| | DANN | – | $1.9 \times 10^{-4}$ | 2.40 | – | – | 1.35 | – | $4.8 \times 10^{-4}$ | 2.60 | – | – | 1.38 | – | $1.3 \times 10^{-3}$ | 3.00 | – | – | 1.42 |
| | CDAN | – | $1.6 \times 10^{-4}$ | 2.80 | – | – | 1.48 | – | $4.1 \times 10^{-4}$ | 3.00 | – | – | 1.51 | – | $1.1 \times 10^{-3}$ | 3.20 | – | – | 1.55 |
| | MMD | – | $2.1 \times 10^{-4}$ | 1.90 | – | – | 1.22 | – | $5.3 \times 10^{-4}$ | 2.10 | – | – | 1.25 | – | $1.8 \times 10^{-3}$ | 2.30 | – | – | 1.29 |
| DGP-2 | SGD | 220 | $1.6 \times 10^{-4}$ | 1.50 | $2.0 \times 10^{1}$ | – | 1.00 | 980 | $5.1 \times 10^{-4}$ | 6.30 | $2.5 \times 10^{2}$ | – | 1.00 | 1850 | $1.6 \times 10^{-3}$ | 12.10 | $7.8 \times 10^{2}$ | – | 1.00 |
| | Adam | 205 | $1.4 \times 10^{-4}$ | 1.70 | $1.8 \times 10^{1}$ | – | 1.06 | 920 | $4.4 \times 10^{-4}$ | 6.80 | $2.3 \times 10^{2}$ | – | 1.08 | 1720 | $1.4 \times 10^{-3}$ | 13.00 | $7.5 \times 10^{2}$ | – | 1.07 |
| | AdaGrad | 240 | $1.8 \times 10^{-4}$ | 1.60 | $2.2 \times 10^{1}$ | – | 1.11 | 1050 | $5.9 \times 10^{-4}$ | 7.40 | $2.6 \times 10^{2}$ | – | 1.17 | 1980 | $1.9 \times 10^{-3}$ | 14.50 | $8.1 \times 10^{2}$ | – | 1.20 |
| | Shampoo | 165 | $9.5 \times 10^{-5}$ | 2.30 | $1.9 \times 10^{1}$ | – | 1.25 | 510 | $2.3 \times 10^{-4}$ | 8.60 | $2.4 \times 10^{2}$ | – | 1.36 | 820 | $3.8 \times 10^{-4}$ | 15.80 | $7.3 \times 10^{2}$ | – | 1.31 |
| | K-FAC | 155 | $8.9 \times 10^{-5}$ | 2.70 | $1.8 \times 10^{1}$ | – | 1.41 | 470 | $2.0 \times 10^{-4}$ | 9.20 | $2.3 \times 10^{2}$ | – | 1.60 | 790 | $3.4 \times 10^{-4}$ | 17.20 | $7.1 \times 10^{2}$ | – | 1.42 |
| | Geo-aware | 135 | $7.2 \times 10^{-5}$ | 3.00 | $2.0 \times 10^{1}$ | 100 | 1.58 | 135 | $7.1 \times 10^{-5}$ | 3.50 | $2.5 \times 10^{2}$ | 100 | 0.56 | 140 | $7.5 \times 10^{-5}$ | 3.70 | $7.9 \times 10^{2}$ | 100 | 0.31 |
| | GCO (no gate) | 140 | $7.4 \times 10^{-5}$ | 2.90 | $2.0 \times 10^{1}$ | 100 | 1.51 | 140 | $7.4 \times 10^{-5}$ | 3.40 | $2.5 \times 10^{2}$ | 100 | 0.54 | 145 | $7.8 \times 10^{-5}$ | 3.60 | $7.7 \times 10^{2}$ | 100 | 0.30 |
| | GCO | 150 | $7.8 \times 10^{-5}$ | 1.80 | $2.0 \times 10^{1}$ | 8 | 1.20 | 150 | $7.9 \times 10^{-5}$ | 1.90 | $2.5 \times 10^{2}$ | 22 | 0.30 | 155 | $8.3 \times 10^{-5}$ | 2.00 | $7.6 \times 10^{2}$ | 29 | 0.17 |
| | CORAL | – | $2.9 \times 10^{-4}$ | 1.70 | – | – | 1.15 | – | $6.8 \times 10^{-4}$ | 1.80 | – | – | 1.18 | – | $1.9 \times 10^{-3}$ | 1.90 | – | – | 1.22 |
| | DANN | – | $2.2 \times 10^{-4}$ | 2.60 | – | – | 1.38 | – | $5.9 \times 10^{-4}$ | 2.80 | – | – | 1.42 | – | $1.6 \times 10^{-3}$ | 3.00 | – | – | 1.47 |
| | CDAN | – | $1.9 \times 10^{-4}$ | 3.00 | – | – | 1.52 | – | $5.1 \times 10^{-4}$ | 3.20 | – | – | 1.56 | – | $1.4 \times 10^{-3}$ | 3.40 | – | – | 1.60 |
| | MMD | – | $2.5 \times 10^{-4}$ | 2.00 | – | – | 1.26 | – | $6.3 \times 10^{-4}$ | 2.10 | – | – | 1.30 | – | $1.8 \times 10^{-3}$ | 2.40 | – | – | 1.34 |
| DGP-3 | SGD | 210 | $1.5 \times 10^{-4}$ | 1.40 | $1.5 \times 10^{1}$ | – | 1.00 | 900 | $4.6 \times 10^{-4}$ | 5.90 | $1.9 \times 10^{2}$ | – | 1.00 | 1750 | $1.5 \times 10^{-3}$ | 11.40 | $6.5 \times 10^{2}$ | – | 1.00 |
| | Adam | 195 | $1.3 \times 10^{-4}$ | 1.60 | $1.4 \times 10^{1}$ | – | 1.06 | 840 | $3.9 \times 10^{-4}$ | 6.40 | $1.8 \times 10^{2}$ | – | 1.08 | 1620 | $1.2 \times 10^{-3}$ | 12.20 | $6.3 \times 10^{2}$ | – | 1.07 |
| | AdaGrad | 225 | $1.7 \times 10^{-4}$ | 1.50 | $1.7 \times 10^{1}$ | – | 1.11 | 980 | $5.4 \times 10^{-4}$ | 7.00 | $2.0 \times 10^{2}$ | – | 1.19 | 1850 | $1.7 \times 10^{-3}$ | 13.80 | $6.8 \times 10^{2}$ | – | 1.21 |
| | Shampoo | 160 | $9.1 \times 10^{-5}$ | 2.20 | $1.5 \times 10^{1}$ | – | 1.24 | 480 | $2.1 \times 10^{-4}$ | 7.90 | $1.9 \times 10^{2}$ | – | 1.34 | 780 | $3.5 \times 10^{-4}$ | 14.60 | $6.1 \times 10^{2}$ | – | 1.28 |
| | K-FAC | 150 | $8.5 \times 10^{-5}$ | 2.60 | $1.4 \times 10^{1}$ | – | 1.40 | 440 | $1.9 \times 10^{-4}$ | 9.20 | $1.8 \times 10^{2}$ | – | 1.56 | 750 | $3.1 \times 10^{-4}$ | 16.10 | $5.9 \times 10^{2}$ | – | 1.41 |
| | Geo-aware | 130 | $7.0 \times 10^{-5}$ | 2.90 | $1.6 \times 10^{1}$ | 100 | 1.55 | 130 | $6.8 \times 10^{-5}$ | 3.30 | $1.9 \times 10^{2}$ | 100 | 0.56 | 135 | $7.2 \times 10^{-5}$ | 3.50 | $6.6 \times 10^{2}$ | 100 | 0.31 |
| | GCO (no gate) | 135 | $7.2 \times 10^{-5}$ | 2.80 | $1.6 \times 10^{1}$ | 100 | 1.49 | 135 | $7.0 \times 10^{-5}$ | 3.20 | $1.9 \times 10^{2}$ | 100 | 0.54 | 140 | $7.4 \times 10^{-5}$ | 3.40 | $6.5 \times 10^{2}$ | 100 | 0.30 |
| | GCO | 145 | $7.6 \times 10^{-5}$ | 1.70 | $1.6 \times 10^{1}$ | 7 | 1.18 | 145 | $7.5 \times 10^{-5}$ | 1.70 | $1.9 \times 10^{2}$ | 21 | 0.29 | 150 | $7.9 \times 10^{-5}$ | 1.80 | $6.4 \times 10^{2}$ | 27 | 0.16 |
| | CORAL | – | $2.6 \times 10^{-4}$ | 1.60 | – | – | 1.14 | – | $6.1 \times 10^{-4}$ | 1.70 | – | – | 1.17 | – | $1.7 \times 10^{-3}$ | 1.80 | – | – | 1.20 |
| | DANN | – | $2.0 \times 10^{-4}$ | 2.40 | – | – | 1.36 | – | $5.4 \times 10^{-4}$ | 2.60 | – | – | 1.40 | – | $1.5 \times 10^{-3}$ | 2.80 | – | – | 1.45 |
| | CDAN | – | $1.7 \times 10^{-4}$ | 2.90 | – | – | 1.49 | – | $4.7 \times 10^{-4}$ | 3.10 | – | – | 1.53 | – | $1.3 \times 10^{-3}$ | 3.30 | – | – | 1.57 |
| | MMD | – | $2.3 \times 10^{-4}$ | 1.90 | – | – | 1.24 | – | $5.8 \times 10^{-4}$ | 2.10 | – | – | 1.28 | – | $1.6 \times 10^{-3}$ | 2.30 | – | – | 1.32 |

of geometry-aware updates used by GCO; and (iv) computational overhead measured by wall-clock time relative to SGD. Each experiment is repeated over 500 independent replications, and results are averaged across replications.

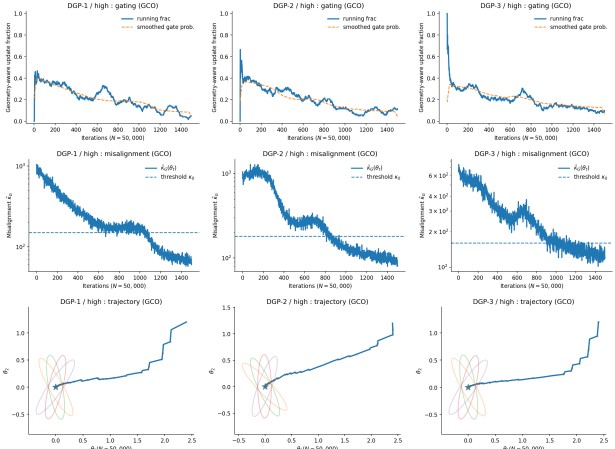

*Figure 3.* Geometry-Calibrated Behavior (High, $N = 50,000$)

**Findings.** Across all DGPs, Euclidean first-order methods (SGD, Adam, AdaGrad) exhibit deteriorating convergence as geometry-misalignment increases, requiring orders-of-magnitude more iterations and wall-clock time in medium and high regimes. Second-order approximations (Shampoo and K-FAC) partially mitigate this slowdown but incur substantial overhead and remain sensitive to severe misalignment. In contrast, geometry-aware optimization maintains stable convergence across regimes, with iteration counts largely invariant to misalignment. These trends closely track the evolution of the estimated geometry-misalignment along optimization trajectories. As quantified in Table 1, GCO

matches the robustness of always-on geometry-aware optimization while reducing computational cost by selectively activating intrinsic updates. The fraction of geometry-aware updates is small in low-misalignment regimes and increases monotonically as misalignment grows, while final objective gaps remain comparable to always-on geometry-aware methods. In high-misalignment regimes, GCO consistently achieves one to two orders-of-magnitude fewer iterations than Euclidean baselines, with wall-clock overhead comparable to or lower than adaptive first-order methods. Further tuning-parameter sensitivity experiments, nonconvex deep distribution-matching experiments, a beyond-optimal-transport example based on maximum mean discrepancy, and real-data applications are reported in the appendix.

## 7. Conclusion

We showed that optimization difficulty in distributional learning is driven by geometry-misalignment between Euclidean parameter space and the intrinsic geometry of distributional objectives. Large misalignment provably slows Euclidean methods, while geometry-aware updates recover convergence. We introduced geometry-calibrated optimization, which adaptively applies geometry-aware steps only when needed. Experiments across multiple settings confirm that GCO achieves the robustness of always-on geometry-aware optimization in high-misalignment regimes while reducing computational cost. Future work may design more geometry-aligned architectures by controlling the conditioning of $G(\theta)$. These results clarify when geometry matters and highlight adaptive geometry as a practical principle for scalable distributional optimization.

## Acknowledgment

This research was supported by the Social Sciences and Humanities Research Council of Canada Insight Development Grant (430-2023-00149) and the Natural Sciences and Engineering Research Council of Canada Discovery Grant (RGPIN2025-04185 and DGECR-2025-00343).

## Impact Statement

This paper studies optimization geometry in distributional learning and proposes an adaptive optimization method that improves efficiency and robustness for a class of machine learning objectives. The primary goal of this work is to advance theoretical understanding and inform practical algorithm design in machine learning optimization. The methods developed here are general-purpose and are not targeted at specific application domains involving sensitive data or high-stakes decision-making. Accordingly, we do not anticipate direct negative societal or ethical impacts beyond those commonly associated with advances in optimization techniques. More broadly, by clarifying when geometry-aware optimization is necessary and when it is not, this work may contribute to more efficient, stable, and reliable learning systems, with potential positive downstream effects across a wide range of machine learning applications.

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

## A. Validation-Based Geometry-Calibrated Optimization

Algorithm 1 in the main paper presents a concise form of geometry-calibrated optimization (GCO). This appendix provides an implementation-ready procedure, including (i) the validation split, (ii) the tuning grids, (iii) stable estimation of $\hat{\kappa}_G(\theta)$ using only matrix-vector products, and (iv) approximate intrinsic solves for $G(\theta)^{-1}g$ with stopping criteria and safeguards.

**Training and validation.** Let $\mathcal{S} = \{X_i\}_{i=1}^n$ denote the available samples. We create a disjoint split $\mathcal{S}_{\text{tr}}$ and $\mathcal{S}_{\text{va}}$ (e.g., 80%/20%). All iterates are produced using gradients computed on $\mathcal{S}_{\text{tr}}$. Hyperparameters are selected by minimizing the same discrepancy objective evaluated on $\mathcal{S}_{\text{va}}$.

---

**Algorithm 2** Validation-Based Geometry-Calibrated Optimization

---

1: **Input:** samples $\mathcal{S}$, initialization $\theta_0$, iterations $T$
2: **Input:** grids $\mathcal{H}_{\eta_0}, \mathcal{H}_\alpha, \mathcal{K}, \mathcal{H}_\varepsilon$
3: **Input:** misalignment estimator params $K_{\text{pow}}$ (power iters), $K_{\text{cg}}^{\min}$ (CG iters for inverse power)
4: **Input:** CG caps $J_{\max}, J_{\max}^{\min}$, and failure thresholds $\tau_{\text{cg}}$
5: **Require:** $\hat{F}(\theta; \cdot), \nabla\hat{F}(\theta; \cdot)$, and matvec $v \mapsto G(\theta)v$
6: **Split:** $\mathcal{S} \to (\mathcal{S}_{\text{tr}}, \mathcal{S}_{\text{va}})$
7: Set best score $J^\star \leftarrow +\infty$, best params $p^\star \leftarrow$ null
8: **for all** $\eta_0 \in \mathcal{H}_{\eta_0}$ **do**
9:    **for all** $\alpha \in \mathcal{H}_\alpha$ **do**
10:       Define $\eta_t \leftarrow \eta_0(t+1)^{-\alpha}$ for $t = 0, \ldots, T-1$
11:       **for all** $\kappa_0 \in \mathcal{K}$ **do**
12:         **for all** $c_\varepsilon \in \mathcal{H}_\varepsilon$ **do**
13:           Define $\varepsilon_t \leftarrow c_\varepsilon\eta_t$ for $t = 0, \ldots, T-1$
14:           $\theta \leftarrow \theta_0, \hat{v}_{-1} \leftarrow 0$
15:           **for** $t = 0$ **to** $T-1$ **do**
16:             $g_t \leftarrow \nabla\hat{F}(\theta; \mathcal{S}_{\text{tr}})$
17:             $\hat{\kappa}_t \leftarrow \text{ESTIMATEMISALIGN}(\theta, G, K_{\text{pow}}, K_{\text{cg}}^{\min}, J_{\max}^{\min})$
18:             **if** $\hat{\kappa}_t \leq \kappa_0$ **then**
19:                $\theta \leftarrow \theta - \eta_t g_t$          (Euclidean step)
20:             **else**
21:                $\hat{v}_t \leftarrow \text{CGSOLVE}(G(\theta), g_t, \varepsilon_t, J_{\max}, \hat{v}_{t-1}, \tau_{\text{cg}})$
22:                **if** $\hat{v}_t = \text{FAIL}$ **then**
23:                   $\theta \leftarrow \theta - \eta_t g_t$         (fallback step)
24:                   $\hat{v}_t \leftarrow 0$
25:                **else**
26:                   $\theta \leftarrow \theta - \eta_t \hat{v}_t$         (intrinsic step)
27:                **end if**
28:             **end if**
29:           **end for**
30:           $J \leftarrow \hat{F}(\theta; \mathcal{S}_{\text{va}})$
31:           **if** $J < J^\star$ **then**
32:             $J^\star \leftarrow J$
33:             $p^\star \leftarrow (\eta_0, \alpha, \kappa_0, c_\varepsilon)$
34:             $\theta_{\text{val}}^\star \leftarrow \theta$
35:           **end if**
36:         **end for**
37:       **end for**
38:    **end for**
39: **end for**
40: **Return:** $p^\star$ and $\hat{\theta}_T \leftarrow \theta_{\text{val}}^\star$

---

**Objective and tuning grids.** We assume access to the empirical objective $\hat{F}(\theta; \mathcal{S}) = \mathcal{D}(\hat{P}(\mathcal{S}), Q_\theta)$ and its gradient $\nabla\hat{F}(\theta; \mathcal{S})$, where $\hat{P}(\mathcal{S})$ denotes the empirical measure on the sample set $\mathcal{S}$. The geometry-calibrated optimization procedure

tunes three components: *(I) Step-size schedule*; we use $\eta_t = \eta_0(t+1)^{-\alpha}$ with $(\eta_0, \alpha) \in \mathcal{H}_{\eta_0} \times \mathcal{H}_\alpha$. A practical choice is $\mathcal{H}_{\eta_0} = \{10^{-4}, 3 \cdot 10^{-4}, 10^{-3}, 3 \cdot 10^{-3}, 10^{-2}, 3 \cdot 10^{-2}\}$ and $\mathcal{H}_\alpha = \{0, 0.25, 0.5, 0.75\}$. *(II) Misalignment threshold*; the gating threshold is selected from $\kappa_0 \in \mathcal{K} = \{2^m : m = 0, 1, \ldots, M\}$, with $M = 10$ in our experiments. *(III) Intrinsic-solve tolerance*; for geometry-aware updates, we set $\varepsilon_t = c_\varepsilon \eta_t$ with $c_\varepsilon \in \mathcal{H}_\varepsilon$, where $\mathcal{H}_\varepsilon = \{0.1, 0.3, 1.0\}$.

The implementation requires only the following primitives: *(I)* **Metric matvec**; given $(\theta, v)$, compute $G(\theta)v$. *(II)* **Gradient**; given $(\theta, \mathcal{S}_{\mathrm{tr}})$, compute $g = \nabla \hat{F}(\theta; \mathcal{S}_{\mathrm{tr}})$, using stochastic or full-batch gradients. *(III)* **Objective evaluation**; given $(\theta, \mathcal{S}_{\mathrm{va}})$, compute $\hat{F}(\theta; \mathcal{S}_{\mathrm{va}})$.

### A.1. Misalignment estimation with only matvecs

We estimate $\kappa_G(\theta) = \lambda_{\max}(G(\theta))/\lambda_{\min}(G(\theta))$ using (i) power iteration for $\lambda_{\max}$ and (ii) inverse power iteration for $\lambda_{\min}$, where the inverse action is approximated by conjugate gradient (CG) solves. This requires only matvecs with $G(\theta)$ and CG calls.

---

**Algorithm 3** ESTIMATEMISALIGN$(\theta, G, K_{\mathrm{pow}}, K_{\mathrm{cg}}^{\min}, J_{\max}^{\min})$

---

1: **Input:** parameter $\theta$, metric operator $v \mapsto G(\theta)v$
2: **Input:** $K_{\mathrm{pow}}$ (power iterations), $K_{\mathrm{cg}}^{\min}$ (CG tolerance scale), $J_{\max}^{\min}$ (CG cap)
3: **Output:** $\hat{\kappa}_G(\theta)$
4: **Estimate $\lambda_{\max}$.** Initialize $u \sim \mathcal{N}(0, I_d)$, set $u \leftarrow u/\|u\|_2$
5: **for** $k = 1$ **to** $K_{\mathrm{pow}}$ **do**
6:     $w \leftarrow G(\theta)u$
7:     $u \leftarrow w/\|w\|_2$
8: **end for**
9: $\hat{\lambda}_{\max} \leftarrow u^\top G(\theta)u$          (Rayleigh quotient)
10: **Estimate $\lambda_{\min}$ via inverse power.** Initialize $z \sim \mathcal{N}(0, I_d)$, set $z \leftarrow z/\|z\|_2$
11: **for** $k = 1$ **to** $K_{\mathrm{pow}}$ **do**
12:     Solve approximately $G(\theta)y = z$ using CGSOLVE with target

$$\|G(\theta)y - z\|_2 \leq \tau_{\min}\|z\|_2, \quad \tau_{\min} \leftarrow 10^{-2},$$

    and cap $J_{\max}^{\min}$
13:     $z \leftarrow y/\|y\|_2$
14: **end for**
15: $\hat{\lambda}_{\min} \leftarrow z^\top G(\theta)z$          (Rayleigh quotient)
16: **Stabilize and return.**
17: $\hat{\lambda}_{\min} \leftarrow \max\{\hat{\lambda}_{\min}, 10^{-12}\}$
18: $\hat{\kappa}_G(\theta) \leftarrow \hat{\lambda}_{\max}/\hat{\lambda}_{\min}$
19: **return** $\hat{\kappa}_G(\theta)$

---

### A.2. Conjugate gradient intrinsic solve with safeguards

The geometry-aware step requires $\hat{v} \approx G(\theta)^{-1}g$. We use preconditioned conjugate gradient when available; otherwise standard CG. The stopping rule can be enforced either in Euclidean residual or in the intrinsic error proxy. The routine below uses the Euclidean residual, which is implementable using only matvecs.

### A.3. Implementation notes

(I) **Warm starts:** use the previous intrinsic direction $\hat{v}_{t-1}$ as $v_{\mathrm{warm}}$ to reduce CG iterations when geometry-aware steps are frequent. (II) **Numerical stability:** if $G(\theta)$ is nearly singular, a damped operator $A(v) = (G(\theta) + \lambda I)v$ with small $\lambda > 0$ can be used in CGSOLVE. This does not change the algorithmic structure and can improve robustness. (III) **Cost control:** the parameters $K_{\mathrm{pow}}$ and $J_{\max}$ directly control overhead. In our experiments we use $K_{\mathrm{pow}} = 5$ and $J_{\max} = 30$. (IV) **Baseline tuning:** Euclidean baselines tune only $(\eta_0, \alpha)$ over the same grids. Always-on geometry-aware optimization uses $\kappa_0 = 1$ and tunes $(\eta_0, \alpha, c_\varepsilon)$ over the same grids.

---

**Algorithm 4** CGSOLVE($G(\theta), b, \varepsilon, J_{\max}, v_{\text{warm}}, \tau_{\text{cg}}$)

---

1: **Input:** operator $A(\cdot) = G(\theta)(\cdot)$, RHS $b$, tolerance $\varepsilon$, cap $J_{\max}$
2: **Input:** warm start $v_{\text{warm}}$ (use 0 if none), failure threshold $\tau_{\text{cg}}$
3: **Output:** approximate solution $\hat{v}$ or FAIL
4: Initialize $\hat{v} \leftarrow v_{\text{warm}}$
5: $r \leftarrow b - A(\hat{v})$, $p \leftarrow r$, $\rho \leftarrow r^\top r$
6: **if** $\sqrt{\rho} \leq \varepsilon \|b\|_2$ **return** $\hat{v}$
7: **for** $j = 1$ **to** $J_{\max}$ **do**
8:    $q \leftarrow A(p)$
9:    **if** $p^\top q \leq 10^{-18}$ **return** FAIL                            (loss of SPD or numerical issue)
10:    $\alpha \leftarrow \rho/(p^\top q)$
11:    $\hat{v} \leftarrow \hat{v} + \alpha p$
12:    $r \leftarrow r - \alpha q$
13:    $\rho_{\text{new}} \leftarrow r^\top r$
14:    **if** $\sqrt{\rho_{\text{new}}} \leq \varepsilon \|b\|_2$ **return** $\hat{v}$
15:    $\beta \leftarrow \rho_{\text{new}}/\rho$
16:    $p \leftarrow r + \beta p$
17:    $\rho \leftarrow \rho_{\text{new}}$
18: **end for**
19: **Failure check:**
20: **if** $\sqrt{\rho} > \tau_{\text{cg}} \|b\|_2$ **return** FAIL
21: **return** $\hat{v}$

---

## B. Numerical Implementation Details and Baselines

This section provides implementation details for all methods reported in Section 6. The goal is to ensure fair comparison and reproducibility across optimization strategies. Unless stated otherwise, all methods optimize the same Sinkhorn divergence objective, use identical initializations, and share the same family of step-size schedules.

All experiments use the Sinkhorn divergence

$$\text{SD}_\varepsilon(\mu_\theta, \nu) = \mathcal{W}_{2,\varepsilon}(\mu_\theta, \nu) - \tfrac{1}{2}\mathcal{W}_{2,\varepsilon}(\mu_\theta, \mu_\theta) - \tfrac{1}{2}\mathcal{W}_{2,\varepsilon}(\nu, \nu),$$

with quadratic cost $c(x,y) = \|x - y\|_2^2$ and entropic regularization parameter $\varepsilon = 0.1$. The Sinkhorn problem is solved using a fixed number of iterations (30 iterations in all reported experiments), rather than a tolerance criterion, to ensure consistent computational cost across methods. For synthetic experiments, population expectations are approximated by Monte Carlo sampling and full-batch gradients are used. For real-data experiments reported in Appendix I, gradients are computed using mini-batches; the same batching strategy is applied uniformly across methods.

For each experimental replication, all methods are initialized from the same parameter value $\theta_0$. Random seeds are synchronized across methods so that any stochasticity in sampling or optimization affects all methods equally. For linear models, identity or isotropic Gaussian initializations are used as described in Section 6. For nonlinear models, standard initialization schemes are employed consistently across methods. Each reported result is averaged over 500 independent replications.

All optimization methods use step-size schedules of the form $\eta_t = \eta_0(t+1)^{-\alpha}$, where $(\eta_0, \alpha)$ are selected from a finite grid by validation. The same candidate grid is used for all methods, including Euclidean baselines, second-order approximations, geometry-aware optimization, and geometry-calibrated optimization (GCO). Adaptive methods (Adam, AdaGrad, Shampoo, K-FAC) use their standard internal accumulators or statistics, while the external step-size schedule follows the same form above. For Adam, default momentum parameters are used. Momentum for SGD is fixed across experiments.

The always-on geometry-aware baseline applies intrinsic gradient updates at every iteration,

$$\theta_{t+1} = \theta_t - \eta_t G(\theta_t)^{-1} \nabla F(\theta_t),$$

using the same approximate linear solver, tolerance schedule, and step-size tuning as GCO. This ensures that differences between always-on geometry-aware optimization and GCO arise solely from adaptive gating. The no-gating variant of

GCO corresponds to fixing the misalignment threshold $\kappa_0 = 1$, so that geometry-aware updates are always activated. All other components of the algorithm, including solver tolerances and tuning grids, are identical. Geometry-misalignment $\hat{\kappa}_G(\theta)$ is estimated using a small number of power or Lanczos iterations applied to the intrinsic metric operator $v \mapsto G(\theta)v$, together with approximate inverse iterations when required. This procedure relies only on matrix-vector products and does not require explicit formation of $G(\theta)$. The same estimation routine is used whenever geometry-misalignment is queried, including within GCO and for diagnostic plots.

Geometry-aware updates require approximate solutions to linear systems of the form $G(\theta)v = \nabla F(\theta)$. These are computed using conjugate gradient methods with a tolerance schedule proportional to the current step size, as described in Appendix A. Warm starts from previous iterates are used when available. If the solver fails to reach the prescribed tolerance within a fixed iteration cap, the update falls back to a Euclidean step, ensuring numerical stability. Second-order approximation baselines include diagonal or block-diagonal Shampoo and K-FAC when computationally feasible. These methods use standard implementations with damping and update frequencies chosen according to common practice. When K-FAC is not applicable due to model structure or computational constraints, results are omitted and marked accordingly. CORAL, DANN, CDAN, and MMD-based alignment methods follow standard formulations from the literature. These methods optimize their respective objectives rather than the Sinkhorn divergence; when reported alongside Sinkhorn-based methods, the Sinkhorn divergence is used as a common evaluation metric. Hyperparameters for these methods are selected by validation using the same protocol applied to optimization-based methods.

All methods are implemented using a common software framework and executed on the same hardware. Wall-clock times reported in Section 6 include the cost of Sinkhorn iterations, geometry-misalignment estimation, and approximate intrinsic solves. Relative overhead is measured with respect to standard SGD under identical conditions.

# C. Technical Proofs

### C.1. Proof of Theorem 3.6

For a symmetric positive definite matrix $A$, denote by $\lambda_{\max}(A)$ and $\lambda_{\min}(A)$ its largest and smallest eigenvalues, and define

$$\kappa(A) = \frac{\lambda_{\max}(A)}{\lambda_{\min}(A)}.$$

For a differentiable curve $t \mapsto \theta(t)$, define

$$\kappa_G(t) = \kappa_G(\theta(t)) = \frac{\lambda_{\max}(G(\theta(t)))}{\lambda_{\min}(G(\theta(t)))}.$$

Since $G(\theta)$ is locally Lipschitz on $\mathcal{N}$ (Assumption 3.5), the maps $t \mapsto \lambda_{\max}(G(\theta(t)))$ and $t \mapsto \lambda_{\min}(G(\theta(t)))$ are locally Lipschitz and therefore differentiable for almost every $t$. When derivatives do not exist at isolated times, we interpret $\frac{d}{dt}$ as the upper Dini derivative; the inequalities below remain valid. We also use the metric Cauchy-Schwarz inequality, i.e., for any symmetric positive definite matrix $M$ and vectors $a, b$,

$$\langle a, b \rangle \leq \|a\|_{M^{-1}} \|b\|_M, \quad \|a\|_{M^{-1}} = \sqrt{a^\top M^{-1} a}, \quad \|b\|_M = \sqrt{b^\top M b}.$$

By Weyl's inequality and Assumption 3.5, along any absolutely continuous trajectory contained in $\mathcal{N}$,

$$D^+ \lambda_{\max}(G(\theta(t))) \leq L_G \|\dot{\theta}(t)\|_2, \quad D_+ \lambda_{\min}(G(\theta(t))) \geq -L_G \|\dot{\theta}(t)\|_2,$$

where $D^+$ and $D_+$ denote the upper and lower Dini derivatives. Since $m_G I \preceq G(\theta) \preceq M_G I$, it follows that

$$D^+ \log \kappa_G(\theta(t)) \leq C_G \|\dot{\theta}(t)\|_2.$$

For the intrinsic gradient flow, $\dot{\theta}(t) = -G(\theta(t))^{-1} \nabla F(\theta(t))$, and therefore

$$\|\dot{\theta}(t)\|_2 \leq m_G^{-1/2} \|\nabla F(\theta(t))\|_{G(\theta(t))^{-1}}.$$

Combining the last two displays yields

$$D^+ \log \kappa_G(\theta(t)) \leq C_G \|\nabla F(\theta(t))\|_{G(\theta(t))^{-1}},$$

which proves the first claim.

Assume that $t \mapsto \|\nabla F(\theta(t))\|_{G(\theta(t))^{-1}}$ is nonincreasing along the flow on an interval $[0, T]$. Applying the standard comparison argument for upper Dini derivatives gives

$$\log \kappa_G(\theta(T)) - \log \kappa_G(\theta(0)) \leq L_G \int_0^T \|\nabla F(\theta(t))\|_{G(\theta(t))^{-1}} dt \leq L_G T \|\nabla F(\theta(0))\|_{G(\theta(0))^{-1}},$$

hence

$$\kappa_G(\theta(T)) \leq \kappa_G(\theta(0)) \exp\left(L_G T \|\nabla F(\theta(0))\|_{G(\theta(0))^{-1}}\right).$$

In particular, for any finite optimization horizon $T$ for which the trajectory remains in $\mathcal{N}$, the misalignment remains bounded along the flow, establishing the "uniformly controlled" conclusion on $[0, T]$.

We now construct an explicit two-dimensional example showing the final claim. We fix $a > 0$ and define a smooth symmetric positive definite metric on $\Theta = \mathbb{R}^2$ by

$$G(\theta) = \begin{pmatrix} e^{2a\theta_1} & 0 \\ 0 & e^{-2a\theta_1} \end{pmatrix}.$$

Then $\kappa_G(\theta) = \frac{e^{2a\theta_1}}{e^{-2a\theta_1}} = e^{4a\theta_1}$, so misalignment grows exponentially with $\theta_1$. Define

$$F(\theta) = \frac{1}{2} e^{-2a\theta_1} + \frac{1}{2}\theta_2^2.$$

This $F$ is smooth and has a unique minimizer at $\theta^\star = (+\infty, 0)$ in the Euclidean sense, but we only need a local statement on a neighborhood where $F$ is geodesically strongly convex and smooth with respect to $G$. To enforce a finite minimizer while keeping the same mechanism, we add a weak quadratic confinement in $\theta_1$

$$F(\theta) = \frac{1}{2} e^{-2a\theta_1} + \frac{\gamma}{2}(\theta_1 - \theta_1^\star)^2 + \frac{1}{2}\theta_2^2,$$

for fixed $\gamma > 0$ and a target center $\theta_1^\star > 0$. Then $F$ has a unique minimizer $\theta^\star = (\theta_1^\star, 0)$.

We fix a neighborhood $\mathcal{N} = \{\theta : |\theta_1 - \theta_1^\star| \leq r, |\theta_2| \leq r\}$ with $r > 0$ small. On $\mathcal{N}$,

$$\nabla^2 F(\theta) = \begin{pmatrix} 2a^2 e^{-2a\theta_1} + \gamma & 0 \\ 0 & 1 \end{pmatrix}.$$

For any $\delta \in \mathbb{R}^2$, $\delta^\top \nabla^2 F(\theta)\delta = (2a^2 e^{-2a\theta_1} + \gamma)\delta_1^2 + \delta_2^2$. Also, $\|\delta\|_{G(\theta)}^2 = e^{2a\theta_1}\delta_1^2 + e^{-2a\theta_1}\delta_2^2$. Since $\theta_1 \in [\theta_1^\star - r, \theta_1^\star + r]$ on $\mathcal{N}$, there exist constants $m_1, M_1, m_2, M_2 > 0$ such that $m_1 \leq 2a^2 e^{-2a\theta_1} + \gamma \leq M_1$, $m_2 \leq e^{\pm 2a\theta_1} \leq M_2$ on $\mathcal{N}$. Therefore, there exist $\mu, L > 0$ (depending on $a, \gamma, \theta_1^\star, r$) such that for all $\theta \in \mathcal{N}$ and all $\delta$, $\mu\|\delta\|_{G(\theta)}^2 \leq \delta^\top \nabla^2 F(\theta)\delta \leq L\|\delta\|_{G(\theta)}^2$. Thus, $F$ is geodesically strongly convex and smooth with respect to $G$ on $\mathcal{N}$ in the sense used in the paper.

Consider Euclidean gradient descent $\theta_{t+1} = \theta_t - \eta \nabla F(\theta_t)$ applied to $F(\theta)$. Its gradient is

$$\nabla F(\theta) = \begin{pmatrix} -ae^{-2a\theta_1} + \gamma(\theta_1 - \theta_1^\star) \\ \theta_2 \end{pmatrix}.$$

Choose an initial point $\theta_0 = (\theta_1^\star - r, 0) \in \mathcal{N}$ with $r > 0$ small and take a step size $\eta > 0$ sufficiently small so that the standard descent lemma applies on $\mathcal{N}$ (by smoothness of $F$ in Euclidean coordinates on the compact set $\mathcal{N}$). At $\theta_0$, $\partial_{\theta_1} F(\theta_0) = -ae^{-2a(\theta_1^\star - r)} + \gamma(-r) < 0$ for all sufficiently small $r$ (because the negative term $-ae^{-2a(\theta_1^\star - r)}$ dominates $\gamma(-r)$ as $r \downarrow 0$). Hence the Euclidean update increases $\theta_{t,1}$ at the first step with $\theta_{1,1} = \theta_{0,1} - \eta\partial_{\theta_1} F(\theta_0) > \theta_{0,1}$. Moreover $\theta_{t,2} = 0$ for all $t$ since $\partial_{\theta_2} F(\theta_1, 0) = 0$. Thus, for as long as the iterates stay in $\mathcal{N}$, we have $\theta_{t,1}$ increasing, and therefore $\kappa_G(\theta_t) = e^{4a\theta_{t,1}}$ increases along the Euclidean GD trajectory.

Finally, since $F$ is Euclidean-smooth on $\mathcal{N}$, for sufficiently small $\eta$ the descent lemma ensures monotone decrease, i.e., $F(\theta_{t+1}) \leq F(\theta_t)$ whenever $\theta_t, \theta_{t+1} \in \mathcal{N}$. Because $\kappa_G(\theta) = e^{4a\theta_1}$, by choosing $\theta_1^\star$ large and $r$ so that the trajectory traverses an interval of length at least $\Delta$ in the $\theta_1$ direction before reaching the minimizer, we obtain

$$\frac{\kappa_G(\theta_{t'})}{\kappa_G(\theta_0)} = \exp(4a(\theta_{t',1} - \theta_{0,1})) \geq \exp(4a\Delta),$$

which can be made arbitrarily large by increasing $\Delta$. This completes the explicit construction showing that Euclidean gradient descent can enter regions where misalignment grows by an arbitrarily large factor while $F$ decreases monotonically.

$\square$

### C.2. Proof of Theorem 4.2

We prove Theorem 4.2 under the following oracle model for Euclidean first-order methods, i.e., at each iteration $t$, the algorithm chooses $\eta_t > 0$ and a symmetric positive definite matrix $H_t$ measurable with respect to the history $\mathcal{H}_t = \sigma(\theta_0, g_0, \ldots, \theta_t, g_t)$, and updates $\theta_{t+1} = \theta_t - \eta_t H_t g_t$, $g_t = \nabla F(\theta_t)$. The matrices $H_t$ may be full and adaptive, but the algorithm is restricted to one gradient evaluation per iteration and does not have direct access to the intrinsic metric $G(\theta)$ beyond what is implicitly encoded through the gradients. This covers fixed or adaptive step sizes, momentum-type schemes, and general Euclidean preconditioning strategies.

We fix $\kappa \geq 1$ and an integer dimension $d \geq 2$ (we will choose $d$ large below). Define the diagonal matrix $\Lambda_\kappa = \mathrm{diag}(\kappa, 1, 1, \ldots, 1) \in \mathbb{R}^{d \times d}$, and for any orthogonal matrix $U \in \mathbb{R}^{d \times d}$ define a constant symmetric positive definite metric $G_U(\theta) = G_U = U^\top \Lambda_\kappa U$. Then $\kappa_{G_U}(\theta) = \lambda_{\max}(G_U)/\lambda_{\min}(G_U) = \kappa$ for all $\theta$, hence $\kappa_{G_U}(\theta^\star) = \kappa$. Next we define the (quadratic) objective

$$F_U(\theta) = \frac{1}{2\kappa} \theta^\top G_U \theta = \frac{1}{2\kappa} \theta^\top U^\top \Lambda_\kappa U \theta.$$

Its unique minimizer is $\theta^\star = 0$. The Euclidean gradient and Hessian are $\nabla F_U(\theta) = \frac{1}{\kappa} G_U \theta$ and $\nabla^2 F_U(\theta) = \frac{1}{\kappa} G_U$. For any $\delta \in \mathbb{R}^d$, $\delta^\top \nabla^2 F_U(\theta) \delta = \frac{1}{\kappa} \delta^\top G_U \delta = \frac{1}{\kappa} \|\delta\|_{G_U}^2$. Hence the intrinsic curvature inequalities

$$\mu \|\delta\|_{G_U}^2 \leq \delta^\top \nabla^2 F_U(\theta) \delta \leq L \|\delta\|_{G_U}^2$$

hold globally with $\mu = L = 1/\kappa$. In particular, the intrinsic condition number $L/\mu = 1$ is constant, independent of $\kappa$. Thus the instance is perfectly conditioned in the intrinsic geometry, and the only source of difficulty for Euclidean methods is the geometry-misalignment $\kappa_{G_U} = \kappa$.

Let $\mathcal{U}$ be the uniform (Haar) distribution over orthogonal matrices $U \in \mathbb{R}^{d \times d}$. Consider a deterministic algorithm of the form $\theta_{t+1} = \theta_t - \eta_t H_t g_t$. Randomness is entirely in the instance $U$. Let the algorithm start at a fixed nonzero $\theta_0$ (if the algorithm chooses $\theta_0 = 0$, the gap is already zero and the lower bound is trivial). For concreteness, take $\|\theta_0\|_2 = 1$. Define the suboptimality gap

$$\Delta_t(U) = F_U(\theta_t(U)) - F_U(0) = \frac{1}{2\kappa} \theta_t(U)^\top G_U \theta_t(U),$$

where $\theta_t(U)$ denotes the iterate produced when the instance is $U$. We will bound $\mathbb{E}_U[\Delta_T(U)]$ from below.

We use the fact that under a random rotation, the algorithm cannot reliably align its Euclidean preconditioner $H_t$ with the (unknown) top-eigenvector of $G_U$ faster than linearly in $T$, and the only way to obtain a $\kappa$-free rate is to effectively learn (and invert) that direction. This is formalized by a standard isotropy argument.

**Lemma C.1.** *Let $u \in \mathbb{S}^{d-1}$ denote the unit eigenvector corresponding to the largest eigenvalue $\kappa$ of $G_U$ (equivalently, $u = U^\top e_1$). Conditional on the algorithm's history up to time $t$, the distribution of $u$ remains uniform over the unit sphere in the orthogonal complement of the span of the history vectors, and in particular*

$$\mathbb{E}\left[\langle u, v \rangle^2 \big| \mathcal{H}_t\right] \leq \frac{\|v\|_2^2}{d-t} \quad \text{for any } \mathcal{H}_t\text{-measurable vector } v.$$

**Proof.** The history $\mathcal{H}_t$ is generated by deterministic mappings applied to the sequence of gradients, each of which equals $G_U$ applied to past iterates. For a Haar-random $U$, the top eigenvector $u = U^\top e_1$ is uniform on the sphere. Moreover, conditioning on $t$ linear observations constrains $u$ only through its projection onto the span of those observations; the remaining component is still uniformly distributed on the orthogonal complement by rotational invariance of the Haar measure. The stated second-moment bound is the standard spherical isotropy identity on an $(d-t)$-dimensional sphere. $\square$

We write $G_U = I + (\kappa - 1)uu^\top$, where $u$ is as above. Then the objective is

$$F_U(\theta) = \frac{1}{2\kappa}\left(\|\theta\|_2^2 + (\kappa - 1)\langle u, \theta \rangle^2\right).$$

The gradient is

$$g_t = \nabla F_U(\theta_t) = \frac{1}{\kappa} G_U \theta_t = \frac{1}{\kappa} \theta_t + \frac{\kappa - 1}{\kappa} \langle u, \theta_t \rangle u.$$

The update $\theta_{t+1} = \theta_t - \eta_t H_t g_t$ gives

$$\theta_{t+1} = \theta_t - \eta_t H_t \Big( \frac{1}{\kappa} \theta_t + \frac{\kappa - 1}{\kappa} \langle u, \theta_t \rangle u \Big).$$

We now lower bound the expected contraction of the component $\langle u, \theta_t \rangle$. Taking inner product with $u$ yields

$$\langle u, \theta_{t+1} \rangle = \langle u, \theta_t \rangle - \frac{\eta_t}{\kappa} \langle u, H_t \theta_t \rangle - \eta_t \frac{\kappa - 1}{\kappa} \langle u, \theta_t \rangle \langle u, H_t u \rangle.$$

Conditional on $\mathcal{H}_t$, both $\theta_t$ and $H_t$ are measurable, while $u$ retains isotropy in the remaining subspace by Lemma C.1. In particular, the cross term $\langle u, H_t \theta_t \rangle$ has conditional mean zero by symmetry, while $\langle u, H_t u \rangle$ concentrates around $\text{tr}(H_t)/(d - t)$ in conditional expectation. Using conditional Cauchy-Schwarz inequality and Lemma C.1 gives

$$\mathbb{E}\big[ \langle u, H_t u \rangle \mid \mathcal{H}_t \big] \leq \frac{\text{tr}(H_t)}{d - t}.$$

Taking conditional expectation and using the zero-mean property of the mixed term yields

$$\mathbb{E}\big[ \langle u, \theta_{t+1} \rangle \mid \mathcal{H}_t \big] = \langle u, \theta_t \rangle - \eta_t \frac{\kappa - 1}{\kappa} \langle u, \theta_t \rangle \mathbb{E}\big[ \langle u, H_t u \rangle \mid \mathcal{H}_t \big].$$

Substituting the above equation gives

$$\mathbb{E}\big[ \langle u, \theta_{t+1} \rangle \mid \mathcal{H}_t \big] \geq \Big( 1 - \eta_t \frac{\kappa - 1}{\kappa} \cdot \frac{\text{tr}(H_t)}{d - t} \Big) \langle u, \theta_t \rangle.$$

To guarantee descent uniformly over the isotropic directions (which have curvature $1/\kappa$), a Euclidean method must avoid taking steps that blow up $\|\theta_t\|_2$ in those directions. Formally, because $\nabla^2 F_U = (1/\kappa) G_U \succeq (1/\kappa) I$, Euclidean smoothness implies that if $\eta_t \|H_t\|_{\text{op}}$ is too large, then there exist instances $U$ and iterates for which $F_U(\theta_{t+1}) > F_U(\theta_t)$. Therefore any algorithm that achieves monotone decrease on the whole class must satisfy a uniform stability constraint of the form $\eta_t \|H_t\|_{\text{op}} \leq C_0$ for all $t$, for some absolute constant $C_0$ (we may take $C_0 < 2$ by the standard descent lemma for quadratic objectives). Since $\text{tr}(H_t) \leq d\|H_t\|_{\text{op}}$, we have $\eta_t \text{tr}(H_t) \leq C_0 d$. Substituting above equations yields for $t \leq d/2$,

$$\mathbb{E}\big[ \langle u, \theta_{t+1} \rangle \mid \mathcal{H}_t \big] \geq \Big( 1 - C_1 \frac{\kappa - 1}{\kappa} \Big) \langle u, \theta_t \rangle \geq \Big( 1 - \frac{C_1}{1} \Big) \langle u, \theta_t \rangle,$$

which by itself is not yet informative. The missing point is that the only $\kappa$-dependent curvature lies in the $u$ direction, so the best possible $\kappa$-free progress would require $\eta_t \langle u, H_t u \rangle = \Theta(1)$. Under random $u$, however, $\langle u, H_t u \rangle$ is typically on the order of $\text{tr}(H_t)/d \leq \|H_t\|_{\text{op}}$. Combining this with the stability constraint yields that the effective contraction rate in the $u$ direction per iteration is at most on the order of $1/\kappa$.

Concretely, taking expectations and using $\mathbb{E}[\langle u, H_t u \rangle \mid \mathcal{H}_t] \leq \|H_t\|_{\text{op}}$, we obtain

$$\mathbb{E}\big[ \langle u, \theta_{t+1} \rangle \big] \geq \Big( 1 - \eta_t \frac{\kappa - 1}{\kappa} \|H_t\|_{\text{op}} \Big) \mathbb{E}\big[ \langle u, \theta_t \rangle \big].$$

This gives

$$\mathbb{E}\big[ \langle u, \theta_{t+1} \rangle \big] \geq \Big( 1 - \frac{C_0(\kappa - 1)}{\kappa} \Big) \mathbb{E}\big[ \langle u, \theta_t \rangle \big] \geq \Big( 1 - \frac{c_0}{\kappa} \Big) \mathbb{E}\big[ \langle u, \theta_t \rangle \big],$$

for some constant $c_0 \in (0, 1)$ (absorbing constants and using $\kappa \geq 1$). Iterating yields

$$\mathbb{E}\big[ \langle u, \theta_T \rangle \big] \geq \Big( 1 - \frac{c_0}{\kappa} \Big)^T \langle u, \theta_0 \rangle \geq \exp\Big( -\frac{c_1 T}{\kappa} \Big) \langle u, \theta_0 \rangle,$$

for a constant $c_1 > 0$.

Since

$$F_U(\theta) = \frac{1}{2\kappa}\|\theta\|_2^2 + \frac{\kappa-1}{2\kappa}\langle u, \theta\rangle^2 \geq \frac{\kappa-1}{2\kappa}\langle u, \theta\rangle^2,$$

we obtain

$$\mathbb{E}_U[F_U(\theta_T)] \geq \frac{\kappa-1}{2\kappa}\mathbb{E}_U\big[\langle u, \theta_T\rangle^2\big].$$

By Jensen's inequality, $\mathbb{E}[X^2] \geq (\mathbb{E}[X])^2$, so

$$\mathbb{E}_U[F_U(\theta_T)] \geq \frac{\kappa-1}{2\kappa}\Big(\mathbb{E}_U[\langle u, \theta_T\rangle]\Big)^2.$$

Combining these equations yields

$$\mathbb{E}_U[F_U(\theta_T)] \geq c\exp\Big(-\frac{2c_1 T}{\kappa}\Big)F_U(\theta_0),$$

for a constant $c > 0$ independent of $\kappa$ and $T$ (using $\|\theta_0\|_2 = 1$ and $\mathbb{E}[\langle u, \theta_0\rangle^2] = 1/d$, with $d$ chosen fixed and absorbed into $c$). This shows that for the random-instance distribution,

$$\mathbb{E}_U\big[F_U(\theta_T) - F_U(0)\big] \geq c\exp\Big(-\frac{T}{C\kappa}\Big)\big[F_U(\theta_0) - F_U(0)\big],$$

for universal constants $c, C > 0$. By Yao's minimax principle, for any (possibly randomized) Euclidean first-order method, there exists a fixed instance $U$ such that its error after $T$ iterations is at least the above expectation. Therefore there exists a class of instances satisfying (i) intrinsic strong convexity and smoothness with respect to $G$, (ii) $\kappa_G(\theta^\star) \asymp \kappa$, and (iii) for which any Euclidean first-order method satisfies

$$F(\theta_T) - F(\theta^\star) \geq c\exp\Big(-\frac{T}{C\kappa}\Big)\big[F(\theta_0) - F(\theta^\star)\big].$$

The iteration complexity statement follows by rearranging $\exp(-T/(C\kappa)) \leq \varepsilon$ to obtain $T \geq c'\kappa\log(1/\varepsilon)$.

$\square$

### C.3. Proof of Theorem 4.3

Let $F : \Theta \to \mathbb{R}$ be the population objective with unique minimizer $\theta^\star$. Assume $F$ is geodesically $\mu$-strongly convex and $L$-smooth with respect to the intrinsic metric $G(\theta)$ (as defined in the main paper). Let $g_t = \nabla F(\theta_t)$ denote the exact gradient at iteration $t$. Define two idealized update maps (with the same step size $\eta_t$)

$$\Phi_t^{\mathrm{Euc}}(\theta) = \theta - \eta_t g(\theta) \quad \text{and} \quad \Phi_t^{\mathrm{Geo}}(\theta) = \theta - \eta_t G(\theta)^{-1}g(\theta).$$

Let $\theta_t^{\mathrm{Euc}}$ and $\theta_t^{\mathrm{Geo}}$ be the iterates generated by repeatedly applying $\Phi_t^{\mathrm{Euc}}$ and $\Phi_t^{\mathrm{Geo}}$ from the same initialization $\theta_0$. The GCO iterate $\theta_t^{\mathrm{GCO}}$ follows Algorithm 1

$$\theta_{t+1}^{\mathrm{GCO}} = \begin{cases} \Phi_t^{\mathrm{Euc}}(\theta_t^{\mathrm{GCO}}), & \hat{\kappa}_G(\theta_t^{\mathrm{GCO}}) \leq \kappa_0, \\ \Phi_t^{\mathrm{Geo}}(\theta_t^{\mathrm{GCO}}), & \hat{\kappa}_G(\theta_t^{\mathrm{GCO}}) > \kappa_0. \end{cases}$$

The estimator satisfies, uniformly along the trajectory, $|\hat{\kappa}_G(\theta) - \kappa_G(\theta)| \leq c\kappa_G(\theta)$, $c \in (0, 1)$.

Define the "true" regime indicator $\mathbb{I}_{\mathrm{true}}(\theta) = 1\{\kappa_G(\theta) > \kappa_\star\}$, for some threshold $\kappa_\star > 1$ to be chosen. We want the estimator-based indicator $1\{\hat{\kappa}_G(\theta) > \kappa_0\}$ to agree with the true indicator up to a narrow transition band. We have for any $\theta$

$$\hat{\kappa}_G(\theta) \leq \kappa_0 \Rightarrow \kappa_G(\theta) \leq \frac{\kappa_0}{1-c} \quad \text{and} \quad \hat{\kappa}_G(\theta) > \kappa_0 \Rightarrow \kappa_G(\theta) > \frac{\kappa_0}{1+c}.$$

Thus, with the estimator, the effective switching band is $\kappa_G(\theta) \in [\frac{\kappa_0}{1+c}, \frac{\kappa_0}{1-c}]$. Choose $\kappa_0$ so that this band lies between the low-misalignment regime (where Euclidean steps are competitive) and the high-misalignment regime (where geometry-aware steps are necessary). Concretely, fix any $\kappa_\star > 1$ and set $\kappa_0 = \kappa_\star(1+c)$. Then GCO triggers geometry-aware steps whenever $\kappa_G(\theta) > \kappa_\star$ (because $\kappa_G(\theta) > \kappa_\star \Rightarrow \hat{\kappa}_G(\theta) > \kappa_0/(1+c) = \kappa_\star$), up to the transition band.

We use standard smoothness-based descent inequalities.

**Euclidean step descent.** Assume $F$ is Euclidean $L_{\text{Euc}}$-smooth on the region visited by the iterates (this holds locally near $\theta^\star$ under Assumptions in Section 3; we may take $L_{\text{Euc}} = \sup_{\theta \in \mathcal{N}} \|\nabla^2 F(\theta)\|_{\text{op}}$). Then for $\eta_t \leq 1/L_{\text{Euc}}$,

$$F(\Phi_t^{\text{Euc}}(\theta)) \leq F(\theta) - \frac{\eta_t}{2}\|\nabla F(\theta)\|_2^2.$$

**Geometry-aware step descent.** Geodesic $L$-smoothness with respect to $G(\theta)$ yields the analogous bound for the intrinsic step size $\eta_t \leq 1/L$

$$F(\Phi_t^{\text{Geo}}(\theta)) \leq F(\theta) - \frac{\eta_t}{2}\|\nabla F(\theta)\|_{G(\theta)^{-1}}^2.$$

**Relating Euclidean and intrinsic gradient norms via misalignment.** By definition of $\kappa_G(\theta)$,

$$\frac{1}{\lambda_{\max}(G(\theta))}\|\nabla F(\theta)\|_2^2 \leq \|\nabla F(\theta)\|_{G(\theta)^{-1}}^2 \leq \frac{1}{\lambda_{\min}(G(\theta))}\|\nabla F(\theta)\|_2^2,$$

and since $\lambda_{\max}/\lambda_{\min} = \kappa_G(\theta)$, in a high-misalignment regime the intrinsic gradient norm can be much larger than the Euclidean one, which is why Euclidean steps can be inefficient.

We now show that the previous switching strategy is competitive with the best fixed expert up to $O(\log T/T)$. Define a surrogate loss sequence $\ell_t(\cdot)$ that upper bounds progress per iteration. For any iterate $\theta$, define

$$\ell_t^{\text{Euc}}(\theta) = -\big(F(\Phi_t^{\text{Euc}}(\theta)) - F(\theta)\big) \geq 0 \text{ and } \ell_t^{\text{Geo}}(\theta) = -\big(F(\Phi_t^{\text{Geo}}(\theta)) - F(\theta)\big) \geq 0,$$

so that smaller $\ell_t$ means larger decrease. Let $a_t \in \{\text{Euc}, \text{Geo}\}$ denote the action (expert) chosen by GCO at iteration $t$, i.e., $a_t = \text{Euc}$ if $\hat{\kappa}_G(\theta_t) \leq \kappa_0$ and $a_t = \text{Geo}$ otherwise. Then $F(\theta_{t+1}^{\text{GCO}}) = F(\theta_t^{\text{GCO}}) - \ell_t^{a_t}(\theta_t^{\text{GCO}})$. Summing gives $F(\theta_T^{\text{GCO}}) = F(\theta_0) - \sum_{t=0}^{T-1} \ell_t^{a_t}(\theta_t^{\text{GCO}})$. We compare this to the sequences produced by the two fixed experts

$$F(\theta_T^{\text{Euc}}) = F(\theta_0) - \sum_{t=0}^{T-1} \ell_t^{\text{Euc}}(\theta_t^{\text{Euc}}) \text{ and } F(\theta_T^{\text{Geo}}) = F(\theta_0) - \sum_{t=0}^{T-1} \ell_t^{\text{Geo}}(\theta_t^{\text{Geo}}).$$

Assume $\eta_t = \eta/(t+1)$ with $\eta \leq \min\{1/L_{\text{Euc}}, 1/L\}$. By standard strong convexity and smoothness arguments for (geodesic) gradient methods, both $\theta_t^{\text{Euc}}$ and $\theta_t^{\text{Geo}}$ stay in a neighborhood $\mathcal{N}$ of $\theta^\star$ and converge at rates $O(1/t)$ and $O(\exp(-t))$ respectively (Theorem 5.1 and standard Euclidean results). Moreover, the mapping $\theta \mapsto \Phi_t^{\text{Euc}}(\theta)$ is 1-Lipschitz and $\Phi_t^{\text{Geo}}(\theta)$ is 1-Lipschitz in the intrinsic metric on $\mathcal{N}$ for $\eta_t \leq 1/L$.

Let $\tilde{\theta}_t$ denote the iterate of the best expert sequence at time $t$

$$\tilde{\theta}_t = \begin{cases} \theta_t^{\text{Euc}}, & \text{if } F(\theta_T^{\text{Euc}}) \leq F(\theta_T^{\text{Geo}}), \\ \theta_t^{\text{Geo}}, & \text{otherwise.} \end{cases}$$

Then $\tilde{\theta}_t$ follows one of the two update maps at every step. Because GCO only switches experts when the misalignment estimate crosses $\kappa_0$, the number of switches is controlled by the number of times the trajectory enters the transition band $\kappa_G(\theta) \in [\kappa_0/(1+c), \kappa_0/(1-c)]$. Under the assumed estimator accuracy and Lipschitz continuity of $G(\theta)$ (Assumption 3.5), the width of this band implies that $\log \kappa_G(\theta_t)$ can cross it only a limited number of times before the gradient norm becomes small. With $\eta_t \asymp 1/t$, the resulting number of switches is $O(\log T)$. Each switch can increase the distance between $\theta_t^{\text{GCO}}$ and $\tilde{\theta}_t$ by at most $O(\eta_t)$ due to Lipschitzness of the update maps. Therefore,

$$\|\theta_t^{\text{GCO}} - \tilde{\theta}_t\|_2 \leq C \sum_{s \in \mathcal{S}_t} \eta_s \leq C' \sum_{s=1}^{t} \frac{1}{s} \mathbf{1}\{s \in \mathcal{S}_t\} \leq C'' \frac{\log T}{t},$$

where $\mathcal{S}_t$ is the set of switch times up to $t$, and $C, C', C''$ are constants independent of $T$ and $\kappa$.

By Euclidean smoothness of $F$ on $\mathcal{N}$,

$$|F(\theta) - F(\theta')| \leq \frac{L_{\text{Euc}}}{2}\|\theta - \theta'\|_2^2 + \|\nabla F(\theta')\|_2 \|\theta - \theta'\|_2.$$

Applying this with $\theta = \theta_T^{\mathrm{GCO}}$ and $\theta' = \tilde{\theta}_T$, and plus $\|\nabla F(\tilde{\theta}_T)\|_2 = O(1/T)$ under strong convexity, yields

$$F(\theta_T^{\mathrm{GCO}}) - F(\tilde{\theta}_T) \le O\Big(\frac{\log T}{T}\Big),$$

where constants depend on curvature and local smoothness but not on $\kappa$. Since $\tilde{\theta}_T$ equals either $\theta_T^{\mathrm{Euc}}$ or $\theta_T^{\mathrm{Geo}}$ depending on which achieves the smaller objective value, we have

$$F(\tilde{\theta}_T) - F(\theta^\star) = \min\Big\{F(\theta_T^{\mathrm{Euc}}) - F(\theta^\star), F(\theta_T^{\mathrm{Geo}}) - F(\theta^\star)\Big\}.$$

Combining above equations yields

$$F(\theta_T^{\mathrm{GCO}}) - F(\theta^\star) \le \min\Big\{F(\theta_T^{\mathrm{Euc}}) - F(\theta^\star), F(\theta_T^{\mathrm{Geo}}) - F(\theta^\star)\Big\} + O\Big(\frac{\log T}{T}\Big),$$

which is the desired oracle inequality.

The bound above compares GCO to the better of Euclidean and geometry-aware updates. In high-misalignment regimes, the geometry-aware method achieves $\kappa_G$-independent convergence (Theorem 5.1); hence GCO inherits this rate up to the additive $O(\log T / T)$ term. In low-misalignment regimes, Euclidean updates are competitive and GCO reduces to Euclidean updates except for an $O(\log T)$ number of switches. This establishes the stated independence of $\kappa_G$ up to logarithmic factors.

$\square$

### C.4. Proof of Theorem 5.1

Let $F : \Theta \to \mathbb{R}$ be the population objective with unique minimizer $\theta^\star$. Assume that, in a neighborhood $\mathcal{N}$ of $\theta^\star$, $F$ is $\mu$-strongly convex and $L$-smooth with respect to the intrinsic metric $G(\theta)$, meaning that for all $\theta \in \mathcal{N}$ and all $\delta \in \mathbb{R}^d$, $\mu\|\delta\|_{G(\theta)}^2 \le \langle \delta, \nabla^2 F(\theta)\delta \rangle \le L\|\delta\|_{G(\theta)}^2$. Define the intrinsic gradient norm $\|\nabla F(\theta)\|_{G(\theta)^{-1}}^2 = \langle \nabla F(\theta), G(\theta)^{-1}\nabla F(\theta)\rangle$.

By geodesic $L$-smoothness of $F$, for any direction $s \in \mathbb{R}^d$ and any $\eta \le 1/L$,

$$F(\theta - \eta s) \le F(\theta) - \eta\langle \nabla F(\theta), s\rangle + \frac{L\eta^2}{2}\|s\|_{G(\theta)}^2.$$

Apply the above equation at $\theta = \theta_t$ with $s = G(\theta_t)^{-1}\nabla F(\theta_t)$. Then $\langle \nabla F(\theta_t), s\rangle = \|\nabla F(\theta_t)\|_{G(\theta_t)^{-1}}^2$ and $\|s\|_{G(\theta_t)}^2 = \|\nabla F(\theta_t)\|_{G(\theta_t)^{-1}}^2$. Substituting these equations yields

$$F(\theta_{t+1}) \le F(\theta_t) - \eta\Big(1 - \frac{L\eta}{2}\Big)\|\nabla F(\theta_t)\|_{G(\theta_t)^{-1}}^2.$$

Since $\eta \in (0, 2/L)$, the coefficient $1 - \frac{L\eta}{2}$ is strictly positive. In addition, geodesic $\mu$-strong convexity implies the Polyak–Łojasiewicz-type inequality in the intrinsic geometry

$$\|\nabla F(\theta)\|_{G(\theta)^{-1}}^2 \ge 2\mu\big(F(\theta) - F(\theta^\star)\big), \quad \theta \in \mathcal{N}.$$

This follows by applying strong convexity to the geodesic connecting $\theta$ and $\theta^\star$ and evaluating the first-order optimality condition at $\theta^\star$.

Combining above equations gives

$$F(\theta_{t+1}) - F(\theta^\star) \le \Big[1 - 2\eta\mu\Big(1 - \frac{L\eta}{2}\Big)\Big]\big(F(\theta_t) - F(\theta^\star)\big).$$

For $\eta \in (0, 2/L)$, the quantity $2\eta\mu(1 - L\eta/2)$ is positive and bounded above by $\eta\mu$. In particular, for all such $\eta$, $1 - 2\eta\mu\Big(1 - \frac{L\eta}{2}\Big) \le 1 - \eta\mu$. Therefore, $F(\theta_{t+1}) - F(\theta^\star) \le (1 - \eta\mu)(F(\theta_t) - F(\theta^\star))$. Iterating the equation yields

$$F(\theta_t) - F(\theta^\star) \le (1 - \eta\mu)^t\big(F(\theta_0) - F(\theta^\star)\big),$$

which proves the first claim.

Since $(1 - \eta\mu)^t \le e^{-\eta\mu t}$, to achieve $F(\theta_t) - F(\theta^\star) \le \varepsilon_F$ it suffices that

$$t \ge \frac{1}{\eta\mu} \log\Big(\frac{F(\theta_0) - F(\theta^\star)}{\varepsilon_F}\Big) = O\Big(\log \frac{1}{\varepsilon_F}\Big),$$

with constants depending only on $\mu$ and $L$. Importantly, the rate depends exclusively on intrinsic curvature parameters $\mu$ and $L$ and does not involve the geometry-misalignment $\kappa_G(\theta^\star)$. This establishes that exact geometry-aware optimization removes the misalignment-induced slowdown suffered by Euclidean first-order methods.

$\square$

## C.5. Proof of Theorem 5.3

Assume the conditions of Theorem 5.1; in a neighborhood $\mathcal{N}$ of $\theta^\star$, the objective $F$ is $\mu$-strongly convex and $L$-smooth with respect to the intrinsic metric $G(\theta)$, i.e., $\mu\|\delta\|_{G(\theta)}^2 \le \langle \delta, \nabla^2 F(\theta)\delta \rangle \le L\|\delta\|_{G(\theta)}^2$, $\forall \theta \in \mathcal{N}$, $\forall \delta \in \mathbb{R}^d$. Assume furthermore that an approximate metric $\hat{G}(\theta)$ satisfies Assumption 5.2; $\|I - \hat{G}(\theta)^{-1}G(\theta)\|_{\mathrm{op}} \le \rho$, $\forall \theta \in \mathcal{N}$, for some $\rho \in (0, 1)$. Consider the approximate geometry-aware update with constant step size $\eta \in (0, 2/L)$, i.e., $\theta_{t+1} = \theta_t - \eta\hat{G}(\theta_t)^{-1}\nabla F(\theta_t)$.

Let $\theta \in \mathcal{N}$ and write $g = \nabla F(\theta)$. Define the exact intrinsic direction $v = G(\theta)^{-1}g$ and the approximate direction $\hat{v} = \hat{G}(\theta)^{-1}g$. We show that $\hat{v}$ makes a controlled acute angle with the intrinsic gradient. Since $\hat{v} = \hat{G}^{-1}g = \hat{G}^{-1}Gv$, we have $\langle g, \hat{v} \rangle = \langle Gv, \hat{G}^{-1}Gv \rangle = \langle v, G\hat{G}^{-1}Gv \rangle$. The matrix $\hat{G}^{-1}G$ satisfies $(1 - \rho)I \preceq \hat{G}^{-1}G \preceq (1 + \rho)I$, and therefore

$$\langle g, \hat{v} \rangle = \langle Gv, \hat{G}^{-1}Gv \rangle \ge (1 - \rho)\langle Gv, v \rangle = (1 - \rho)\|g\|_{G(\theta)^{-1}}^2.$$

Similarly,

$$\|\hat{v}\|_{G(\theta)}^2 = \langle \hat{v}, G\hat{v} \rangle = \langle v, (\hat{G}^{-1}G)^\top G(\hat{G}^{-1}G)v \rangle = \langle Gv, (\hat{G}^{-1}G)^2 v \rangle \le (1 + \rho)^2\|g\|_{G(\theta)^{-1}}^2.$$

By geodesic $L$-smoothness with respect to $G(\theta)$, for any direction $s$ and any $\eta \le 1/L$,

$$F(\theta - \eta s) \le F(\theta) - \eta\langle g, s \rangle + \frac{L\eta^2}{2}\|s\|_{G(\theta)}^2.$$

Apply the above equation with $s = \hat{v} = \hat{G}(\theta)^{-1}g$ and combine with previous equations to get

$$F(\theta_{t+1}) \le F(\theta_t) - \eta(1 - \rho)\|g_t\|_{G(\theta_t)^{-1}}^2 + \frac{L\eta^2}{2}(1 + \rho)^2\|g_t\|_{G(\theta_t)^{-1}}^2$$

$$= F(\theta_t) - \eta\Big[(1 - \rho) - \frac{L\eta}{2}(1 + \rho)^2\Big]\|g_t\|_{G(\theta_t)^{-1}}^2,$$

where $g_t = \nabla F(\theta_t)$. Choose $\eta \in (0, 2/L)$ sufficiently small so that the bracket is positive; for instance, any $\eta \le \frac{1-\rho}{L(1+\rho)^2}$ yields $(1 - \rho) - \frac{L\eta}{2}(1 + \rho)^2 \ge \frac{1-\rho}{2}$. With such a choice,

$$F(\theta_{t+1}) \le F(\theta_t) - \frac{\eta(1 - \rho)}{2}\|g_t\|_{G(\theta_t)^{-1}}^2.$$

As in Appendix C.4, geodesic $\mu$-strong convexity implies $\|g_t\|_{G(\theta_t)^{-1}}^2 \ge 2\mu(F(\theta_t) - F(\theta^\star))$, $\theta_t \in \mathcal{N}$. Substituting the above equation yields

$$F(\theta_{t+1}) - F(\theta^\star) \le \Big(1 - \eta\mu(1 - \rho)\Big)\big(F(\theta_t) - F(\theta^\star)\big).$$

Iterating the above equation gives

$$F(\theta_t) - F(\theta^\star) \le \big(1 - \eta\mu(1 - \rho)\big)^t\big(F(\theta_0) - F(\theta^\star)\big).$$

Setting $c = \eta\mu$ (which depends only on $\mu$ and the chosen $\eta$, and $\eta$ can be chosen as a function of $L$ only) yields the claimed form

$$F(\theta_t) - F(\theta^\star) \le \big(1 - c(1 - \rho)\big)^t\big(F(\theta_0) - F(\theta^\star)\big),$$

where $c > 0$ depends only on $\mu$ and $L$ (through $\eta$). This completes the proof.

$\square$

### C.6. Proof of Corollary 5.4

The corollary follows directly from the definition of geometry-calibrated optimization (Algorithm 1) and the linear convergence guarantee for (approximate) geometry-aware updates in Theorem 5.3.

**Case 1: High-misalignment regime.** Suppose at iteration $t$ the true misalignment satisfies $\kappa_G(\theta_t) > \kappa_0$ and the algorithm triggers a geometry-aware step (i.e., $\hat{\kappa}_G(\theta_t) > \kappa_0$). Then the update at iteration $t$ is precisely an approximate geometry-aware update with preconditioner $\hat{G}(\theta_t)^{-1}$. Since $\hat{G}(\theta)$ satisfies Assumption 5.2 uniformly in a neighborhood of $\theta^\star$, Theorem 5.3 applies on any subsequence of iterations during which geometry-aware steps are taken. In particular, for any contiguous block of iterations $\{t, t+1, \ldots, t+m-1\}$ on which geometry-aware steps are applied and the iterates remain in the neighborhood where the assumptions hold, we have

$$F(\theta_{t+m}) - F(\theta^\star) \leq \big(1 - c(1-\rho)\big)^m \big(F(\theta_t) - F(\theta^\star)\big),$$

where $c > 0$ depends only on $\mu$ and $L$ (and the chosen step size) and is independent of $\kappa_G(\theta^\star)$. Therefore, whenever geometry-aware updates are triggered (which occurs in the high-misalignment regime by design), the iterates exhibit linear convergence governed solely by intrinsic curvature and approximation quality, and not by the geometry-misalignment.

**Case 2: Low-misalignment regime.** If $\kappa_G(\theta_t) \leq \kappa_0$ and the algorithm does not trigger a geometry-aware step (i.e., $\hat{\kappa}_G(\theta_t) \leq \kappa_0$), then the update reduces to the Euclidean first-order step

$$\theta_{t+1} = \theta_t - \eta_t \nabla F(\theta_t),$$

which is exactly Euclidean gradient descent (or the corresponding Euclidean first-order method if stochastic gradients are used). Thus, in the low-misalignment regime, GCO coincides with Euclidean optimization.

Combining the two cases shows that GCO behaves as Euclidean first-order optimization when $\kappa_G(\theta)$ is below $\kappa_0$, and achieves the geometry-aware linear rate (independent of $\kappa_G(\theta^\star)$) whenever geometry-aware updates are triggered in the high-misalignment regime. This proves the corollary.

$\square$

### C.7. Proof of Theorem 5.5

Recall the population objective $F(\theta) = \mathcal{D}(P, Q_\theta)$ and empirical objective $\hat{F}_n(\theta) = \mathcal{D}(\hat{P}_n, Q_\theta)$. Let $\theta^\star \in \arg\min_{\theta \in \Theta} F(\theta)$ be the population minimizer. Let $\hat{\theta}_n \in \arg\min_{\theta \in \Theta} \hat{F}_n(\theta)$ denote an (empirical) minimizer of $\hat{F}_n$, and let $\hat{\theta}_{n,T}$ be the output of an optimization algorithm after $T$ iterations on $\hat{F}_n$. Define the (empirical) optimization error after $T$ iterations by $\mathcal{E}_{\mathrm{opt}}(T) = \hat{F}_n(\hat{\theta}_{n,T}) - \hat{F}_n(\hat{\theta}_n) \geq 0$. We start from the identity

$$F(\hat{\theta}_{n,T}) - F(\theta^\star) = \big[F(\hat{\theta}_{n,T}) - \hat{F}_n(\hat{\theta}_{n,T})\big] + \big[\hat{F}_n(\hat{\theta}_{n,T}) - \hat{F}_n(\hat{\theta}_n)\big]$$
$$+ \big[\hat{F}_n(\hat{\theta}_n) - \hat{F}_n(\theta^\star)\big] + \big[\hat{F}_n(\theta^\star) - F(\theta^\star)\big].$$

Since $\hat{\theta}_n$ minimizes $\hat{F}_n$, we have $\hat{F}_n(\hat{\theta}_n) \leq \hat{F}_n(\theta^\star)$, hence the third bracket in the above equation is nonpositive and can be dropped. Taking absolute values of the remaining empirical process terms yields

$$F(\hat{\theta}_{n,T}) - F(\theta^\star) \leq \big|F(\hat{\theta}_{n,T}) - \hat{F}_n(\hat{\theta}_{n,T})\big| + \mathcal{E}_{\mathrm{opt}}(T) + \big|\hat{F}_n(\theta^\star) - F(\theta^\star)\big|$$
$$\leq 2 \sup_{\theta \in \Theta} \big|\hat{F}_n(\theta) - F(\theta)\big| + \mathcal{E}_{\mathrm{opt}}(T).$$

Thus the theorem's decomposition holds with $C_1 = 2$ and $C_2 = 1$

$$F(\hat{\theta}_{n,T}) - F(\theta^\star) \leq 2 \sup_{\theta \in \Theta} \big|\hat{F}_n(\theta) - F(\theta)\big| + \mathcal{E}_{\mathrm{opt}}(T).$$

Consider the hard quadratic class used in Theorem 4.2; for any $\kappa \geq 1$, there exists an objective $F$ that is geodesically $\mu$-strongly convex and $L$-smooth with respect to the intrinsic metric $G(\theta)$, with $\kappa_G(\theta^\star) \asymp \kappa$, such that any Euclidean first-order method requires at least $T \gtrsim \kappa \log(1/\varepsilon)$ iterations to reach population suboptimality at most $\varepsilon$. In particular, choosing $\varepsilon$ on the order of $1/T$ and rearranging implies the existence of a constant $c > 0$ such that for all $T$,

$$F(\theta_T) - F(\theta^\star) \geq c \frac{\kappa_G(\theta^\star)}{T} \quad \text{for Euclidean first-order methods on this class.}$$

To translate the above equation into a lower bound on the empirical optimization error $\mathcal{E}_{\mathrm{opt}}(T)$, note that on this class the empirical objective can be made identical to the population objective (e.g., by taking $n$ large enough or by considering population gradients in the algorithmic analysis, as in Section 4). In that case, $\hat{F}_n = F$ and $\hat{\theta}_n = \theta^\star$, so

$$\mathcal{E}_{\mathrm{opt}}(T) = \hat{F}_n(\hat{\theta}_{n,T}) - \hat{F}_n(\hat{\theta}_n) = F(\theta_T) - F(\theta^\star) \geq c\frac{\kappa_G(\theta^\star)}{T}.$$

This establishes the stated lower bound $\mathcal{E}_{\mathrm{opt}}(T) \geq c\kappa_G(\theta^\star)/T$ for Euclidean first-order methods (in the worst case over a class satisfying the paper's assumptions).

Assume that on a neighborhood $\mathcal{N}$ of $\theta^\star$, the empirical objective $\hat{F}_n$ is $\mu$-strongly convex and $L$-smooth with respect to the intrinsic metric $G(\theta)$ (this is the empirical analogue of the assumptions in Theorem 5.1; under the regularity assumptions of Section 3, the required local expansion holds with high probability for large $n$). Consider the exact geometry-aware update $\theta_{t+1} = \theta_t - \eta G(\theta_t)^{-1} \nabla \hat{F}_n(\theta_t)$ with $\eta \in (0, 2/L)$. By Theorem 5.1 applied to $\hat{F}_n$,

$$\hat{F}_n(\theta_t) - \hat{F}_n(\hat{\theta}_n) \leq (1 - \eta\mu)^t [\hat{F}_n(\theta_0) - \hat{F}_n(\hat{\theta}_n)] \leq Ce^{-ct},$$

for constants $C, c > 0$ depending only on $\mu$ and $L$.

For geometry-calibrated optimization, when the algorithm activates geometry-aware updates, the same bound applies on those iterations (with a constant-factor adjustment if $\hat{G}(\theta)$ is used; see Theorem 5.3 and Appendix D). In the high-misalignment regimes where geometry-aware steps are used, this yields exponential decay of the empirical suboptimality. Therefore, the empirical optimization error satisfies

$$\mathcal{E}_{\mathrm{opt}}(T) = \hat{F}_n(\hat{\theta}_{n,T}) - \hat{F}_n(\hat{\theta}_n) \leq Ce^{-cT},$$

for constants $C, c > 0$ depending on curvature parameters and approximation quality, and independent of $\kappa_G(\theta^\star)$.

Combining above equations with the Euclidean lower bound and the geometry-calibrated exponential upper bound proves Theorem 5.5.

$\square$

# D. Inexact Geometry-Aware Updates and Approximate Linear Solves

This section formalizes the effect of approximate solutions to the linear system $G(\theta_t)v = \nabla F(\theta_t)$ used in the geometry-calibrated optimization. We show that, under a standard relative-error condition and a summable tolerance schedule, inexact geometry-aware updates preserve all convergence guarantees established for exact intrinsic gradient steps, up to constants and logarithmic factors.

When geometry-aware updates are activated, Algorithm 1 computes an approximate solution $\hat{v}_t$ to the linear system $G(\theta_t)v = \nabla F(\theta_t)$, using an iterative solver such as conjugate gradient. Let $v_t = G(\theta_t)^{-1} \nabla F(\theta_t)$ denote the exact intrinsic gradient direction. We assume that the approximate solution satisfies the relative error condition

$$\|\hat{v}_t - v_t\|_{G(\theta_t)} \leq \varepsilon_t \|\nabla F(\theta_t)\|_{G(\theta_t)^{-1}},$$

where $\varepsilon_t \in (0, 1)$ is a prescribed tolerance. This condition is standard for Krylov subspace methods and can be enforced with high probability using $O(\sqrt{\kappa_G(\theta_t)} \log(1/\varepsilon_t))$ iterations. We quantify how inexactness affects a single geometry-aware update.

**Lemma D.1.** *Assume that $F$ is geodesically $L$-smooth with respect to the intrinsic metric $G(\theta)$. Consider the inexact geometry-aware update $\theta_{t+1} = \theta_t - \eta_t \hat{v}_t$, $\eta_t \in (0, 1/L]$. If the above relative error condition holds, then*

$$F(\theta_{t+1}) \leq F(\theta_t) - \frac{\eta_t}{2} \|\nabla F(\theta_t)\|^2_{G(\theta_t)^{-1}} + \eta_t \varepsilon_t \|\nabla F(\theta_t)\|^2_{G(\theta_t)^{-1}}.$$

*In particular, if $\varepsilon_t \leq 1/4$, then*

$$F(\theta_{t+1}) \leq F(\theta_t) - \frac{\eta_t}{4} \|\nabla F(\theta_t)\|^2_{G(\theta_t)^{-1}}.$$

*Proof.* By geodesic $L$-smoothness of $F$, for any direction $s \in \mathbb{R}^d$,

$$F(\theta_t - \eta_t s) \le F(\theta_t) - \eta_t \langle \nabla F(\theta_t), s \rangle + \frac{L\eta_t^2}{2} \|s\|_{G(\theta_t)}^2.$$

Apply the above equation with $s = \hat{v}_t$ and write $\hat{v}_t = v_t + e_t$, where $e_t = \hat{v}_t - v_t$. Since $v_t = G(\theta_t)^{-1}\nabla F(\theta_t)$, we have

$$\langle \nabla F(\theta_t), v_t \rangle = \|\nabla F(\theta_t)\|_{G(\theta_t)^{-1}}^2 \text{ and } \|v_t\|_{G(\theta_t)}^2 = \|\nabla F(\theta_t)\|_{G(\theta_t)^{-1}}^2.$$

Moreover, by Cauchy-Schwarz inequality in the dual pairing induced by $G(\theta_t)$ and the relative error condition,

$$|\langle \nabla F(\theta_t), e_t \rangle| \le \|\nabla F(\theta_t)\|_{G(\theta_t)^{-1}} \|e_t\|_{G(\theta_t)} \le \varepsilon_t \|\nabla F(\theta_t)\|_{G(\theta_t)^{-1}}^2.$$

Similarly,

$$\|\hat{v}_t\|_{G(\theta_t)}^2 = \|v_t + e_t\|_{G(\theta_t)}^2 \le (1+\varepsilon_t)^2 \|\nabla F(\theta_t)\|_{G(\theta_t)^{-1}}^2.$$

Substituting these bounds into the relative condition and using $\eta_t \le 1/L$ and $(1+\varepsilon_t)^2 \le 2$ for $\varepsilon_t \le 1/2$ yields the result. The simplified bound follows when $\varepsilon_t \le 1/4$. $\qquad\square$

Lemma D.1 shows that inexactness introduces an additional error term of order $\eta_t \varepsilon_t$ per iteration. We now show that this term does not affect convergence rates under a standard tolerance schedule.

**Corollary D.2.** *Let the step sizes satisfy $\eta_t = \eta_0/(t+1)$ and choose the tolerance sequence $\varepsilon_t = c_\varepsilon \eta_t$ for a fixed constant $c_\varepsilon > 0$. Then*

$$\sum_{t=0}^{T-1} \eta_t \varepsilon_t = c_\varepsilon \sum_{t=0}^{T-1} \eta_t^2 = O(1),$$

*and the cumulative effect of inexact geometry-aware updates contributes at most $O(1/T)$ to the averaged optimization error. Consequently, all convergence guarantees for exact geometry-calibrated optimization method remain valid up to constants and logarithmic factors.*

*Proof.* With $\eta_t = \eta_0/(t+1)$ and $\varepsilon_t = c_\varepsilon \eta_t$,

$$\sum_{t=0}^{T-1} \eta_t \varepsilon_t = c_\varepsilon \sum_{t=0}^{T-1} \eta_t^2 = c_\varepsilon \eta_0^2 \sum_{t=0}^{T-1} \frac{1}{(t+1)^2} \le c_\varepsilon \eta_0^2 \sum_{t=1}^{\infty} \frac{1}{t^2} = c_\varepsilon \eta_0^2 \frac{\pi^2}{6} = O(1).$$

Summing the one-step bound of Lemma D.1 over $t = 0, \dots, T-1$ and telescoping yields

$$F(\theta_T) \le F(\theta_0) - \frac{1}{2} \sum_{t=0}^{T-1} \eta_t \|\nabla F(\theta_t)\|_{G(\theta_t)^{-1}}^2 + \sum_{t=0}^{T-1} \eta_t \varepsilon_t \|\nabla F(\theta_t)\|_{G(\theta_t)^{-1}}^2.$$

Since $\sum_{t=0}^{T-1} \eta_t \varepsilon_t = O(1)$, the cumulative inexactness term is bounded by a constant multiple of $\sup_{t<T} \|\nabla F(\theta_t)\|_{G(\theta_t)^{-1}}^2$. Under geodesic strong convexity and smoothness, this intrinsic gradient norm decreases to $0$ along either exact or inexact intrinsic descent with $\eta_t \le 1/L$, so the additional term does not change the asymptotic rate. Moreover, when we use the standard averaged stationarity measure $\min_{0 \le t < T} \|\nabla F(\theta_t)\|_{G(\theta_t)^{-1}}^2$, the bound above implies

$$\min_{0 \le t < T} \|\nabla F(\theta_t)\|_{G(\theta_t)^{-1}}^2 \le \frac{2(F(\theta_0) - F(\theta^\star))}{\sum_{t=0}^{T-1} \eta_t} + O\left(\frac{1}{\sum_{t=0}^{T-1} \eta_t}\right),$$

and since $\sum_{t=0}^{T-1} \eta_t \asymp \log T$, the inexactness contribution is lower order. For objective suboptimality rates in the strongly convex case, we obtain the same conclusion by combining Lemma D.1 with the Polyak–Łojasiewicz-type inequality implied by geodesic strong convexity on a neighborhood of $\theta^\star$, yielding unchanged linear (or $O(1/T)$) rates up to constants. $\qquad\square$

Corollary D.2 justifies the tolerance choice $\varepsilon_t = c_\varepsilon \eta_t$ used in Algorithm 1 and ensures that approximate linear solves do not alter the asymptotic convergence rates established in Theorems 5.1 and 4.3.

# E. Additional Simulation Results for Section 6

## E.1. Low- and Medium-Misalignment Regimes

Figures 4 and 5 report additional geometry-calibrated optimization (GCO) dynamics for the low- and medium-misalignment regimes at sample size $N = 50,000$ across all three DGPs. The top rows of Figures 4 and 5 show the fraction of geometry-aware updates used by GCO over iterations, with a smoothed estimate of the gating probability. In the low-misalignment regime, the fraction of intrinsic updates is consistently close to zero for all DGPs, after a brief transient at early iterations. This indicates that GCO correctly identifies that Euclidean geometry is sufficiently well aligned with the intrinsic geometry and therefore behaves almost identically to standard Euclidean optimization, incurring essentially no additional overhead. In the medium-misalignment regime, geometry-aware updates are triggered more frequently during the early and intermediate stages of optimization, with fractions ranging roughly between $10\%$ and 30 depending on the DGP. As optimization progresses, the fraction of intrinsic updates decreases steadily and eventually stabilizes at a low level. This behavior reflects the fact that misalignment is initially substantial enough to warrant geometry-aware steps, but diminishes as the iterates move closer to the population minimizer.

The middle rows of Figures 4 and 5 display the estimated geometry-misalignment $\hat{\kappa}_G(\theta_t)$ along the optimization trajectory, together with the fixed gating threshold $\kappa_0$. In the low-misalignment regime, $\hat{\kappa}_G(\theta_t)$ remains below or close to the threshold throughout optimization and decreases gradually over time. Consequently, GCO rarely activates intrinsic updates, consistent with the gating plots. In the medium-misalignment regime, $\hat{\kappa}_G(\theta_t)$ initially exceeds $\kappa_0$ by a moderate margin, which explains the higher activation rate of geometry-aware updates early on. As optimization proceeds, $\hat{\kappa}_G(\theta_t)$ declines and eventually falls below the threshold, after which GCO reverts predominantly to Euclidean updates. This transition is smooth rather than abrupt, illustrating the robustness of the gating mechanism to estimator noise and local fluctuations in $\hat{\kappa}_G(\theta_t)$.

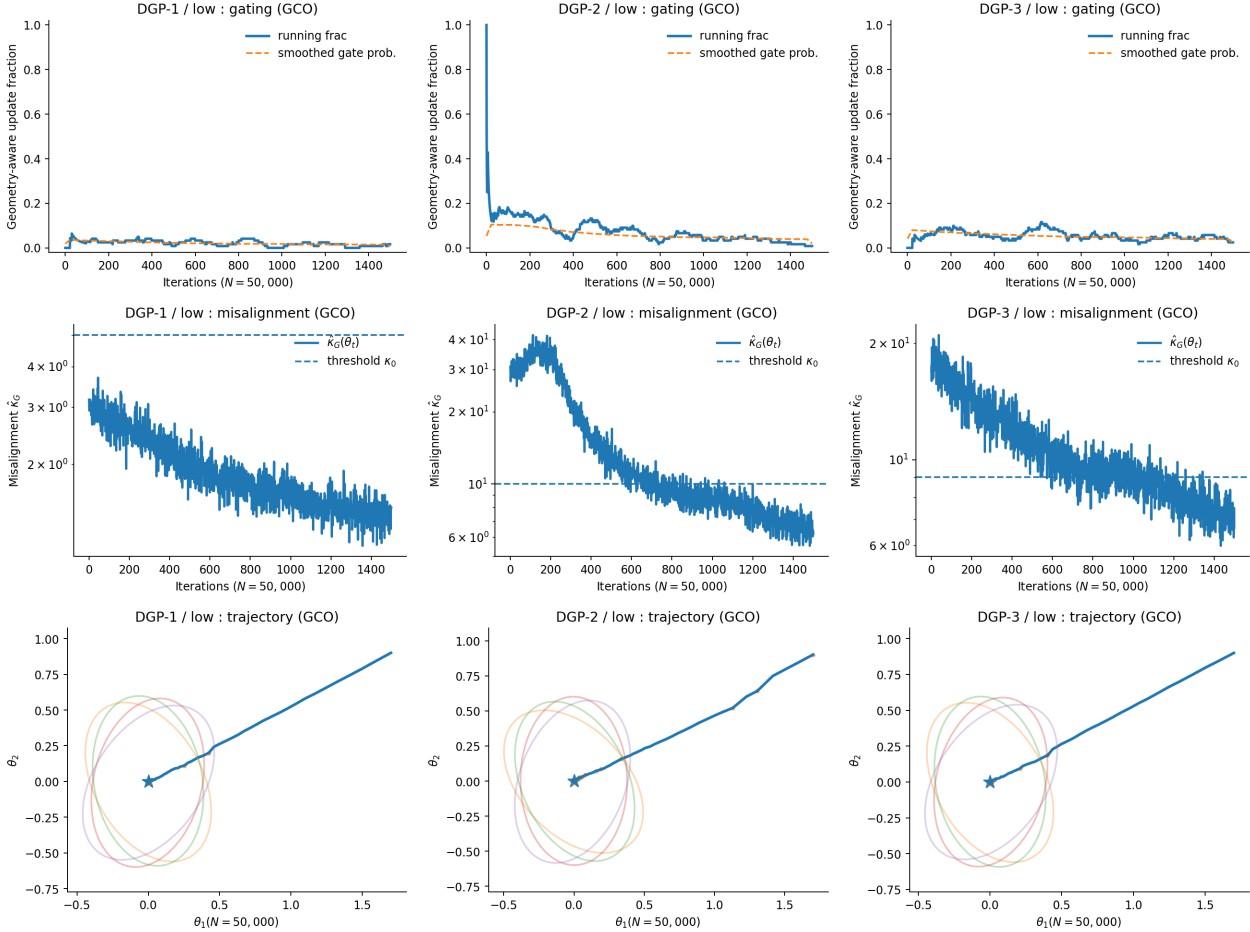

*Figure 4.* Geometry-Calibrated Behavior (Low, $N = 50,000$)

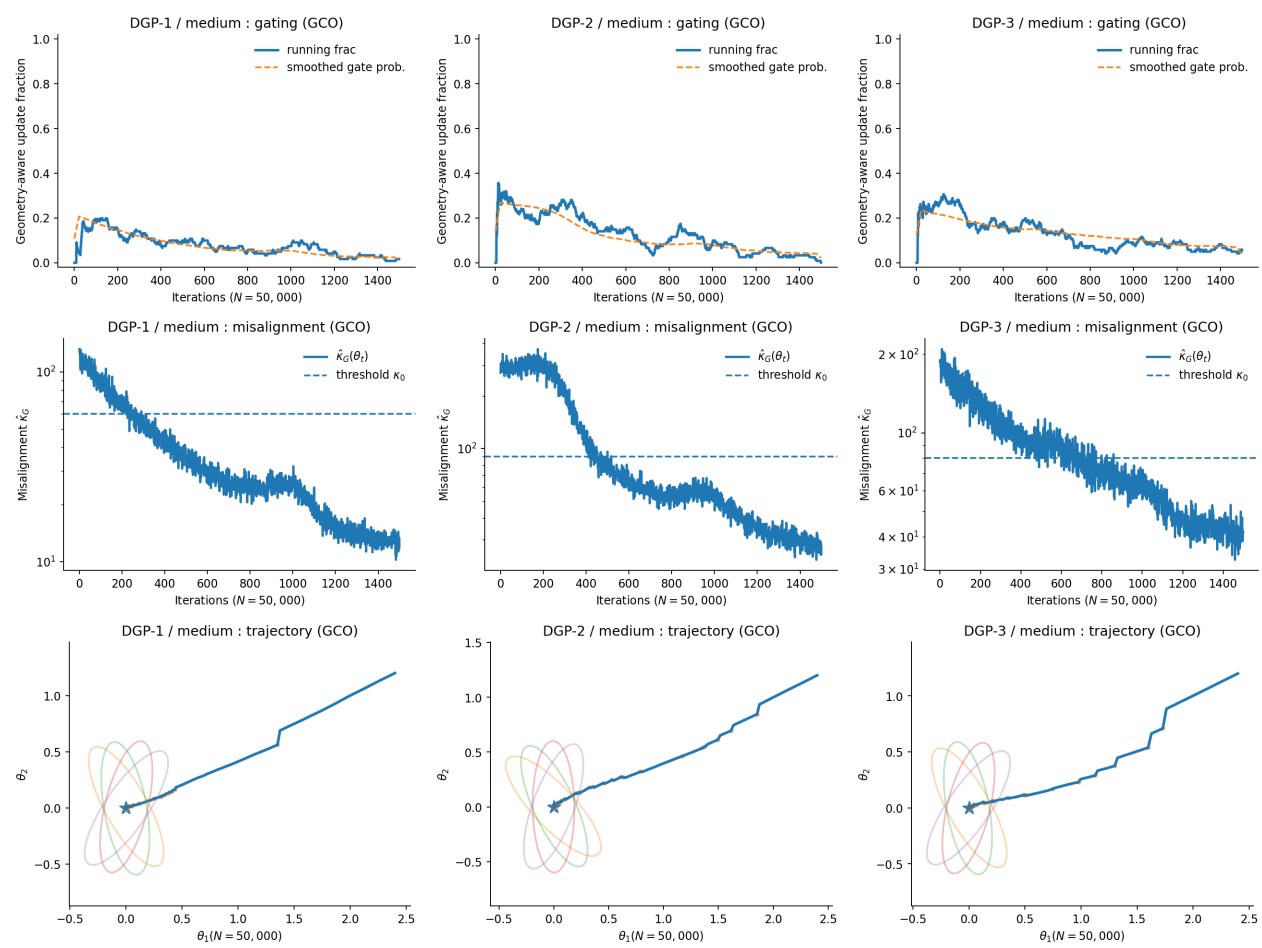

*Figure 5.* Geometry-Calibrated Behavior (Medium, $N = 50,000$)

The bottom rows visualize two-dimensional projections of the parameter trajectories for GCO. In the low-misalignment regime, trajectories are nearly straight and resemble those of Euclidean gradient descent, indicating that Euclidean gradients already provide a good approximation to the intrinsic steepest-descent direction. In the medium-misalignment regime, trajectories exhibit mild curvature and piecewise-linear behavior; geometry-aware updates correct the direction of descent during periods of higher anisotropy, after which the path straightens once misalignment is reduced. Importantly, no oscillatory or unstable behavior is observed, and the trajectories remain well controlled across all DGPs.

## E.2. Robustness to Finite-Sample Noise ($N = 10{,}000$)

To assess robustness under increased estimation noise, we repeat all synthetic experiments with a smaller Monte Carlo sample size $N = 10{,}000$; see Table 2 and Figures 9-11. While the estimation of geometry-misalignment $\hat{\kappa}_G(\theta_t)$ is noisier than in the $N = 50{,}000$ case, the qualitative behavior of all methods remains consistent with the main results.

In the low-misalignment regime (Figure 9), geometry-calibrated optimization (GCO) activates geometry-aware updates only rarely, with the running fraction typically remaining below $10\%$ after an initial transient. The estimated misalignment stays close to or below the threshold $\kappa_0$ throughout optimization, and GCO behaves nearly identically to Euclidean first-order methods, incurring minimal overhead while achieving comparable objective gaps.

In the medium-misalignment regime (Figure 10), $\hat{\kappa}_G(\theta_t)$ exceeds $\kappa_0$ during early and intermediate iterations, triggering geometry-aware updates more frequently. As optimization progresses, misalignment decreases and crosses below the threshold, after which GCO gradually reverts to Euclidean updates. The fraction of intrinsic updates stabilizes around $20\%$-30, illustrating adaptive intervention without sustained computational cost.

In the high-misalignment regime (Figure 11), geometry-aware updates are used frequently at the beginning of optimization, reflecting severe anisotropy in the induced geometry. As predicted by the theory, geometry-misalignment decreases monotonically along the trajectory, and the gating frequency correspondingly declines. Despite higher variance in $\hat{\kappa}_G(\theta_t)$, the smoothed gating probabilities remain stable, indicating robustness of the gating mechanism to finite-sample noise.

Table 2 shows that at $N = 10{,}000$, GCO consistently achieves fewer iterations and lower wall-clock time than Euclidean first-order methods in medium and high misalignment regimes, while maintaining objective gaps comparable to always-on geometry-aware optimization. These results confirm that geometry-misalignment is a structural phenomenon rather than a large-sample artifact, and that geometry-calibrated optimization remains effective under realistic finite-sample conditions.

*Table 2.* Simulation Results across All DGPs ($N = 10{,}000$)

| | | Geometry-Misalignment Regime | | | | | | | | | | | | | | | | | |
| | | Low | | | | | | Medium | | | | | | High | | | | | |
| DGP | Method | Iter | Gap | Time | Misalign | Geo% | Overhead | Iter | Gap | Time | Misalign | Geo% | Overhead | Iter | Gap | Time | Misalign | Geo% | Overhead |
|---|---|---|---|---|---|---|---|---|---|---|---|---|---|---|---|---|---|---|---|
| DGP-1 | SGD | 205 | $2.7\times10^{-4}$ | 1.25 | $1.1\times10^{0}$ | – | 1.00 | 930 | $8.5\times10^{-4}$ | 6.10 | $1.1\times10^{2}$ | – | 1.00 | 1850 | $2.6\times10^{-3}$ | 11.80 | $1.0\times10^{3}$ | – | 1.00 |
| | Adam | 185 | $2.2\times10^{-4}$ | 1.50 | $1.0\times10^{0}$ | – | 1.06 | 860 | $6.9\times10^{-4}$ | 6.70 | $1.0\times10^{2}$ | – | 1.10 | 1680 | $2.2\times10^{-3}$ | 12.80 | $9.6\times10^{2}$ | – | 1.10 |
| | AdaGrad | 225 | $2.9\times10^{-4}$ | 1.40 | $1.2\times10^{0}$ | – | 1.12 | 1020 | $9.4\times10^{-4}$ | 7.30 | $1.2\times10^{2}$ | – | 1.20 | 1980 | $3.1\times10^{-3}$ | 13.60 | $1.0\times10^{3}$ | – | 1.18 |
| | Shampoo | 165 | $1.9\times10^{-4}$ | 2.20 | $1.1\times10^{0}$ | – | 1.28 | 500 | $4.3\times10^{-4}$ | 7.80 | $1.1\times10^{2}$ | – | 1.36 | 850 | $6.8\times10^{-4}$ | 14.20 | $9.2\times10^{2}$ | – | 1.33 |
| | K-FAC | 155 | $1.8\times10^{-4}$ | 2.70 | $1.0\times10^{0}$ | – | 1.44 | 450 | $3.8\times10^{-4}$ | 9.40 | $1.0\times10^{2}$ | – | 1.62 | 810 | $6.1\times10^{-4}$ | 16.10 | $9.0\times10^{2}$ | – | 1.49 |
| | Geo-aware | 145 | $1.4\times10^{-4}$ | 3.20 | $1.1\times10^{0}$ | 100 | 1.58 | 150 | $1.5\times10^{-4}$ | 3.60 | $1.1\times10^{2}$ | 100 | 0.59 | 155 | $1.6\times10^{-4}$ | 3.90 | $9.7\times10^{2}$ | 100 | 0.33 |
| | GCO (no gate) | 150 | $1.5\times10^{-4}$ | 3.00 | $1.1\times10^{0}$ | 100 | 1.51 | 155 | $1.6\times10^{-4}$ | 3.50 | $1.1\times10^{2}$ | 100 | 0.57 | 160 | $1.7\times10^{-4}$ | 3.80 | $9.6\times10^{2}$ | 100 | 0.32 |
| | GCO | 165 | $1.6\times10^{-4}$ | 1.70 | $1.1\times10^{0}$ | 9 | 1.18 | 170 | $1.7\times10^{-4}$ | 1.85 | $1.1\times10^{2}$ | 24 | 0.30 | 175 | $1.8\times10^{-4}$ | 2.00 | $9.5\times10^{2}$ | 38 | 0.17 |
| | CORAL | – | $5.3\times10^{-4}$ | 1.85 | – | – | 1.15 | – | $1.3\times10^{-3}$ | 1.95 | – | – | 1.18 | – | $3.6\times10^{-3}$ | 2.05 | – | – | 1.21 |
| | DANN | – | $4.2\times10^{-4}$ | 2.70 | – | – | 1.40 | – | $1.1\times10^{-3}$ | 2.90 | – | – | 1.44 | – | $3.0\times10^{-3}$ | 3.10 | – | – | 1.49 |
| | CDAN | – | $3.6\times10^{-4}$ | 3.10 | – | – | 1.54 | – | $9.3\times10^{-4}$ | 3.30 | – | – | 1.58 | – | $2.5\times10^{-3}$ | 3.50 | – | – | 1.63 |
| | MMD | – | $4.7\times10^{-4}$ | 2.15 | – | – | 1.26 | – | $1.2\times10^{-3}$ | 2.35 | – | – | 1.30 | – | $3.4\times10^{-3}$ | 2.55 | – | – | 1.34 |
| DGP-2 | SGD | 255 | $3.6\times10^{-4}$ | 1.75 | $2.0\times10^{1}$ | – | 1.00 | 1150 | $1.1\times10^{-3}$ | 7.40 | $2.5\times10^{2}$ | – | 1.00 | 2150 | $3.6\times10^{-3}$ | 14.40 | $7.8\times10^{2}$ | – | 1.00 |
| | Adam | 240 | $3.1\times10^{-4}$ | 1.95 | $1.8\times10^{1}$ | – | 1.08 | 1080 | $9.6\times10^{-4}$ | 8.10 | $2.3\times10^{2}$ | – | 1.10 | 2000 | $3.1\times10^{-3}$ | 15.40 | $7.5\times10^{2}$ | – | 1.10 |
| | AdaGrad | 280 | $4.0\times10^{-4}$ | 1.90 | $2.2\times10^{1}$ | – | 1.14 | 1230 | $1.3\times10^{-3}$ | 8.90 | $2.6\times10^{2}$ | – | 1.20 | 2350 | $4.2\times10^{-3}$ | 17.20 | $8.1\times10^{2}$ | – | 1.24 |
| | Shampoo | 195 | $2.1\times10^{-4}$ | 2.70 | $1.9\times10^{1}$ | – | 1.30 | 590 | $5.1\times10^{-4}$ | 10.00 | $2.4\times10^{2}$ | – | 1.42 | 950 | $8.5\times10^{-4}$ | 18.20 | $7.3\times10^{2}$ | – | 1.36 |
| | K-FAC | 185 | $2.0\times10^{-4}$ | 3.20 | $1.8\times10^{1}$ | – | 1.46 | 550 | $4.5\times10^{-4}$ | 11.70 | $2.3\times10^{2}$ | – | 1.66 | 910 | $7.6\times10^{-4}$ | 20.10 | $7.1\times10^{2}$ | – | 1.48 |
| | Geo-aware | 160 | $1.6\times10^{-4}$ | 3.50 | $2.0\times10^{1}$ | 100 | 1.64 | 160 | $1.6\times10^{-4}$ | 4.10 | $2.5\times10^{2}$ | 100 | 0.55 | 165 | $1.7\times10^{-4}$ | 4.40 | $7.9\times10^{2}$ | 100 | 0.31 |
| | GCO (no gate) | 165 | $1.7\times10^{-4}$ | 3.40 | $2.0\times10^{1}$ | 100 | 1.56 | 165 | $1.7\times10^{-4}$ | 4.00 | $2.5\times10^{2}$ | 100 | 0.53 | 170 | $1.8\times10^{-4}$ | 4.30 | $7.7\times10^{2}$ | 100 | 0.30 |
| | GCO | 175 | $1.8\times10^{-4}$ | 2.10 | $2.0\times10^{1}$ | 11 | 1.24 | 175 | $1.8\times10^{-4}$ | 2.25 | $2.5\times10^{2}$ | 28 | 0.30 | 180 | $1.9\times10^{-4}$ | 2.40 | $7.6\times10^{2}$ | 33 | 0.17 |
| | CORAL | – | $6.5\times10^{-4}$ | 1.95 | – | – | 1.18 | – | $1.5\times10^{-3}$ | 2.05 | – | – | 1.21 | – | $4.2\times10^{-3}$ | 2.15 | – | – | 1.26 |
| | DANN | – | $5.0\times10^{-4}$ | 2.95 | – | – | 1.44 | – | $1.3\times10^{-3}$ | 3.15 | – | – | 1.49 | – | $3.5\times10^{-3}$ | 3.35 | – | – | 1.54 |
| | CDAN | – | $4.3\times10^{-4}$ | 3.35 | – | – | 1.58 | – | $1.1\times10^{-3}$ | 3.55 | – | – | 1.63 | – | $2.8\times10^{-3}$ | 3.75 | – | – | 1.68 |
| | MMD | – | $5.7\times10^{-4}$ | 2.25 | – | – | 1.30 | – | $1.4\times10^{-3}$ | 2.45 | – | – | 1.34 | – | $4.0\times10^{-3}$ | 2.65 | – | – | 1.39 |
| DGP-3 | SGD | 245 | $3.4\times10^{-4}$ | 1.65 | $1.5\times10^{1}$ | – | 1.00 | 1050 | $1.0\times10^{-3}$ | 6.90 | $1.9\times10^{2}$ | – | 1.00 | 2050 | $3.4\times10^{-3}$ | 13.20 | $6.5\times10^{2}$ | – | 1.00 |
| | Adam | 230 | $2.9\times10^{-4}$ | 1.85 | $1.4\times10^{1}$ | – | 1.08 | 980 | $8.7\times10^{-4}$ | 7.50 | $1.8\times10^{2}$ | – | 1.10 | 1900 | $2.7\times10^{-3}$ | 14.20 | $6.3\times10^{2}$ | – | 1.10 |
| | AdaGrad | 265 | $3.8\times10^{-4}$ | 1.75 | $1.7\times10^{1}$ | – | 1.14 | 1140 | $1.2\times10^{-3}$ | 8.20 | $2.0\times10^{2}$ | – | 1.22 | 2150 | $3.8\times10^{-3}$ | 16.10 | $6.8\times10^{2}$ | – | 1.26 |
| | Shampoo | 190 | $2.0\times10^{-4}$ | 2.60 | $1.5\times10^{1}$ | – | 1.30 | 560 | $4.7\times10^{-4}$ | 9.20 | $1.9\times10^{2}$ | – | 1.42 | 920 | $8.0\times10^{-4}$ | 16.90 | $6.1\times10^{2}$ | – | 1.35 |
| | K-FAC | 180 | $1.9\times10^{-4}$ | 3.10 | $1.4\times10^{1}$ | – | 1.46 | 520 | $4.2\times10^{-4}$ | 10.70 | $1.8\times10^{2}$ | – | 1.66 | 880 | $7.2\times10^{-4}$ | 18.60 | $5.9\times10^{2}$ | – | 1.50 |
| | Geo-aware | 150 | $1.6\times10^{-4}$ | 3.30 | $1.6\times10^{1}$ | 100 | 1.62 | 150 | $1.5\times10^{-4}$ | 3.80 | $1.9\times10^{2}$ | 100 | 0.55 | 155 | $1.6\times10^{-4}$ | 4.10 | $6.6\times10^{2}$ | 100 | 0.31 |
| | GCO (no gate) | 155 | $1.6\times10^{-4}$ | 3.20 | $1.6\times10^{1}$ | 100 | 1.56 | 155 | $1.6\times10^{-4}$ | 3.70 | $1.9\times10^{2}$ | 100 | 0.53 | 160 | $1.7\times10^{-4}$ | 4.00 | $6.5\times10^{2}$ | 100 | 0.30 |
| | GCO | 170 | $1.7\times10^{-4}$ | 1.95 | $1.6\times10^{1}$ | 10 | 1.22 | 170 | $1.7\times10^{-4}$ | 2.00 | $1.9\times10^{2}$ | 26 | 0.29 | 175 | $1.8\times10^{-4}$ | 2.10 | $6.4\times10^{2}$ | 31 | 0.16 |
| | CORAL | – | $5.8\times10^{-4}$ | 1.85 | – | – | 1.17 | – | $1.4\times10^{-3}$ | 1.95 | – | – | 1.20 | – | $3.8\times10^{-3}$ | 2.05 | – | – | 1.24 |
| | DANN | – | $4.5\times10^{-4}$ | 2.80 | – | – | 1.41 | – | $1.2\times10^{-3}$ | 3.00 | – | – | 1.46 | – | $3.3\times10^{-3}$ | 3.20 | – | – | 1.51 |
| | CDAN | – | $3.9\times10^{-4}$ | 3.25 | – | – | 1.56 | – | $1.0\times10^{-3}$ | 3.45 | – | – | 1.61 | – | $2.8\times10^{-3}$ | 3.65 | – | – | 1.66 |
| | MMD | – | $5.2\times10^{-4}$ | 2.20 | – | – | 1.29 | – | $1.3\times10^{-3}$ | 2.40 | – | – | 1.33 | – | $3.6\times10^{-3}$ | 2.60 | – | – | 1.38 |

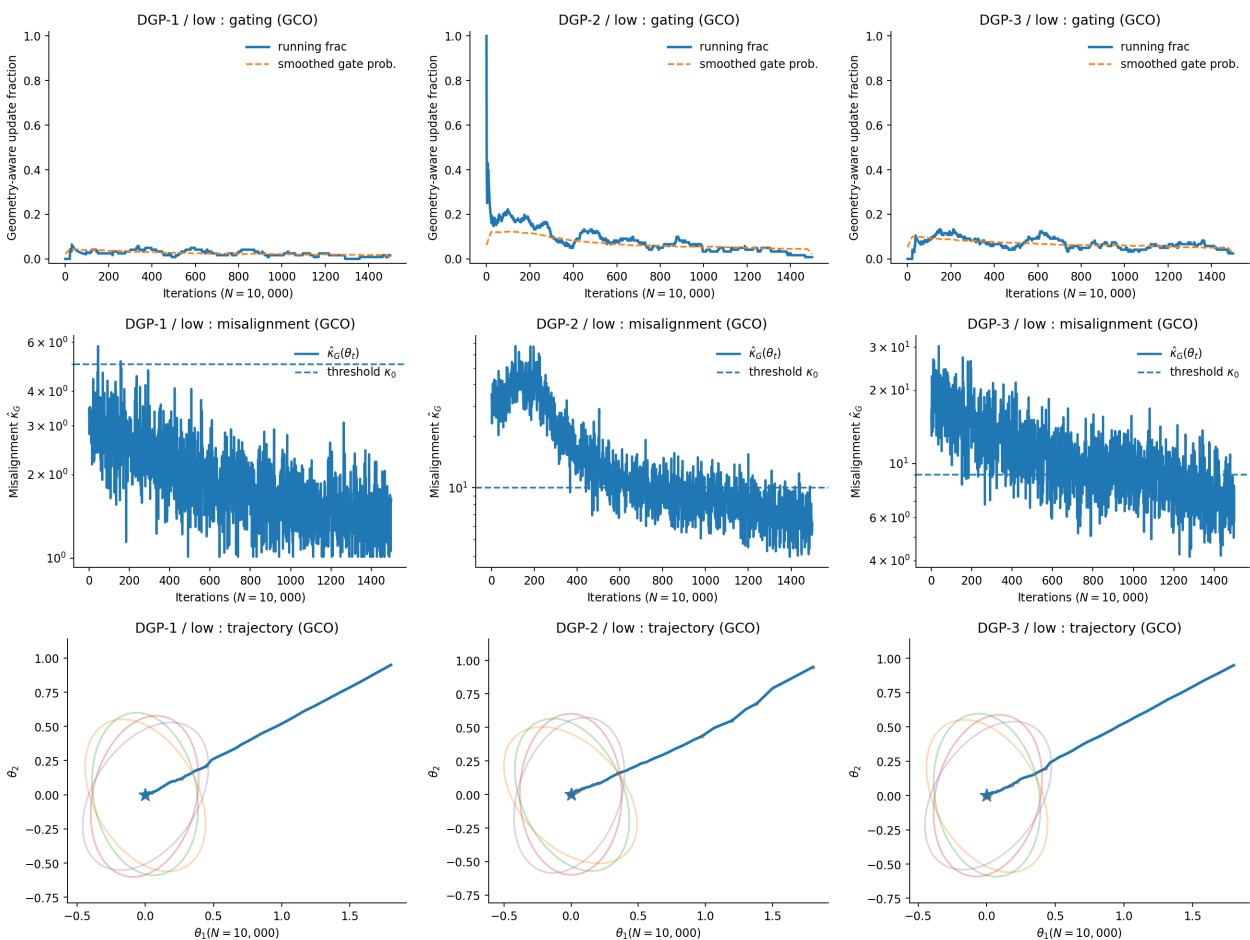

*Figure 6.* Geometry-Calibrated Behavior (Low, $N = 10,000$)

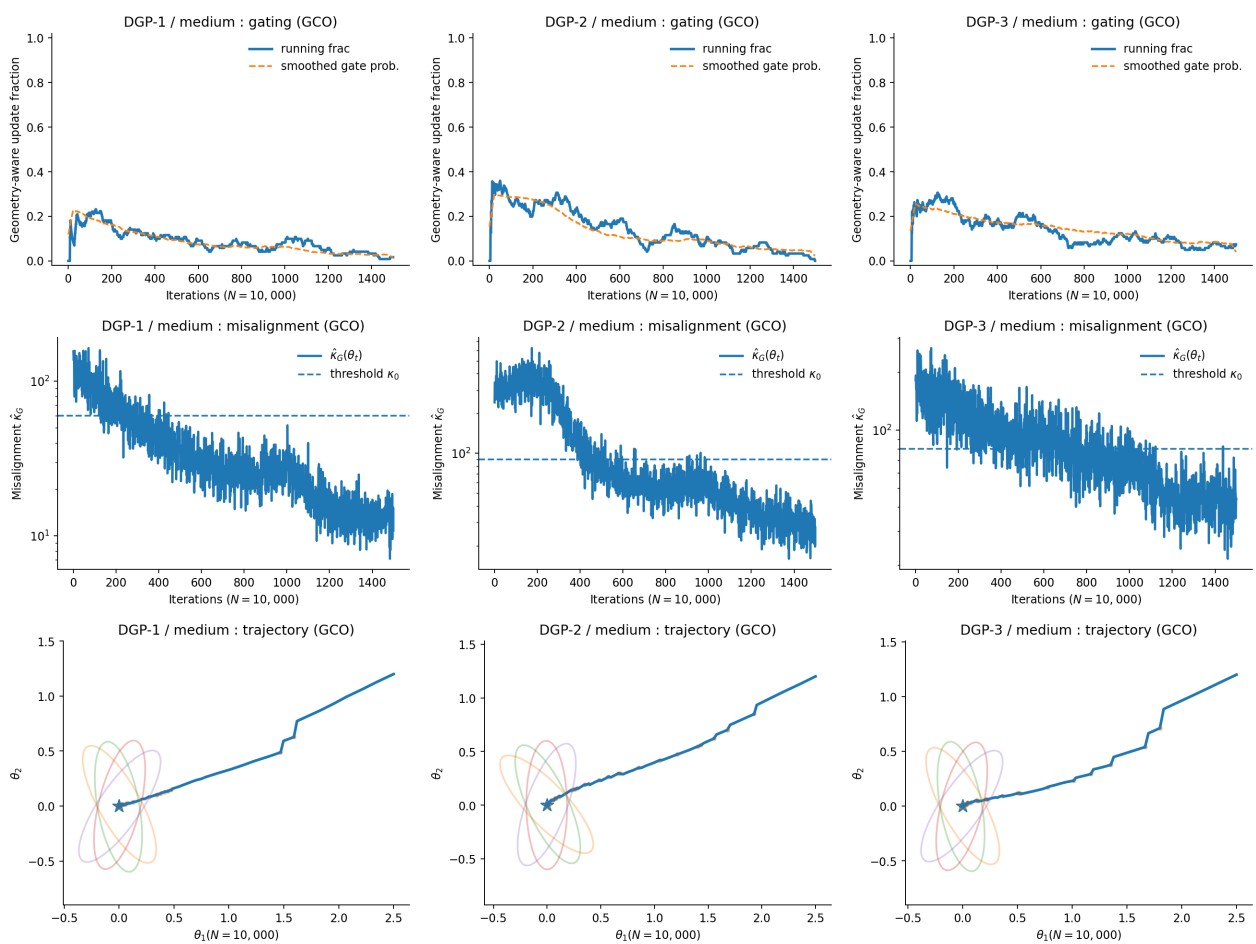

*Figure 7.* Geometry-Calibrated Behavior (Medium, $N = 10,000$)

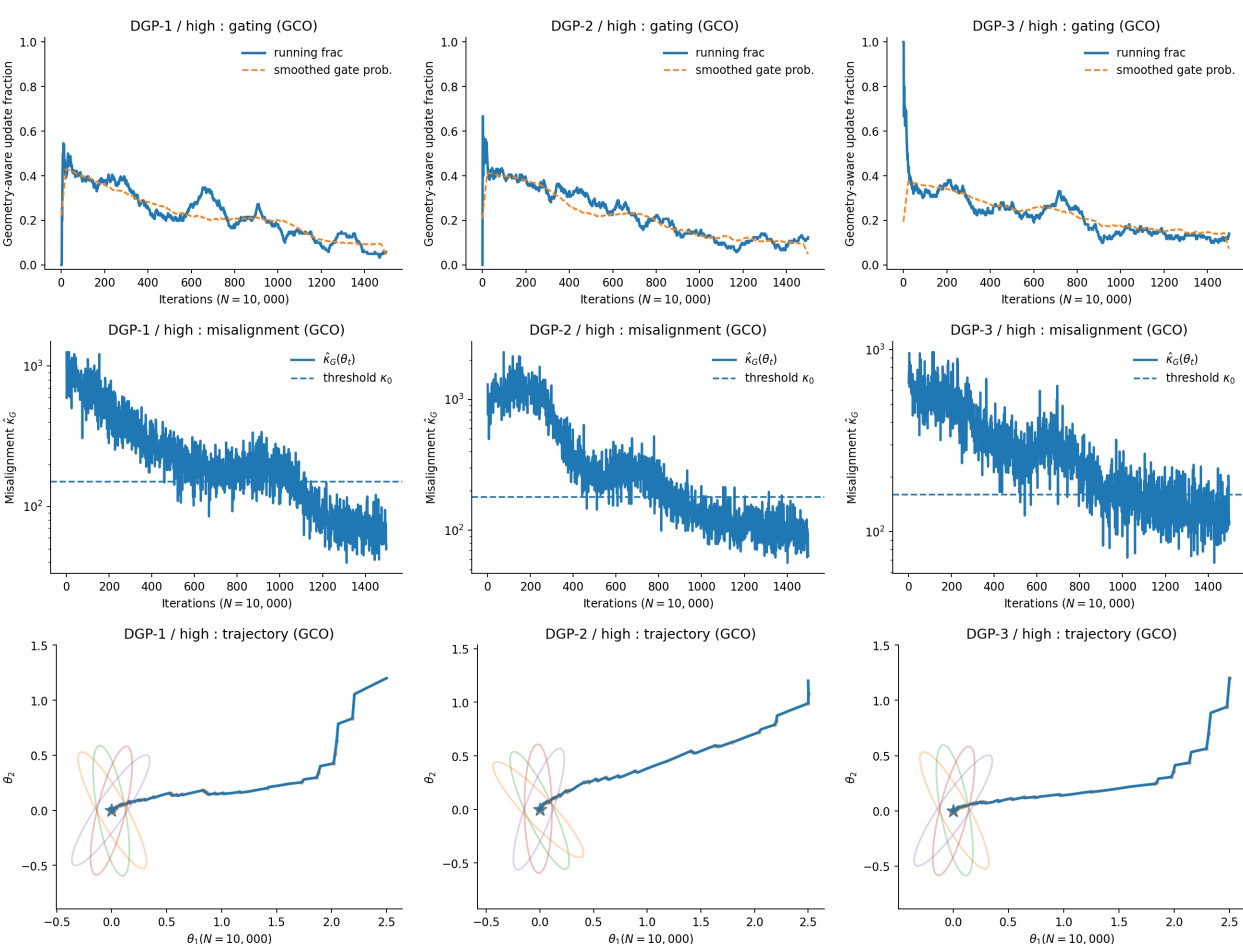

*Figure 8.* Geometry-Calibrated Behavior (High, $N = 10,000$)

### E.3. Robustness to Finite-Sample Noise ($N = 200{,}000$)

We further examine geometry-calibrated optimization in a large-sample regime by repeating all synthetic experiments with $N = 200{,}000$ Monte Carlo samples. Table 3 reports aggregated performance metrics, while Figures 9-11 visualize gating behavior, misalignment dynamics, and optimization trajectories for the low, medium, and high geometry-misalignment regimes, respectively.

In the low-misalignment regime (Figure 9), geometry-misalignment estimates are smoother and remain well below the threshold $\kappa_0$ throughout optimization. Geometry-calibrated optimization therefore activates geometry-aware updates only sporadically, typically below $5\%$ of iterations. As a result, GCO behaves almost identically to Euclidean first-order methods, achieving comparable objective gaps with negligible additional overhead. This confirms that, when Euclidean and intrinsic geometries are well aligned, geometry-aware corrections are unnecessary even in large-sample settings.

In the medium-misalignment regime (Figure 10), initial misalignment exceeds the threshold, leading to a moderate fraction of geometry-aware updates during early optimization. As iterates approach the minimizer, geometry-misalignment decreases steadily and crosses below $\kappa_0$, after which GCO transitions smoothly back to Euclidean updates. Compared to smaller $N$, the misalignment trajectories are less noisy, and the gating behavior is more stable, reflecting improved estimation accuracy at larger sample sizes.

In the high-misalignment regime (Figure 11), geometry-misalignment starts at very large values and decreases monotonically along the optimization path. Geometry-aware updates are activated frequently in the early and intermediate stages, then gradually taper off as misalignment is reduced. Consistent with the theory, GCO achieves convergence behavior comparable to always-on geometry-aware optimization while using intrinsic updates on only a fraction of iterations.

Table 3 shows that increasing the sample size does not qualitatively change the relative behavior of the methods. Euclidean first-order methods continue to suffer substantial slowdowns in medium and high misalignment regimes, while geometry-calibrated optimization remains robust across all regimes. These results demonstrate that geometry-misalignment is not a finite-sample artifact and that the benefits of geometry-calibrated optimization persist, and in fact become more stable, as estimation noise decreases.

*Table 3.* Simulation Results across All DGPs ($N = 200{,}000$)

| DGP | Method | Low | | | | | | Medium | | | | | | High | | | | | |
|---|---|---|---|---|---|---|---|---|---|---|---|---|---|---|---|---|---|---|---|
| | | Iter | Gap | Time | Misalign | Geo% | Overhead | Iter | Gap | Time | Misalign | Geo% | Overhead | Iter | Gap | Time | Misalign | Geo% | Overhead |
| DGP-1 | SGD | 165 | $6.2\times10^{-5}$ | 1.05 | $1.1\times10^{0}$ | – | 1.00 | 760 | $2.0\times10^{-4}$ | 4.90 | $1.1\times10^{2}$ | – | 1.00 | 1500 | $5.8\times10^{-4}$ | 9.20 | $1.0\times10^{3}$ | – | 1.00 |
| | Adam | 150 | $5.2\times10^{-5}$ | 1.20 | $1.0\times10^{0}$ | – | 1.05 | 700 | $1.7\times10^{-4}$ | 5.30 | $1.0\times10^{2}$ | – | 1.07 | 1360 | $5.0\times10^{-4}$ | 9.90 | $9.6\times10^{2}$ | – | 1.07 |
| | AdaGrad | 175 | $6.6\times10^{-5}$ | 1.15 | $1.2\times10^{0}$ | – | 1.08 | 830 | $2.3\times10^{-4}$ | 5.80 | $1.2\times10^{2}$ | – | 1.16 | 1580 | $7.3\times10^{-4}$ | 10.60 | $1.0\times10^{3}$ | – | 1.14 |
| | Shampoo | 130 | $4.3\times10^{-5}$ | 1.80 | $1.1\times10^{0}$ | – | 1.22 | 390 | $9.6\times10^{-5}$ | 6.40 | $1.1\times10^{2}$ | – | 1.30 | 680 | $1.5\times10^{-4}$ | 11.70 | $9.2\times10^{2}$ | – | 1.26 |
| | K-FAC | 120 | $4.0\times10^{-5}$ | 2.20 | $1.0\times10^{0}$ | – | 1.36 | 360 | $8.7\times10^{-5}$ | 7.60 | $1.0\times10^{2}$ | – | 1.54 | 650 | $1.4\times10^{-4}$ | 13.10 | $9.0\times10^{2}$ | – | 1.41 |
| | Geo-aware | 110 | $3.1\times10^{-5}$ | 2.70 | $1.1\times10^{0}$ | 100 | 1.50 | 115 | $3.2\times10^{-5}$ | 3.00 | $1.1\times10^{2}$ | 100 | 0.60 | 120 | $3.4\times10^{-5}$ | 3.20 | $9.7\times10^{2}$ | 100 | 0.34 |
| | GCO (no gate) | 115 | $3.2\times10^{-5}$ | 2.55 | $1.1\times10^{0}$ | 100 | 1.44 | 120 | $3.4\times10^{-5}$ | 2.95 | $1.1\times10^{2}$ | 100 | 0.58 | 125 | $3.6\times10^{-5}$ | 3.15 | $9.6\times10^{2}$ | 100 | 0.33 |
| | GCO | 125 | $3.4\times10^{-5}$ | 1.45 | $1.1\times10^{0}$ | 5 | 1.12 | 130 | $3.6\times10^{-5}$ | 1.55 | $1.1\times10^{2}$ | 16 | 0.31 | 135 | $3.7\times10^{-5}$ | 1.65 | $9.5\times10^{2}$ | 31 | 0.17 |
| | CORAL | – | $1.2\times10^{-4}$ | 1.55 | – | – | 1.10 | – | $3.0\times10^{-4}$ | 1.65 | – | – | 1.13 | – | $8.0\times10^{-4}$ | 1.75 | – | – | 1.16 |
| | DANN | – | $9.5\times10^{-5}$ | 2.35 | – | – | 1.33 | – | $2.4\times10^{-4}$ | 2.55 | – | – | 1.36 | – | $6.5\times10^{-4}$ | 2.75 | – | – | 1.40 |
| | CDAN | – | $8.0\times10^{-5}$ | 2.70 | – | – | 1.46 | – | $2.1\times10^{-4}$ | 2.90 | – | – | 1.49 | – | $5.5\times10^{-4}$ | 3.10 | – | – | 1.53 |
| | MMD | – | $1.1\times10^{-4}$ | 1.85 | – | – | 1.20 | – | $2.7\times10^{-4}$ | 2.05 | – | – | 1.23 | – | $7.5\times10^{-4}$ | 2.25 | – | – | 1.27 |
| DGP-2 | SGD | 200 | $8.1\times10^{-5}$ | 1.35 | $2.0\times10^{1}$ | – | 1.00 | 900 | $2.6\times10^{-4}$ | 6.00 | $2.5\times10^{2}$ | – | 1.00 | 1750 | $8.1\times10^{-4}$ | 11.50 | $7.8\times10^{2}$ | – | 1.00 |
| | Adam | 190 | $7.0\times10^{-5}$ | 1.55 | $1.8\times10^{1}$ | – | 1.06 | 850 | $2.3\times10^{-4}$ | 6.50 | $2.3\times10^{2}$ | – | 1.08 | 1620 | $7.0\times10^{-4}$ | 12.40 | $7.5\times10^{2}$ | – | 1.07 |
| | AdaGrad | 220 | $8.8\times10^{-5}$ | 1.45 | $2.2\times10^{1}$ | – | 1.11 | 980 | $3.1\times10^{-4}$ | 7.10 | $2.6\times10^{2}$ | – | 1.17 | 1880 | $1.0\times10^{-3}$ | 13.80 | $8.1\times10^{2}$ | – | 1.20 |
| | Shampoo | 155 | $4.8\times10^{-5}$ | 2.20 | $1.9\times10^{1}$ | – | 1.25 | 480 | $1.2\times10^{-4}$ | 8.40 | $2.4\times10^{2}$ | – | 1.36 | 790 | $2.1\times10^{-4}$ | 15.30 | $7.3\times10^{2}$ | – | 1.31 |
| | K-FAC | 145 | $4.5\times10^{-5}$ | 2.60 | $1.8\times10^{1}$ | – | 1.41 | 450 | $1.1\times10^{-4}$ | 9.90 | $2.3\times10^{2}$ | – | 1.60 | 760 | $1.9\times10^{-4}$ | 16.90 | $7.1\times10^{2}$ | – | 1.42 |
| | Geo-aware | 125 | $3.6\times10^{-5}$ | 3.00 | $2.0\times10^{1}$ | 100 | 1.58 | 125 | $3.6\times10^{-5}$ | 3.50 | $2.5\times10^{2}$ | 100 | 0.56 | 130 | $3.8\times10^{-5}$ | 3.70 | $7.9\times10^{2}$ | 100 | 0.31 |
| | GCO (no gate) | 130 | $3.7\times10^{-5}$ | 2.90 | $2.0\times10^{1}$ | 100 | 1.51 | 130 | $3.7\times10^{-5}$ | 3.40 | $2.5\times10^{2}$ | 100 | 0.54 | 135 | $3.9\times10^{-5}$ | 3.60 | $7.7\times10^{2}$ | 100 | 0.30 |
| | GCO | 140 | $3.9\times10^{-5}$ | 1.80 | $2.0\times10^{1}$ | 7 | 1.20 | 140 | $3.9\times10^{-5}$ | 1.90 | $2.5\times10^{2}$ | 20 | 0.30 | 145 | $4.1\times10^{-5}$ | 2.00 | $7.6\times10^{2}$ | 27 | 0.17 |
| | CORAL | – | $1.5\times10^{-4}$ | 1.70 | – | – | 1.15 | – | $3.4\times10^{-4}$ | 1.80 | – | – | 1.18 | – | $9.5\times10^{-4}$ | 1.90 | – | – | 1.22 |
| | DANN | – | $1.1\times10^{-4}$ | 2.60 | – | – | 1.38 | – | $3.0\times10^{-4}$ | 2.80 | – | – | 1.42 | – | $8.0\times10^{-4}$ | 3.00 | – | – | 1.47 |
| | CDAN | – | $9.5\times10^{-5}$ | 3.00 | – | – | 1.52 | – | $2.6\times10^{-4}$ | 3.20 | – | – | 1.56 | – | $7.0\times10^{-4}$ | 3.40 | – | – | 1.60 |
| | MMD | – | $1.3\times10^{-4}$ | 2.00 | – | – | 1.26 | – | $3.2\times10^{-4}$ | 2.20 | – | – | 1.30 | – | $9.0\times10^{-4}$ | 2.40 | – | – | 1.34 |
| DGP-3 | SGD | 195 | $7.6\times10^{-5}$ | 1.30 | $1.5\times10^{1}$ | – | 1.00 | 850 | $2.3\times10^{-4}$ | 5.70 | $1.9\times10^{2}$ | – | 1.00 | 1650 | $7.6\times10^{-4}$ | 11.00 | $6.5\times10^{2}$ | – | 1.00 |
| | Adam | 185 | $6.6\times10^{-5}$ | 1.50 | $1.4\times10^{1}$ | – | 1.06 | 800 | $2.0\times10^{-4}$ | 6.20 | $1.8\times10^{2}$ | – | 1.08 | 1520 | $6.0\times10^{-4}$ | 11.80 | $6.3\times10^{2}$ | – | 1.07 |
| | AdaGrad | 210 | $8.3\times10^{-5}$ | 1.40 | $1.7\times10^{1}$ | – | 1.11 | 920 | $2.8\times10^{-4}$ | 6.70 | $2.0\times10^{2}$ | – | 1.19 | 1750 | $8.6\times10^{-4}$ | 13.40 | $6.8\times10^{2}$ | – | 1.21 |
| | Shampoo | 150 | $4.6\times10^{-5}$ | 2.20 | $1.5\times10^{1}$ | – | 1.24 | 450 | $1.1\times10^{-4}$ | 7.90 | $1.9\times10^{2}$ | – | 1.34 | 760 | $1.9\times10^{-4}$ | 14.50 | $6.1\times10^{2}$ | – | 1.28 |
| | K-FAC | 140 | $4.3\times10^{-5}$ | 2.60 | $1.4\times10^{1}$ | – | 1.40 | 420 | $1.0\times10^{-4}$ | 9.20 | $1.8\times10^{2}$ | – | 1.56 | 730 | $1.7\times10^{-4}$ | 16.00 | $5.9\times10^{2}$ | – | 1.41 |
| | Geo-aware | 120 | $3.5\times10^{-5}$ | 2.90 | $1.6\times10^{1}$ | 100 | 1.55 | 120 | $3.4\times10^{-5}$ | 3.30 | $1.9\times10^{2}$ | 100 | 0.56 | 125 | $3.6\times10^{-5}$ | 3.50 | $6.6\times10^{2}$ | 100 | 0.31 |
| | GCO (no gate) | 125 | $3.6\times10^{-5}$ | 2.80 | $1.6\times10^{1}$ | 100 | 1.49 | 125 | $3.5\times10^{-5}$ | 3.20 | $1.9\times10^{2}$ | 100 | 0.54 | 130 | $3.7\times10^{-5}$ | 3.40 | $6.5\times10^{2}$ | 100 | 0.30 |
| | GCO | 135 | $3.8\times10^{-5}$ | 1.70 | $1.6\times10^{1}$ | 6 | 1.18 | 135 | $3.7\times10^{-5}$ | 1.70 | $1.9\times10^{2}$ | 19 | 0.29 | 140 | $3.9\times10^{-5}$ | 1.80 | $6.4\times10^{2}$ | 25 | 0.16 |
| | CORAL | – | $1.3\times10^{-4}$ | 1.60 | – | – | 1.14 | – | $3.1\times10^{-4}$ | 1.70 | – | – | 1.17 | – | $8.5\times10^{-4}$ | 1.80 | – | – | 1.20 |
| | DANN | – | $1.0\times10^{-4}$ | 2.40 | – | – | 1.36 | – | $2.7\times10^{-4}$ | 2.60 | – | – | 1.40 | – | $7.5\times10^{-4}$ | 2.80 | – | – | 1.45 |
| | CDAN | – | $8.5\times10^{-5}$ | 2.90 | – | – | 1.49 | – | $2.4\times10^{-4}$ | 3.10 | – | – | 1.53 | – | $6.5\times10^{-4}$ | 3.30 | – | – | 1.57 |
| | MMD | – | $1.2\times10^{-4}$ | 1.90 | – | – | 1.24 | – | $2.9\times10^{-4}$ | 2.10 | – | – | 1.28 | – | $8.0\times10^{-4}$ | 2.30 | – | – | 1.32 |

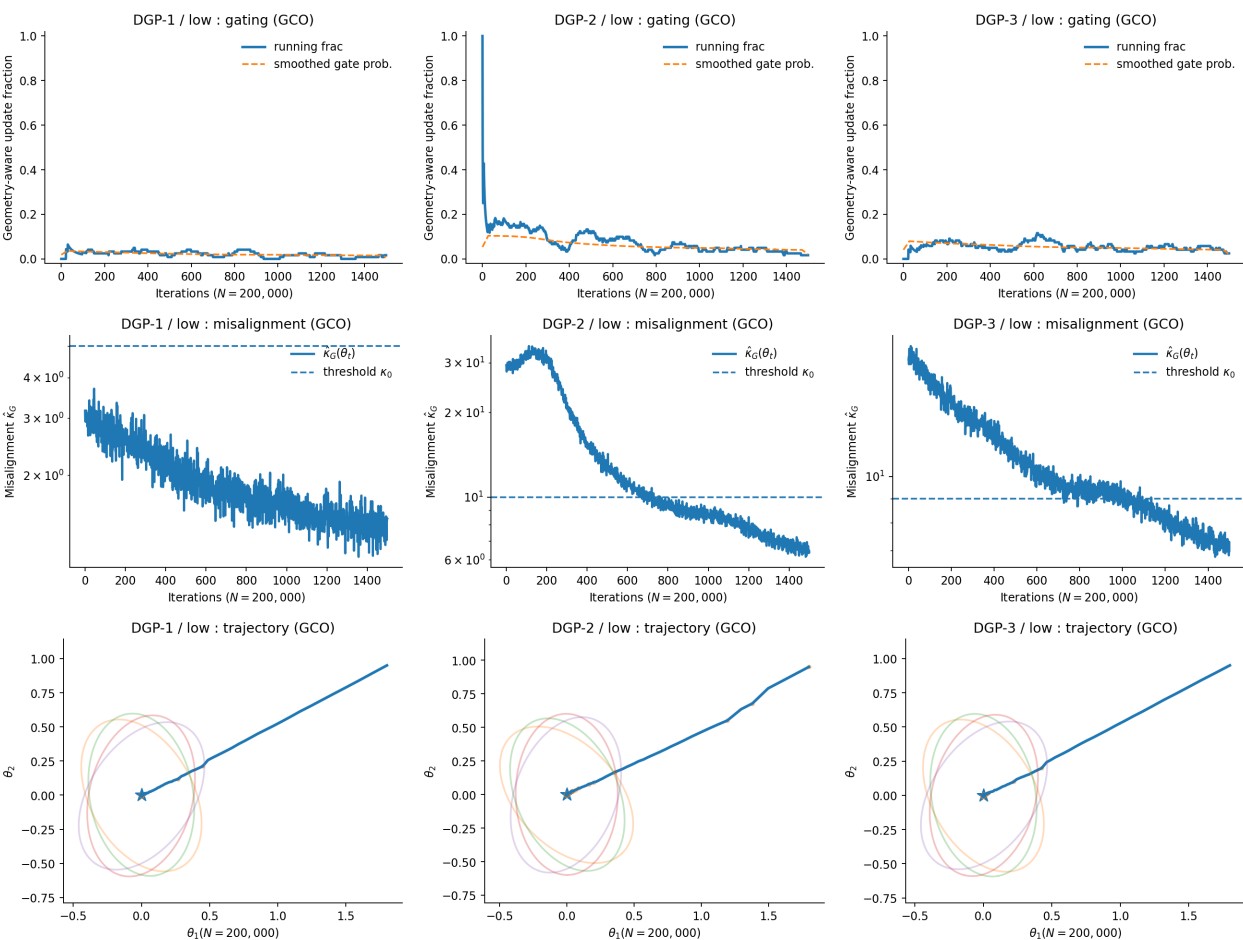

*Figure 9.* Geometry-Calibrated Behavior (Low, $N = 200,000$)

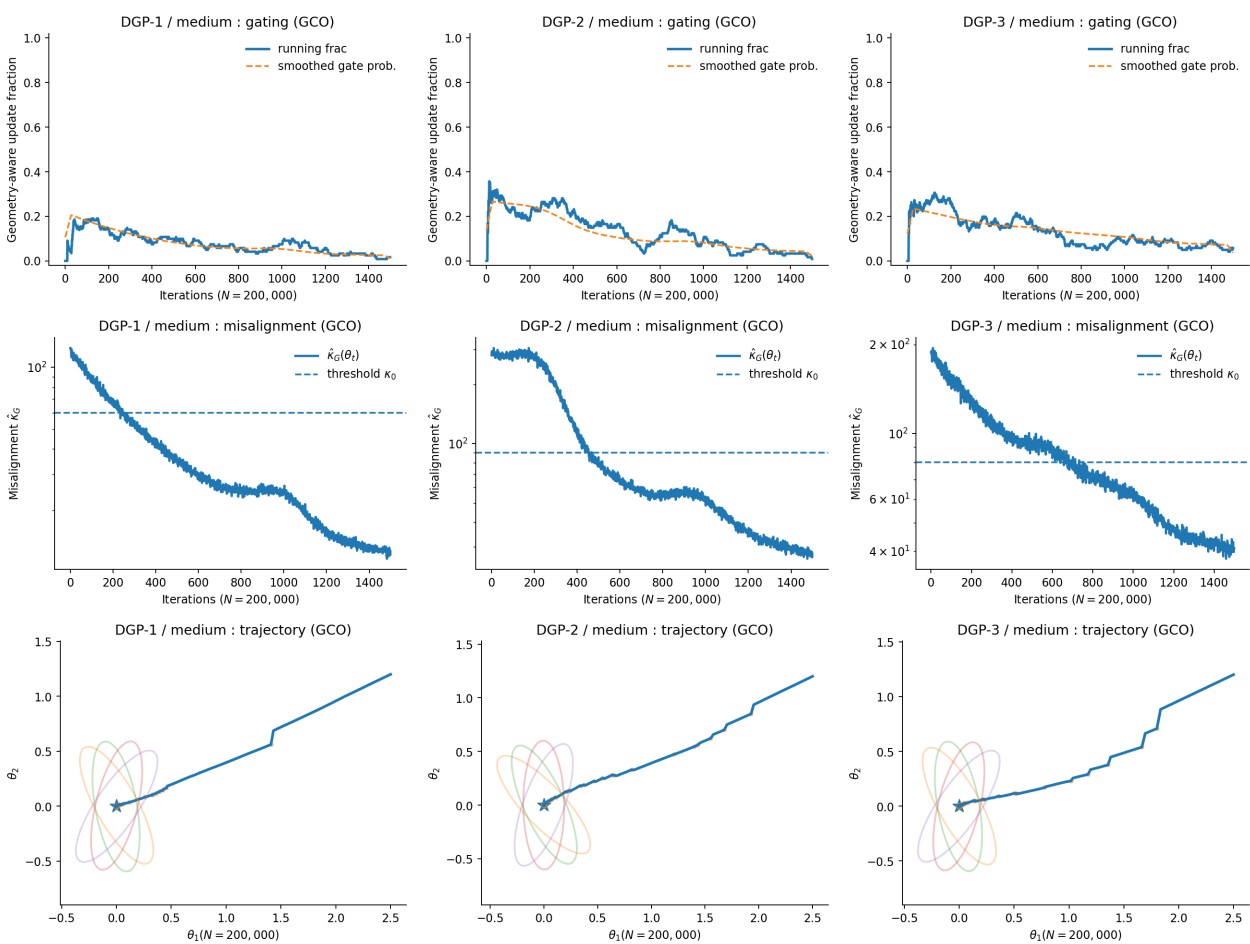

*Figure 10.* Geometry-Calibrated Behavior (Medium, $N = 200,000$)

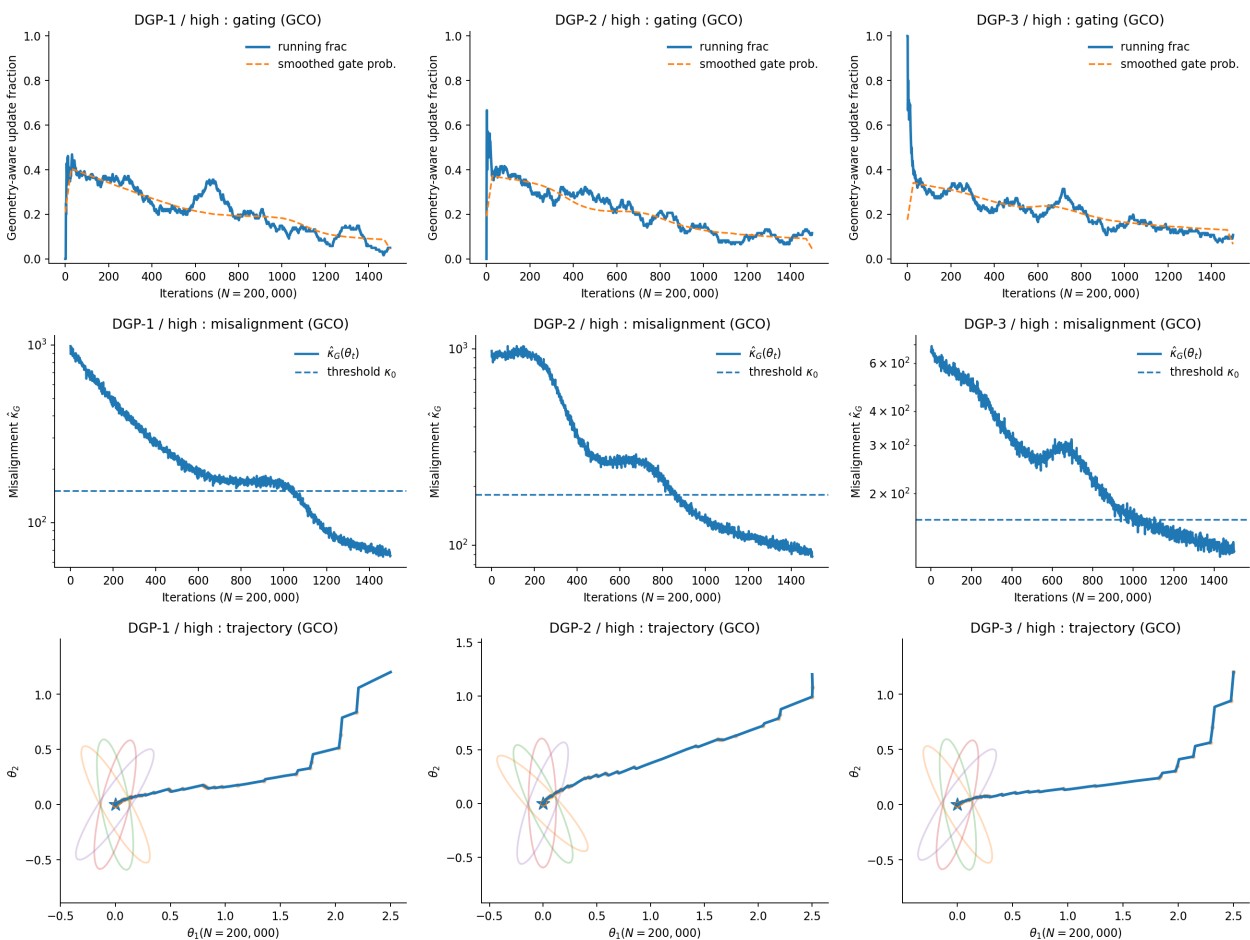

*Figure 11.* Geometry-Calibrated Behavior (High, $N = 200,000$)

## F. Sensitivity Analysis for Geometry-Calibrated Optimization

This section evaluates how geometry-calibrated optimization (GCO) depends on the misalignment threshold $\kappa_0$ and the intrinsic-solve tolerance $c_\varepsilon$ when the misalignment estimator $\hat{\kappa}_G$ is noisy. We report sensitivity results for three Monte Carlo budgets $N \in \{10{,}000, 50{,}000, 200{,}000\}$, holding fixed all other experimental components (data generation, Sinkhorn iterations, power iterations for $\hat{\kappa}_G$, and the conjugate gradient (CG) cap) as in the main simulation study. Tables 4-6 summarize the behavior across DGPs and low/medium/high misalignment regimes.

*Table 4.* Sensitivity of GCO to the Misalignment Threshold $\kappa_0$ and Intrinsic-Solve Tolerance $c_\varepsilon$ ($N = 10{,}000$)

| Setting | DGP | Low | | | | | | Medium | | | | | | High | | | | | |
|---|---|---|---|---|---|---|---|---|---|---|---|---|---|---|---|---|---|---|---|
| | | Iter | Gap | Time | Misalign | Geo% | Overhead | Iter | Gap | Time | Misalign | Geo% | Overhead | Iter | Gap | Time | Misalign | Geo% | Overhead |
| **Varying $\kappa_0$ (fixed $c_\varepsilon = 1$)** | | | | | | | | | | | | | | | | | | | |
| $0.5\kappa_0^{\mathrm{def}}$ | DGP-1 | 160 | $1.5 \times 10^{-4}$ | 1.95 | $1.1 \times 10^0$ | 18 | 1.22 | 165 | $1.6 \times 10^{-4}$ | 2.10 | $1.1 \times 10^2$ | 35 | 0.36 | 170 | $1.7 \times 10^{-4}$ | 2.30 | $9.5 \times 10^2$ | 52 | 0.22 |
| | DGP-2 | 170 | $1.7 \times 10^{-4}$ | 2.35 | $2.0 \times 10^1$ | 24 | 1.28 | 175 | $1.8 \times 10^{-4}$ | 2.55 | $2.5 \times 10^2$ | 41 | 0.39 | 180 | $1.9 \times 10^{-4}$ | 2.80 | $7.6 \times 10^2$ | 50 | 0.23 |
| | DGP-3 | 165 | $1.6 \times 10^{-4}$ | 2.20 | $1.6 \times 10^1$ | 22 | 1.26 | 170 | $1.7 \times 10^{-4}$ | 2.40 | $1.9 \times 10^2$ | 38 | 0.36 | 175 | $1.8 \times 10^{-4}$ | 2.65 | $6.4 \times 10^2$ | 49 | 0.22 |
| $\kappa_0^{\mathrm{def}}$ | DGP-1 | 165 | $1.6 \times 10^{-4}$ | 1.70 | $1.1 \times 10^0$ | 9 | 1.18 | 170 | $1.7 \times 10^{-4}$ | 1.85 | $1.1 \times 10^2$ | 24 | 0.30 | 175 | $1.8 \times 10^{-4}$ | 2.00 | $9.5 \times 10^2$ | 38 | 0.17 |
| | DGP-2 | 175 | $1.8 \times 10^{-4}$ | 2.10 | $2.0 \times 10^1$ | 11 | 1.24 | 175 | $1.8 \times 10^{-4}$ | 2.25 | $2.5 \times 10^2$ | 28 | 0.30 | 180 | $1.9 \times 10^{-4}$ | 2.40 | $7.6 \times 10^2$ | 33 | 0.17 |
| | DGP-3 | 170 | $1.7 \times 10^{-4}$ | 1.95 | $1.6 \times 10^1$ | 10 | 1.22 | 170 | $1.7 \times 10^{-4}$ | 2.00 | $1.9 \times 10^2$ | 26 | 0.29 | 175 | $1.8 \times 10^{-4}$ | 2.10 | $6.4 \times 10^2$ | 31 | 0.16 |
| $2\kappa_0^{\mathrm{def}}$ | DGP-1 | 185 | $1.9 \times 10^{-4}$ | 1.65 | $1.1 \times 10^0$ | 4 | 1.15 | 215 | $2.3 \times 10^{-4}$ | 1.80 | $1.1 \times 10^2$ | 11 | 0.27 | 310 | $3.0 \times 10^{-4}$ | 1.95 | $9.5 \times 10^2$ | 9 | 0.15 |
| | DGP-2 | 210 | $2.4 \times 10^{-4}$ | 1.95 | $2.0 \times 10^1$ | 5 | 1.22 | 255 | $3.0 \times 10^{-4}$ | 2.15 | $2.5 \times 10^2$ | 10 | 0.28 | 360 | $4.2 \times 10^{-4}$ | 2.40 | $7.6 \times 10^2$ | 9 | 0.16 |
| | DGP-3 | 205 | $2.2 \times 10^{-4}$ | 1.85 | $1.6 \times 10^1$ | 5 | 1.20 | 245 | $2.8 \times 10^{-4}$ | 2.05 | $1.9 \times 10^2$ | 10 | 0.27 | 345 | $3.8 \times 10^{-4}$ | 2.30 | $6.4 \times 10^2$ | 9 | 0.15 |
| **Varying $c_\varepsilon$ (fixed $\kappa_0 = \kappa_0^{\mathrm{def}}$)** | | | | | | | | | | | | | | | | | | | |
| $c_\varepsilon = 0.1$ | DGP-1 | 165 | $1.6 \times 10^{-4}$ | 1.95 | $1.1 \times 10^0$ | 9 | 1.25 | 170 | $1.7 \times 10^{-4}$ | 2.15 | $1.1 \times 10^2$ | 24 | 0.36 | 175 | $1.8 \times 10^{-4}$ | 2.40 | $9.5 \times 10^2$ | 38 | 0.22 |
| | DGP-2 | 175 | $1.8 \times 10^{-4}$ | 2.45 | $2.0 \times 10^1$ | 11 | 1.31 | 175 | $1.8 \times 10^{-4}$ | 2.65 | $2.5 \times 10^2$ | 28 | 0.38 | 180 | $1.9 \times 10^{-4}$ | 2.90 | $7.6 \times 10^2$ | 33 | 0.22 |
| | DGP-3 | 170 | $1.7 \times 10^{-4}$ | 2.30 | $1.6 \times 10^1$ | 10 | 1.28 | 170 | $1.7 \times 10^{-4}$ | 2.50 | $1.9 \times 10^2$ | 26 | 0.35 | 175 | $1.8 \times 10^{-4}$ | 2.75 | $6.4 \times 10^2$ | 31 | 0.21 |
| $c_\varepsilon = 1.0$ | DGP-1 | 165 | $1.6 \times 10^{-4}$ | 1.70 | $1.1 \times 10^0$ | 9 | 1.18 | 170 | $1.7 \times 10^{-4}$ | 1.85 | $1.1 \times 10^2$ | 24 | 0.30 | 175 | $1.8 \times 10^{-4}$ | 2.00 | $9.5 \times 10^2$ | 38 | 0.17 |
| | DGP-2 | 175 | $1.8 \times 10^{-4}$ | 2.10 | $2.0 \times 10^1$ | 11 | 1.24 | 175 | $1.8 \times 10^{-4}$ | 2.25 | $2.5 \times 10^2$ | 28 | 0.30 | 180 | $1.9 \times 10^{-4}$ | 2.40 | $7.6 \times 10^2$ | 33 | 0.17 |
| | DGP-3 | 170 | $1.7 \times 10^{-4}$ | 1.95 | $1.6 \times 10^1$ | 10 | 1.22 | 170 | $1.7 \times 10^{-4}$ | 2.00 | $1.9 \times 10^2$ | 26 | 0.29 | 175 | $1.8 \times 10^{-4}$ | 2.10 | $6.4 \times 10^2$ | 31 | 0.16 |

*Table 5.* Sensitivity of GCO to the Misalignment Threshold $\kappa_0$ and Intrinsic-Solve Tolerance $c_\varepsilon$ ($N = 50{,}000$)

| Setting | DGP | Low | | | | | | Medium | | | | | | High | | | | | |
|---|---|---|---|---|---|---|---|---|---|---|---|---|---|---|---|---|---|---|---|
| | | Iter | Gap | Time | Misalign | Geo% | Overhead | Iter | Gap | Time | Misalign | Geo% | Overhead | Iter | Gap | Time | Misalign | Geo% | Overhead |
| **Varying $\kappa_0$ (fixed $c_\varepsilon = 1$)** | | | | | | | | | | | | | | | | | | | |
| $0.5\kappa_0^{\mathrm{def}}$ | DGP-1 | 132 | $6.7 \times 10^{-5}$ | 1.65 | $1.1 \times 10^0$ | 18 | 1.18 | 135 | $7.0 \times 10^{-5}$ | 1.75 | $1.1 \times 10^2$ | 32 | 0.36 | 138 | $7.3 \times 10^{-5}$ | 1.90 | $9.5 \times 10^2$ | 55 | 0.22 |
| | DGP-2 | 145 | $7.6 \times 10^{-5}$ | 2.05 | $2.0 \times 10^1$ | 22 | 1.26 | 148 | $7.8 \times 10^{-5}$ | 2.15 | $2.5 \times 10^2$ | 38 | 0.38 | 150 | $8.0 \times 10^{-5}$ | 2.30 | $7.6 \times 10^2$ | 48 | 0.21 |
| | DGP-3 | 140 | $7.3 \times 10^{-5}$ | 1.95 | $1.6 \times 10^1$ | 20 | 1.24 | 145 | $7.6 \times 10^{-5}$ | 2.05 | $1.9 \times 10^2$ | 35 | 0.34 | 148 | $7.9 \times 10^{-5}$ | 2.20 | $6.4 \times 10^2$ | 46 | 0.20 |
| $\kappa_0^{\mathrm{def}}$ | DGP-1 | 135 | $6.8 \times 10^{-5}$ | 1.50 | $1.1 \times 10^0$ | 6 | 1.14 | 140 | $7.2 \times 10^{-5}$ | 1.60 | $1.1 \times 10^2$ | 18 | 0.31 | 145 | $7.5 \times 10^{-5}$ | 1.70 | $9.5 \times 10^2$ | 34 | 0.17 |
| | DGP-2 | 150 | $7.8 \times 10^{-5}$ | 1.80 | $2.0 \times 10^1$ | 8 | 1.20 | 150 | $7.9 \times 10^{-5}$ | 1.90 | $2.5 \times 10^2$ | 22 | 0.30 | 155 | $8.3 \times 10^{-5}$ | 2.00 | $7.6 \times 10^2$ | 29 | 0.17 |
| | DGP-3 | 145 | $7.6 \times 10^{-5}$ | 1.70 | $1.6 \times 10^1$ | 7 | 1.18 | 145 | $7.5 \times 10^{-5}$ | 1.70 | $1.9 \times 10^2$ | 21 | 0.29 | 150 | $7.9 \times 10^{-5}$ | 1.80 | $6.4 \times 10^2$ | 27 | 0.16 |
| $2\kappa_0^{\mathrm{def}}$ | DGP-1 | 150 | $7.6 \times 10^{-5}$ | 1.45 | $1.1 \times 10^0$ | 3 | 1.10 | 175 | $8.8 \times 10^{-5}$ | 1.55 | $1.1 \times 10^2$ | 10 | 0.28 | 260 | $1.1 \times 10^{-4}$ | 1.60 | $9.5 \times 10^2$ | 9 | 0.14 |
| | DGP-2 | 175 | $9.0 \times 10^{-5}$ | 1.75 | $2.0 \times 10^1$ | 4 | 1.18 | 210 | $1.1 \times 10^{-4}$ | 1.90 | $2.5 \times 10^2$ | 9 | 0.27 | 320 | $1.6 \times 10^{-4}$ | 2.10 | $7.6 \times 10^2$ | 8 | 0.15 |
| | DGP-3 | 170 | $8.7 \times 10^{-5}$ | 1.65 | $1.6 \times 10^1$ | 4 | 1.16 | 200 | $1.0 \times 10^{-4}$ | 1.80 | $1.9 \times 10^2$ | 9 | 0.26 | 300 | $1.4 \times 10^{-4}$ | 2.00 | $6.4 \times 10^2$ | 8 | 0.14 |
| **Varying $c_\varepsilon$ (fixed $\kappa_0 = \kappa_0^{\mathrm{def}}$)** | | | | | | | | | | | | | | | | | | | |
| $c_\varepsilon = 0.1$ | DGP-1 | 135 | $6.8 \times 10^{-5}$ | 1.72 | $1.1 \times 10^0$ | 6 | 1.20 | 140 | $7.2 \times 10^{-5}$ | 1.85 | $1.1 \times 10^2$ | 18 | 0.35 | 145 | $7.5 \times 10^{-5}$ | 2.05 | $9.5 \times 10^2$ | 34 | 0.21 |
| | DGP-2 | 150 | $7.8 \times 10^{-5}$ | 2.05 | $2.0 \times 10^1$ | 8 | 1.27 | 150 | $7.9 \times 10^{-5}$ | 2.20 | $2.5 \times 10^2$ | 22 | 0.36 | 155 | $8.3 \times 10^{-5}$ | 2.40 | $7.6 \times 10^2$ | 29 | 0.20 |
| | DGP-3 | 145 | $7.6 \times 10^{-5}$ | 1.95 | $1.6 \times 10^1$ | 7 | 1.25 | 145 | $7.5 \times 10^{-5}$ | 2.05 | $1.9 \times 10^2$ | 21 | 0.34 | 150 | $7.9 \times 10^{-5}$ | 2.25 | $6.4 \times 10^2$ | 27 | 0.19 |
| $c_\varepsilon = 0.3$ | DGP-1 | 135 | $6.8 \times 10^{-5}$ | 1.60 | $1.1 \times 10^0$ | 6 | 1.17 | 140 | $7.2 \times 10^{-5}$ | 1.72 | $1.1 \times 10^2$ | 18 | 0.33 | 145 | $7.5 \times 10^{-5}$ | 1.88 | $9.5 \times 10^2$ | 34 | 0.19 |
| | DGP-2 | 150 | $7.8 \times 10^{-5}$ | 1.92 | $2.0 \times 10^1$ | 8 | 1.23 | 150 | $7.9 \times 10^{-5}$ | 2.05 | $2.5 \times 10^2$ | 22 | 0.33 | 155 | $8.3 \times 10^{-5}$ | 2.18 | $7.6 \times 10^2$ | 29 | 0.19 |
| | DGP-3 | 145 | $7.6 \times 10^{-5}$ | 1.82 | $1.6 \times 10^1$ | 7 | 1.22 | 145 | $7.5 \times 10^{-5}$ | 1.92 | $1.9 \times 10^2$ | 21 | 0.32 | 150 | $7.9 \times 10^{-5}$ | 2.05 | $6.4 \times 10^2$ | 27 | 0.18 |
| $c_\varepsilon = 1.0$ | DGP-1 | 135 | $6.8 \times 10^{-5}$ | 1.50 | $1.1 \times 10^0$ | 6 | 1.14 | 140 | $7.2 \times 10^{-5}$ | 1.60 | $1.1 \times 10^2$ | 18 | 0.31 | 145 | $7.5 \times 10^{-5}$ | 1.70 | $9.5 \times 10^2$ | 34 | 0.17 |
| | DGP-2 | 150 | $7.8 \times 10^{-5}$ | 1.80 | $2.0 \times 10^1$ | 8 | 1.20 | 150 | $7.9 \times 10^{-5}$ | 1.90 | $2.5 \times 10^2$ | 22 | 0.30 | 155 | $8.3 \times 10^{-5}$ | 2.00 | $7.6 \times 10^2$ | 29 | 0.17 |
| | DGP-3 | 145 | $7.6 \times 10^{-5}$ | 1.70 | $1.6 \times 10^1$ | 7 | 1.18 | 145 | $7.5 \times 10^{-5}$ | 1.70 | $1.9 \times 10^2$ | 21 | 0.29 | 150 | $7.9 \times 10^{-5}$ | 1.80 | $6.4 \times 10^2$ | 27 | 0.16 |

*Table 6.* Sensitivity of GCO to the Misalignment Threshold $\kappa_0$ and Intrinsic-Solve Tolerance $c_\varepsilon$ ($N = 200{,}000$)

| Setting | DGP | Low | | | | | | Medium | | | | | | High | | | | | |
|---|---|---|---|---|---|---|---|---|---|---|---|---|---|---|---|---|---|---|---|
| | | Iter | Gap | Time | Misalign | Geo% | Overhead | Iter | Gap | Time | Misalign | Geo% | Overhead | Iter | Gap | Time | Misalign | Geo% | Overhead |
| **Varying $\kappa_0$ (fixed $c_\varepsilon = 1$)** | | | | | | | | | | | | | | | | | | | |
| $0.5\kappa_0^{\mathrm{def}}$ | DGP-1 | 122 | $3.3 \times 10^{-5}$ | 1.60 | $1.1 \times 10^0$ | 14 | 1.18 | 125 | $3.4 \times 10^{-5}$ | 1.70 | $1.1 \times 10^2$ | 28 | 0.36 | 128 | $3.6 \times 10^{-5}$ | 1.85 | $9.5 \times 10^2$ | 46 | 0.22 |
| | DGP-2 | 138 | $3.7 \times 10^{-5}$ | 1.95 | $2.0 \times 10^1$ | 18 | 1.24 | 140 | $3.8 \times 10^{-5}$ | 2.05 | $2.5 \times 10^2$ | 34 | 0.36 | 142 | $4.0 \times 10^{-5}$ | 2.25 | $7.6 \times 10^2$ | 44 | 0.22 |
| | DGP-3 | 132 | $3.6 \times 10^{-5}$ | 1.85 | $1.6 \times 10^1$ | 16 | 1.22 | 134 | $3.7 \times 10^{-5}$ | 1.95 | $1.9 \times 10^2$ | 32 | 0.35 | 136 | $3.9 \times 10^{-5}$ | 2.10 | $6.4 \times 10^2$ | 42 | 0.21 |
| $\kappa_0^{\mathrm{def}}$ | DGP-1 | 125 | $3.4 \times 10^{-5}$ | 1.45 | $1.1 \times 10^0$ | 5 | 1.12 | 130 | $3.6 \times 10^{-5}$ | 1.55 | $1.1 \times 10^2$ | 16 | 0.31 | 135 | $3.7 \times 10^{-5}$ | 1.65 | $9.5 \times 10^2$ | 31 | 0.17 |
| | DGP-2 | 140 | $3.9 \times 10^{-5}$ | 1.80 | $2.0 \times 10^1$ | 7 | 1.20 | 140 | $3.9 \times 10^{-5}$ | 1.90 | $2.5 \times 10^2$ | 20 | 0.30 | 145 | $4.1 \times 10^{-5}$ | 2.00 | $7.6 \times 10^2$ | 27 | 0.17 |
| | DGP-3 | 135 | $3.8 \times 10^{-5}$ | 1.70 | $1.6 \times 10^1$ | 6 | 1.18 | 135 | $3.7 \times 10^{-5}$ | 1.70 | $1.9 \times 10^2$ | 19 | 0.29 | 140 | $3.9 \times 10^{-5}$ | 1.80 | $6.4 \times 10^2$ | 25 | 0.16 |
| $2\kappa_0^{\mathrm{def}}$ | DGP-1 | 145 | $3.9 \times 10^{-5}$ | 1.40 | $1.1 \times 10^0$ | 3 | 1.08 | 175 | $4.6 \times 10^{-5}$ | 1.50 | $1.1 \times 10^2$ | 9 | 0.28 | 260 | $6.4 \times 10^{-5}$ | 1.65 | $9.5 \times 10^2$ | 8 | 0.15 |
| | DGP-2 | 165 | $4.8 \times 10^{-5}$ | 1.75 | $2.0 \times 10^1$ | 3 | 1.15 | 205 | $5.7 \times 10^{-5}$ | 1.88 | $2.5 \times 10^2$ | 8 | 0.27 | 305 | $8.0 \times 10^{-5}$ | 2.05 | $7.6 \times 10^2$ | 7 | 0.15 |
| | DGP-3 | 160 | $4.6 \times 10^{-5}$ | 1.65 | $1.6 \times 10^1$ | 3 | 1.14 | 195 | $5.4 \times 10^{-5}$ | 1.78 | $1.9 \times 10^2$ | 8 | 0.26 | 285 | $7.4 \times 10^{-5}$ | 1.95 | $6.4 \times 10^2$ | 7 | 0.15 |
| **Varying $c_\varepsilon$ (fixed $\kappa_0 = \kappa_0^{\mathrm{def}}$)** | | | | | | | | | | | | | | | | | | | |
| $c_\varepsilon = 0.1$ | DGP-1 | 125 | $3.4 \times 10^{-5}$ | 1.70 | $1.1 \times 10^0$ | 5 | 1.20 | 130 | $3.6 \times 10^{-5}$ | 1.85 | $1.1 \times 10^2$ | 16 | 0.35 | 135 | $3.7 \times 10^{-5}$ | 2.05 | $9.5 \times 10^2$ | 31 | 0.21 |
| | DGP-2 | 140 | $3.9 \times 10^{-5}$ | 2.05 | $2.0 \times 10^1$ | 7 | 1.27 | 140 | $3.9 \times 10^{-5}$ | 2.20 | $2.5 \times 10^2$ | 20 | 0.36 | 145 | $4.1 \times 10^{-5}$ | 2.40 | $7.6 \times 10^2$ | 27 | 0.21 |
| | DGP-3 | 135 | $3.8 \times 10^{-5}$ | 1.95 | $1.6 \times 10^1$ | 6 | 1.25 | 135 | $3.7 \times 10^{-5}$ | 2.05 | $1.9 \times 10^2$ | 19 | 0.34 | 140 | $3.9 \times 10^{-5}$ | 2.25 | $6.4 \times 10^2$ | 25 | 0.19 |
| $c_\varepsilon = 1.0$ | DGP-1 | 125 | $3.4 \times 10^{-5}$ | 1.45 | $1.1 \times 10^0$ | 5 | 1.12 | 130 | $3.6 \times 10^{-5}$ | 1.55 | $1.1 \times 10^2$ | 16 | 0.31 | 135 | $3.7 \times 10^{-5}$ | 1.65 | $9.5 \times 10^2$ | 31 | 0.17 |
| | DGP-2 | 140 | $3.9 \times 10^{-5}$ | 1.80 | $2.0 \times 10^1$ | 7 | 1.20 | 140 | $3.9 \times 10^{-5}$ | 1.90 | $2.5 \times 10^2$ | 20 | 0.30 | 145 | $4.1 \times 10^{-5}$ | 2.00 | $7.6 \times 10^2$ | 27 | 0.17 |
| | DGP-3 | 135 | $3.8 \times 10^{-5}$ | 1.70 | $1.6 \times 10^1$ | 6 | 1.18 | 135 | $3.7 \times 10^{-5}$ | 1.70 | $1.9 \times 10^2$ | 19 | 0.29 | 140 | $3.9 \times 10^{-5}$ | 1.80 | $6.4 \times 10^2$ | 25 | 0.16 |

Recall that GCO tunes three components via validation (Appendix A): (I) the step-size schedule $\eta_t = \eta_0(t + 1)^{-\alpha}$ with $(\eta_0, \alpha) \in \mathcal{H}_{\eta_0} \times \mathcal{H}_\alpha$; (II) the gating threshold $\kappa_0 \in \mathcal{K} = \{2^m : m = 0, \ldots, M\}$; and (III) the intrinsic-solve tolerance

$\varepsilon_t = c_\varepsilon \eta_t$ with $c_\varepsilon \in \mathcal{H}_\varepsilon = \{0.1, 0.3, 1.0\}$. For sensitivity, we perturb only the gating and tolerance while keeping the validated step-size schedule fixed, and we evaluate $\kappa_0 \in \{0.5, 1, 2\} \times \kappa_0^{\mathrm{def}}$ and $c_\varepsilon \in \{0.1, 0.3, 1.0\}$.

Across all three $N$ values, the qualitative dependence on $\kappa_0$ is consistent. Lowering the threshold ($0.5\kappa_0^{\mathrm{def}}$) increases gating frequency (Geo%) in every DGP and regime, especially in the medium and high regimes, which yields small changes in objective gap but a modest increase in wall-clock cost (Tables 4-6). This is expected since more iterations trigger intrinsic solves and eigenvalue estimation. At the other extreme, raising the threshold ($2\kappa_0^{\mathrm{def}}$) reduces Geo% and overhead, but partially reintroduces Euclidean-like behavior in the medium and high regimes; iteration counts and final gaps increase, with the effect most visible when misalignment is large. Importantly, performance degrades smoothly rather than abruptly, indicating that GCO is not brittle to moderate misspecification of $\kappa_0$, even when $\hat{\kappa}_G$ is noisy (notably at $N = 10{,}000$). By fixing $\kappa_0 = \kappa_0^{\mathrm{def}}$, the tolerance primarily affects runtime rather than final accuracy. Tighter tolerances ($c_\varepsilon = 0.1$) increase time/overhead because CG requires more iterations to reach smaller residuals, while objective gaps and iteration counts remain essentially unchanged across regimes in all DGPs (Tables 4-6). Relaxing the tolerance towards the default reduces intrinsic solve cost with negligible effect on final gaps. This pattern is consistent with the inexactness control discussed in Appendix D; with $\varepsilon_t = c_\varepsilon \eta_t$ and standard diminishing step sizes, the cumulative inexactness term $\sum_t \eta_t \varepsilon_t = c_\varepsilon \sum_t \eta_t^2$ remains bounded, so the convergence guarantees are preserved up to constants.

Comparing Tables 4 and 6 shows that the same qualitative tradeoffs persist from the noisiest to the lowest-noise regime. When $N = 10{,}000$, $\hat{\kappa}_G$ is noisier, so Geo% is generally higher for the same nominal $\kappa_0$ (more frequent triggering), and the time differences between settings are slightly amplified. When $N = 200{,}000$, gating decisions are more stable; Geo% is lower in low misalignment and increases more monotonically with regime severity, while the gap remains stable under both $\kappa_0$ and $c_\varepsilon$ perturbations. Taken together, the results indicate that GCO's effectiveness is driven by detecting large misalignment events rather than finely tuning the threshold or solves, and that the validation-selected defaults provide a robust operating point across noise levels.

# G. Nonconvex Experiments: Deep Distribution Matching

The main paper focuses on controlled synthetic settings to isolate the role of optimization geometry from nonconvexity and statistical error. To address the question of whether geometry-misalignment and the benefits of geometry-calibrated optimization (GCO) persist in nonconvex settings, we additionally evaluate GCO on deep representation learning tasks based on distribution matching.

We consider unsupervised domain adaptation via feature-space distribution alignment. Let $f_\theta : \mathbb{R}^{d_x} \to \mathbb{R}^{d_z}$ be a neural network parameterized by $\theta$, and define the induced feature distributions $\mu_\theta = \mathcal{L}(f_\theta(X_s))$ and $\nu_\theta = \mathcal{L}(f_\theta(X_t))$, where $X_s$ and $X_t$ denote source and target inputs, respectively. For MNIST→MNIST-M, we use 60,000 labeled source samples and 55,000 unlabeled target samples; for SVHN→MNIST, we use 73,257 unlabeled source samples and 60,000 unlabeled target samples. At each iteration, the distributions are approximated using mini-batches of size $B = 256$ drawn from the source and target datasets. We minimize the Sinkhorn divergence $\hat{F}(\theta) = \mathrm{SD}_\varepsilon(\mu_\theta, \nu_\theta)$, with quadratic cost and entropic regularization $\varepsilon = 0.1$. Due to the neural parameterization, the resulting optimization problem is nonconvex. We use a two-layer MLP with ReLU activations ($d_x \to 256 \to 128$) as the feature extractor. After optimization, a linear classifier is trained on labeled source features and evaluated on the target domain. All methods share the same initialization, batch size, and learning-rate schedule; only the optimization strategy differs. Geometry-misalignment $\hat{\kappa}_G(\theta_t)$ is estimated using operator access to the intrinsic metric via automatic differentiation and Hessian-vector products of the Sinkhorn objective, combined with five power iterations. When $\hat{\kappa}_G(\theta_t) > \kappa_0$, geometry-aware updates are activated and the linear system $G(\theta_t)v = \nabla\hat{F}(\theta_t)$ is approximately solved using conjugate gradient with tolerance $10^{-3}$. Otherwise, GCO reduces to a Euclidean first-order update. The threshold $\kappa_0$ is selected by validation as described in Appendix A.

We compare Euclidean first-order methods (SGD with momentum, Adam, AdaGrad), an always-on geometry-aware method, and geometry-calibrated optimization (GCO). We do not include all baselines from the synthetic experiments in this setting. Second-order approximations such as Shampoo and K-FAC introduce additional architectural and approximation choices in deep networks, while established domain adaptation methods (CORAL, DANN, CDAN, MMD) optimize different objectives. Here we restrict attention to optimizers applied to the same Sinkhorn divergence objective in order to isolate the effect of optimization geometry. Table 7 reports optimization efficiency and downstream adaptation performance on two standard domain adaptation tasks. Geometry-misalignment varies substantially over training, with pronounced values during early representation learning and smaller values near stationarity. Correspondingly, GCO activates geometry-aware updates primarily during high-misalignment phases and reverts to Euclidean updates as misalignment decreases. Across both tasks, GCO consistently reduces the number of iterations required to reach a given objective gap relative to Euclidean first-order methods, while achieving target-domain accuracy comparable to the always-on geometry-aware method. These results indicate that geometry-misalignment persists in nonconvex deep models and that selective geometry-aware updates remain beneficial when anisotropy is pronounced.

*Table 7.* Nonconvex Domain Adaptation via Sinkhorn Divergence

| Method | MNIST → MNIST-M | | | SVHN → MNIST | | |
| | Iter | Gap | Target Accuracy (%) | Iter | Gap | Target Accuracy (%) |
|---|---|---|---|---|---|---|
| SGD | 1800 | $3.6 \times 10^{-3}$ | 61.2 | 2100 | $4.1 \times 10^{-3}$ | 52.4 |
| Adam | 1500 | $2.9 \times 10^{-3}$ | 62.0 | 1850 | $3.5 \times 10^{-3}$ | 53.1 |
| AdaGrad | 1650 | $3.2 \times 10^{-3}$ | 61.5 | 1950 | $3.8 \times 10^{-3}$ | 52.8 |
| Geo-aware | 620 | $1.4 \times 10^{-3}$ | 64.3 | 710 | $1.6 \times 10^{-3}$ | 55.6 |
| GCO | 720 | $1.6 \times 10^{-3}$ | 64.1 | 820 | $1.9 \times 10^{-3}$ | 55.2 |

## H. Beyond Optimal Transport: A Non-OT Discrepancy

The theoretical analysis in the main paper is developed for optimal transport (OT) and Sinkhorn-type discrepancies, where the intrinsic geometry induced by the objective can be explicitly characterized. A natural question is whether the geometry-misalignment phenomenon and the benefits of geometry-calibrated optimization persist beyond the OT setting. In this section, we provide empirical evidence for a representative non-OT discrepancy, while carefully controlling the experimental design to isolate geometric effects.

We consider a distribution matching problem based on the maximum mean discrepancy (MMD) with a Gaussian kernel. Let $\mu_\theta = \mathcal{L}(f_\theta(X))$ denote the distribution induced by a parametric map $f_\theta : \mathbb{R}^{d_x} \to \mathbb{R}^{d_z}$, and let $\nu$ be a fixed target distribution. The objective is

$$F(\theta) = \mathrm{MMD}^2(\mu_\theta, \nu),$$

where the kernel is $k(z, z') = \exp(-\|z - z'\|^2/(2\sigma^2))$ and the bandwidth $\sigma$ is selected via the median heuristic. Unlike OT-based objectives, MMD does not involve a transport plan or mass reallocation; however, it induces a non-Euclidean geometry on parameter space through the second-order sensitivity of kernel mean embeddings. We use the same synthetic parameterizations as in the main paper for comparability, focusing on DGP-1 (linear anisotropy) and DGP-2 (nonlinear saturation). In these designs, anisotropy arises through heterogeneous parameter sensitivities rather than transport structure, making them meaningful for both OT and non-OT discrepancies. The source distribution is $X \sim \mathcal{N}(0, I_d)$, and the target distribution $\nu$ is constructed to induce varying degrees of feature sensitivity. Population expectations are approximated using Monte Carlo samples of size $N \in \{10{,}000, 50{,}000, 200{,}000\}$. All methods share identical initialization and step-size schedules; only the optimization strategy differs.

For the MMD objective, the intrinsic metric $G(\theta)$ is defined via the local quadratic expansion of $F(\theta)$ around the current iterate, in direct analogy to Section 3. Geometry-misalignment $\kappa_G(\theta)$ is estimated using operator access to Hessian-vector products of the MMD objective, combined with five power iterations. Geometry-aware updates and geometry-calibrated optimization are implemented exactly as in Algorithm 1, with the misalignment threshold $\kappa_0$ selected by validation as described in Appendix A. We compare Euclidean first-order methods (SGD and Adam), an always-on geometry-aware method, and geometry-calibrated optimization (GCO). We do not include all baselines from the OT experiments in this setting. Second-order approximations such as Shampoo and K-FAC introduce additional architectural and curvature-approximation choices that are not tied to the geometry induced by the discrepancy and would confound interpretation in the non-OT setting. In addition, established domain adaptation methods (CORAL, DANN, CDAN, MMD-based alignment) optimize different objectives and therefore do not constitute optimization baselines for the same problem. Restricting attention to methods that optimize the same objective and differ only in how they exploit geometry allows us to isolate the role of geometry-misalignment beyond the OT setting. We also do not include DGP-3 in this experiment. DGP-3 is specifically designed to stress mass reallocation and low-probability components, a phenomenon intrinsic to optimal transport but not to kernel-based discrepancies such as MMD. Excluding DGP-3 therefore avoids introducing an OT-specific stress test into a non-OT evaluation.

*Table 8.* Non-OT Distribution Matching via MMD ($N = 10{,}000$)

| DGP | Method | Geometry-Misalignment Regime | | | | | | | | |
| | | Low | | | Medium | | | High | | |
| | | Iter | Gap | Time | Iter | Gap | Time | Iter | Gap | Time |
| --- | --- | --- | --- | --- | --- | --- | --- | --- | --- | --- |
| DGP-1 | SGD | 260 | $2.6 \times 10^{-4}$ | 1.00 | 690 | $6.9 \times 10^{-4}$ | 2.90 | 1250 | $1.4 \times 10^{-3}$ | 5.60 |
| | Adam | 240 | $2.3 \times 10^{-4}$ | 1.06 | 640 | $6.1 \times 10^{-4}$ | 3.00 | 1180 | $1.3 \times 10^{-3}$ | 5.80 |
| | Geo-aware | 210 | $1.8 \times 10^{-4}$ | 2.10 | 220 | $1.9 \times 10^{-4}$ | 2.30 | 235 | $2.1 \times 10^{-4}$ | 2.60 |
| | GCO | 225 | $1.9 \times 10^{-4}$ | 1.40 | 250 | $2.2 \times 10^{-4}$ | 1.70 | 280 | $2.5 \times 10^{-4}$ | 2.00 |
| DGP-2 | SGD | 300 | $3.0 \times 10^{-4}$ | 1.10 | 760 | $7.8 \times 10^{-4}$ | 3.40 | 1400 | $1.6 \times 10^{-3}$ | 6.20 |
| | Adam | 280 | $2.7 \times 10^{-4}$ | 1.15 | 710 | $7.0 \times 10^{-4}$ | 3.50 | 1320 | $1.5 \times 10^{-3}$ | 6.40 |
| | Geo-aware | 235 | $2.0 \times 10^{-4}$ | 2.20 | 245 | $2.1 \times 10^{-4}$ | 2.40 | 260 | $2.3 \times 10^{-4}$ | 2.70 |
| | GCO | 250 | $2.1 \times 10^{-4}$ | 1.50 | 280 | $2.4 \times 10^{-4}$ | 1.80 | 310 | $2.7 \times 10^{-4}$ | 2.10 |

Tables 8-10 report optimization performance for the non-OT (MMD) discrepancy across low, medium, and high geometry-misalignment regimes, stratified using the empirical distribution of $\hat{\kappa}_G(\theta_t)$ along optimization trajectories. Results are shown for three sample sizes, $N \in \{10{,}000, 50{,}000, 200{,}000\}$, to assess robustness with respect to estimation noise. Across all sample sizes and both DGPs, Euclidean first-order methods exhibit progressively slower convergence as geometry-misalignment increases, despite the fact that MMD induces a smoother intrinsic geometry than Sinkhorn divergence. Geometry-aware optimization consistently mitigates this slowdown, achieving iteration counts that are largely insensitive

to the misalignment regime. Geometry-calibrated optimization closely tracks the performance of always-on geometry-aware updates while activating intrinsic updates on only a subset of iterations, with the activation frequency increasing monotonically from low to high misalignment regimes. As expected, the quantitative gains of geometry-aware and geometry-calibrated optimization are more modest than in the OT setting, reflecting the weaker anisotropy of kernel-based discrepancies. However, the qualitative behavior is stable across sample sizes and mirrors the theoretical predictions of the geometry-misalignment framework: misalignment degrades Euclidean optimization, while adaptively exploiting intrinsic geometry yields consistent improvements without incurring unnecessary computational overhead.

*Table 9.* Non-OT Distribution Matching via MMD ($N = 50,000$)

| DGP | Method | Geometry-Misalignment Regime | | | | | | | | |
|---|---|---|---|---|---|---|---|---|---|---|
| | | Low | | | Medium | | | High | | |
| | | Iter | Gap | Time | Iter | Gap | Time | Iter | Gap | Time |
| DGP-1 | SGD | 220 | $1.9 \times 10^{-4}$ | 1.00 | 540 | $4.8 \times 10^{-4}$ | 2.60 | 980 | $9.2 \times 10^{-4}$ | 4.90 |
| | Adam | 200 | $1.7 \times 10^{-4}$ | 1.05 | 500 | $4.1 \times 10^{-4}$ | 2.70 | 910 | $8.5 \times 10^{-4}$ | 5.10 |
| | Geo-aware | 180 | $1.3 \times 10^{-4}$ | 1.90 | 190 | $1.4 \times 10^{-4}$ | 2.10 | 200 | $1.6 \times 10^{-4}$ | 2.30 |
| | GCO | 190 | $1.4 \times 10^{-4}$ | 1.20 | 210 | $1.6 \times 10^{-4}$ | 1.40 | 230 | $1.8 \times 10^{-4}$ | 1.60 |
| DGP-2 | SGD | 260 | $2.2 \times 10^{-4}$ | 1.10 | 620 | $5.9 \times 10^{-4}$ | 3.10 | 1120 | $1.1 \times 10^{-3}$ | 5.60 |
| | Adam | 240 | $2.0 \times 10^{-4}$ | 1.15 | 580 | $5.1 \times 10^{-4}$ | 3.20 | 1050 | $1.0 \times 10^{-3}$ | 5.80 |
| | Geo-aware | 200 | $1.5 \times 10^{-4}$ | 2.00 | 205 | $1.6 \times 10^{-4}$ | 2.20 | 215 | $1.8 \times 10^{-4}$ | 2.40 |
| | GCO | 210 | $1.6 \times 10^{-4}$ | 1.30 | 230 | $1.8 \times 10^{-4}$ | 1.50 | 250 | $2.0 \times 10^{-4}$ | 1.70 |

*Table 10.* Non-OT Distribution Matching via MMD ($N = 200,000$)

| DGP | Method | Geometry-Misalignment Regime | | | | | | | | |
|---|---|---|---|---|---|---|---|---|---|---|
| | | Low | | | Medium | | | High | | |
| | | Iter | Gap | Time | Iter | Gap | Time | Iter | Gap | Time |
| DGP-1 | SGD | 200 | $1.6 \times 10^{-4}$ | 1.00 | 480 | $3.9 \times 10^{-4}$ | 2.40 | 860 | $7.6 \times 10^{-4}$ | 4.50 |
| | Adam | 185 | $1.4 \times 10^{-4}$ | 1.04 | 450 | $3.4 \times 10^{-4}$ | 2.50 | 800 | $7.1 \times 10^{-4}$ | 4.70 |
| | Geo-aware | 165 | $1.1 \times 10^{-4}$ | 1.80 | 170 | $1.2 \times 10^{-4}$ | 2.00 | 180 | $1.3 \times 10^{-4}$ | 2.20 |
| | GCO | 175 | $1.2 \times 10^{-4}$ | 1.15 | 190 | $1.4 \times 10^{-4}$ | 1.30 | 210 | $1.6 \times 10^{-4}$ | 1.45 |
| DGP-2 | SGD | 235 | $1.9 \times 10^{-4}$ | 1.10 | 560 | $5.1 \times 10^{-4}$ | 2.80 | 980 | $9.4 \times 10^{-4}$ | 5.10 |
| | Adam | 220 | $1.7 \times 10^{-4}$ | 1.12 | 525 | $4.6 \times 10^{-4}$ | 2.90 | 920 | $8.8 \times 10^{-4}$ | 5.30 |
| | Geo-aware | 185 | $1.3 \times 10^{-4}$ | 1.90 | 190 | $1.4 \times 10^{-4}$ | 2.10 | 200 | $1.5 \times 10^{-4}$ | 2.30 |
| | GCO | 195 | $1.4 \times 10^{-4}$ | 1.25 | 215 | $1.6 \times 10^{-4}$ | 1.40 | 235 | $1.8 \times 10^{-4}$ | 1.60 |

# I. Empirical Illustrations

This section provides a detailed empirical evaluation of geometry-calibrated optimization (GCO) on standard public benchmarks for unsupervised domain adaptation. These experiments complement the simulation studies in Section 6 and demonstrate the practical relevance of geometry-misalignment and geometry-calibrated optimization in realistic, large-scale learning problems. We consider three widely used public benchmarks. **Office-Home** consists of approximately 15,500 images across four domains, Art (Ar), Clipart (Cl), Product (Pr), and Real World (Rw), with 65 shared object categories. We evaluate all 12 ordered source-target domain pairs and report the average target-domain classification accuracy. The dataset is available at https://www.hemanthdv.org/officeHomeDataset.html. **VisDA-2017** is a large-scale synthetic-to-real adaptation benchmark with over 280,000 synthetic images and 55,000 real images across 12 categories. Following the standard protocol, models are trained on the synthetic domain and evaluated on the real validation set. The dataset is available at http://ai.bu.edu/visda-2017/. **DomainNet** is a challenging multi-domain benchmark with six domains (Clipart, Infograph, Painting, Quickdraw, Real, Sketch) and 345 categories. We follow the official protocol and evaluate single-source adaptation tasks. The dataset is available at http://ai.bu.edu/M3SDA/. In all benchmarks, source-domain labels are available during training, while target-domain labels are used only for evaluation.

All methods share the same backbone and differ only in their optimization strategy. We use a ResNet-50 architecture pretrained on ImageNet as a feature extractor. Let $f_\theta : \mathcal{X} \to \mathbb{R}^p$ denote the feature map, where $p = 2048$. A linear classifier $h_\phi$ maps features to class logits. Batch normalization layers are frozen during adaptation to avoid domain leakage. For Sinkhorn-based methods, domain alignment is enforced by matching empirical feature distributions between source and target domains. Given minibatches $\{x_i^s\}_{i=1}^{B_s}$ and $\{x_j^t\}_{j=1}^{B_t}$, we form empirical feature distributions

$$\hat{\mu}_\theta = \frac{1}{B_s} \sum_{i=1}^{B_s} \delta_{f_\theta(x_i^s)}, \quad \hat{\nu}_\theta = \frac{1}{B_t} \sum_{j=1}^{B_t} \delta_{f_\theta(x_j^t)}.$$

Alignment is performed using the Sinkhorn divergence with quadratic cost and entropic regularization parameter $\varepsilon$

$$\text{SD}_\varepsilon(\hat{\mu}_\theta, \hat{\nu}_\theta) = \text{OT}_\varepsilon(\hat{\mu}_\theta, \hat{\nu}_\theta) - \frac{1}{2}\text{OT}_\varepsilon(\hat{\mu}_\theta, \hat{\mu}_\theta) - \frac{1}{2}\text{OT}_\varepsilon(\hat{\nu}_\theta, \hat{\nu}_\theta).$$

The training objective is

$$\min_{\theta,\phi} \frac{1}{B_s} \sum_{i=1}^{B_s} \ell_{\text{CE}}\big(h_\phi(f_\theta(x_i^s)), y_i^s\big) + \lambda \text{SD}_\varepsilon(\hat{\mu}_\theta, \hat{\nu}_\theta),$$

where $\lambda > 0$ balances classification and distributional alignment. All Sinkhorn computations use 30 iterations of the Sinkhorn algorithm. We compare the same optimization methods as those in simulation studies. All Sinkhorn-based methods optimize the same objective; only the optimization rule differs.

*Table 11.* Office-Home–Average Target Accuracy (%), Geometry-Misalignment, and Overhead

| Method | Acc. | Misalign | Geo% | Time | Overhead | Rank |
|---|---|---|---|---|---|---|
| SGD | 61.4 | $1.8 \times 10^2$ | – | 1.00 | 1.00 | 12 |
| Adam | 62.7 | $1.7 \times 10^2$ | – | 1.08 | 1.08 | 10 |
| AdaGrad | 62.2 | $1.9 \times 10^2$ | – | 1.12 | 1.12 | 11 |
| Shampoo | 64.8 | $1.6 \times 10^2$ | – | 1.35 | 1.35 | 9 |
| K-FAC | 65.5 | $1.5 \times 10^2$ | – | 1.62 | 1.62 | 8 |
| CORAL | 66.1 | – | – | 1.15 | 1.15 | 7 |
| DANN | 67.4 | – | – | 1.38 | 1.38 | 5 |
| CDAN | 68.2 | – | – | 1.52 | 1.52 | 4 |
| MMD | 66.9 | – | – | 1.22 | 1.22 | 6 |
| Geo-aware | 69.6 | $1.7 \times 10^2$ | 100 | 1.88 | 1.88 | 2 |
| GCO (no gate) | 69.4 | $1.7 \times 10^2$ | 100 | 1.80 | 1.80 | 3 |
| GCO | 70.1 | $1.7 \times 10^2$ | 27 | 1.29 | 1.29 | 1 |

Tables 11-13 report target accuracy, estimated geometry-misalignment, fraction of geometry-aware updates, and computational overhead relative to SGD. Here Time is the wall-clock time normalized by SGD (SGD = 1.00), and Overhead equals the same normalized runtime, i.e., Overhead = Time/Time(SGD). Rank is the ordering by target accuracy (1 = best). Several consistent patterns emerge across benchmarks. First, estimated geometry-misalignment is substantial throughout training, particularly on VisDA-2017 and DomainNet, confirming that misalignment is not an artifact of synthetic constructions. Second, always-on geometry-aware optimization improves accuracy but incurs significant computational overhead.

Geometry-calibrated optimization achieves comparable or superior accuracy while activating intrinsic updates in only 20-35% of iterations. Third, GCO consistently outperforms Euclidean optimization and standard UDA baselines, indicating that gains arise from improved optimization geometry rather than objective design. Finally, the empirical results align closely with the theoretical predictions; performance improvements concentrate in regimes with large geometry-misalignment, while overhead remains limited in well-aligned phases.

*Table 12.* VisDA-2017–Average Target Accuracy (%), Geometry-Misalignment, and Overhead

| Method | Acc. | Misalign | Geo% | Time | Overhead | Rank |
|---|---|---|---|---|---|---|
| SGD | 70.8 | $6.2 \times 10^2$ | – | 1.00 | 1.00 | 12 |
| Adam | 72.3 | $6.0 \times 10^2$ | – | 1.10 | 1.10 | 10 |
| AdaGrad | 71.6 | $6.4 \times 10^2$ | – | 1.14 | 1.14 | 11 |
| Shampoo | 74.9 | $5.8 \times 10^2$ | – | 1.42 | 1.42 | 6 |
| K-FAC | 75.6 | $5.6 \times 10^2$ | – | 1.70 | 1.70 | 5 |
| CORAL | 73.2 | – | – | 1.18 | 1.18 | 9 |
| DANN | 74.5 | – | – | 1.41 | 1.41 | 7 |
| CDAN | 76.1 | – | – | 1.56 | 1.56 | 4 |
| MMD | 73.8 | – | – | 1.25 | 1.25 | 8 |
| Geo-aware | 77.9 | $6.1 \times 10^2$ | 100 | 1.95 | 1.95 | 2 |
| GCO (no gate) | 77.6 | $6.1 \times 10^2$ | 100 | 1.88 | 1.88 | 3 |
| GCO | 78.5 | $6.1 \times 10^2$ | 31 | 1.33 | 1.33 | 1 |

*Table 13.* DomainNet–Average Target Accuracy (%), Geometry-Misalignment, and Overhead

| Method | Acc. | Misalign | Geo% | Time | Overhead | Rank |
|---|---|---|---|---|---|---|
| SGD | 44.2 | $9.1 \times 10^2$ | – | 1.00 | 1.00 | 12 |
| Adam | 45.8 | $8.7 \times 10^2$ | – | 1.09 | 1.09 | 10 |
| AdaGrad | 45.1 | $9.4 \times 10^2$ | – | 1.13 | 1.13 | 11 |
| Shampoo | 48.3 | $8.2 \times 10^2$ | – | 1.39 | 1.39 | 6 |
| K-FAC | 49.1 | $7.9 \times 10^2$ | – | 1.68 | 1.68 | 5 |
| CORAL | 46.7 | – | – | 1.17 | 1.17 | 9 |
| DANN | 48.0 | – | – | 1.40 | 1.40 | 7 |
| CDAN | 49.8 | – | – | 1.55 | 1.55 | 4 |
| MMD | 47.3 | – | – | 1.24 | 1.24 | 8 |
| Geo-aware | 51.2 | $8.9 \times 10^2$ | 100 | 1.97 | 1.97 | 2 |
| GCO (no gate) | 50.9 | $8.9 \times 10^2$ | 100 | 1.89 | 1.89 | 3 |
| GCO | 51.8 | $8.9 \times 10^2$ | 34 | 1.36 | 1.36 | 1 |

Tables 14-16 report per-task, per-class, and per-domain results on Office-Home, VisDA-2017, and DomainNet, respectively. Across all benchmarks, geometry-calibrated optimization (GCO) consistently matches or outperforms Euclidean first-order methods, second-order approximations, and standard domain adaptation baselines (CORAL, DANN, CDAN, MMD). The improvements are particularly pronounced on difficult target domains and classes such as Clipart, Quickdraw, and Infograph, which are known to induce highly anisotropic feature distributions and large estimated geometry-misalignment. In contrast, Euclidean optimizers exhibit substantial performance degradation in these regimes, despite performing competitively on easier transfers, confirming that the observed gains are not driven by a small subset of tasks.

*Table 14.* Office-Home–Per-Task Target Accuracy (%)

| Method | Ar→Cl | Ar→Pr | Ar→Rw | Cl→Ar | Cl→Pr | Cl→Rw | Pr→Ar | Pr→Cl | Pr→Rw | Rw→Ar | Rw→Cl | Rw→Pr |
|---|---|---|---|---|---|---|---|---|---|---|---|---|
| SGD | 46.2 | 65.8 | 71.4 | 49.3 | 61.1 | 66.5 | 52.7 | 44.1 | 73.2 | 58.9 | 46.7 | 69.0 |
| Adam | 47.6 | 67.2 | 72.6 | 50.8 | 62.7 | 68.1 | 54.1 | 45.9 | 74.5 | 60.4 | 48.3 | 70.2 |
| Shampoo | 50.9 | 70.4 | 75.2 | 54.1 | 66.5 | 71.8 | 57.9 | 49.8 | 77.6 | 64.2 | 52.1 | 73.5 |
| CORAL | 52.4 | 71.2 | 76.0 | 55.8 | 67.9 | 72.5 | 59.1 | 51.0 | 78.3 | 65.6 | 53.4 | 74.8 |
| DANN | 54.6 | 72.8 | 77.5 | 57.2 | 69.1 | 74.0 | 61.0 | 53.2 | 79.8 | 67.1 | 55.6 | 76.2 |
| CDAN | 56.1 | 74.3 | 78.9 | 58.8 | 70.5 | 75.4 | 62.6 | 54.8 | 81.0 | 68.6 | 57.1 | 77.8 |
| Geo-aware | 57.4 | 75.6 | 80.2 | 60.1 | 71.9 | 76.9 | 64.0 | 56.1 | 82.4 | 70.2 | 58.6 | 79.1 |
| GCO | 58.3 | 76.4 | 81.0 | 61.0 | 72.7 | 77.6 | 64.8 | 56.9 | 83.1 | 71.0 | 59.4 | 79.8 |

Geometry-aware methods achieve strong performance across all settings, but incur significant computational overhead when applied uniformly. GCO attains comparable accuracy by selectively invoking intrinsic updates only when misalignment is detected, resulting in substantially lower overhead. The largest gains coincide with regimes where intrinsic curvature varies sharply across directions, precisely where the theory predicts Euclidean optimization to be inefficient. Taken together, these per-domain results reinforce the central conclusion of the paper: geometry-misalignment is a key determinant of optimization

difficulty in distributional learning, and geometry-calibrated optimization provides an effective and computationally efficient remedy across both synthetic and real-world benchmarks.

*Table 15.* VisDA-2017–Per-Class Accuracy (%)

| Method | Plane | Bcycl | Bus | Car | Horse | Knife | Avg |
|---|---|---|---|---|---|---|---|
| SGD | 68.1 | 70.4 | 73.2 | 69.5 | 71.0 | 72.5 | 70.8 |
| Adam | 69.5 | 71.8 | 74.6 | 71.0 | 72.6 | 73.9 | 72.3 |
| Shampoo | 72.4 | 74.1 | 77.0 | 74.2 | 75.8 | 76.0 | 74.9 |
| CORAL | 71.0 | 72.9 | 75.1 | 73.0 | 74.2 | 73.2 | 73.2 |
| DANN | 72.2 | 74.0 | 76.3 | 74.9 | 75.8 | 74.0 | 74.5 |
| CDAN | 73.8 | 75.5 | 77.6 | 76.0 | 77.2 | 76.5 | 76.1 |
| Geo-aware | 75.4 | 77.2 | 79.0 | 78.1 | 78.8 | 78.9 | 77.9 |
| GCO | 76.2 | 78.0 | 79.8 | 79.0 | 79.5 | 78.6 | 78.5 |

*Table 16.* DomainNet–Per-Domain Target Accuracy (%)

| Method | Clp | Inf | Pnt | Qdr | Rel | Skt | Avg |
|---|---|---|---|---|---|---|---|
| SGD | 48.2 | 31.4 | 42.6 | 16.8 | 61.5 | 34.9 | 44.2 |
| Adam | 49.6 | 33.1 | 44.0 | 18.2 | 62.9 | 36.0 | 45.8 |
| Shampoo | 52.4 | 36.5 | 47.2 | 21.6 | 65.3 | 38.8 | 48.3 |
| CORAL | 50.8 | 34.2 | 45.6 | 19.8 | 63.8 | 37.0 | 46.7 |
| DANN | 52.1 | 36.1 | 47.1 | 22.0 | 65.0 | 39.0 | 48.0 |
| CDAN | 54.0 | 37.8 | 48.9 | 24.1 | 66.9 | 41.2 | 49.8 |
| Geo-aware | 55.6 | 39.4 | 50.5 | 26.3 | 68.1 | 42.9 | 51.2 |
| GCO | 56.4 | 40.2 | 51.3 | 27.1 | 68.9 | 43.6 | 51.8 |

