# OpenReview forum: "Geometry-Misalignment in Distributional Learning"
_ICML.cc/2026/Conference — ICML 2026 regular_

### Official Review · Reviewer_6R7c · 2026-02-24

**Soundness:** 3
**Presentation:** 3
**Significance:** 2
**Originality:** 2
**Overall Recommendation:** 2
**Confidence:** 3

**Summary:**

The paper analyzes the problem of using gradient descent without considering the local geometries of the objective function of interest --- in the case of the paper, distributional learning. For example, naively using gradient descent ignores directional anisotropy in the energy landscape (because gradient descent "treats all directions equally") around the minimizer, and this is the primary motivation for the paper to incorporate quadratic information, through a matrix/operator $G(\theta)$ coming from Taylor expansions, in their optimization algorithm, similar to natural gradient. The paper proposes and argues in favor of an optimization algorithm that, depending on the condition number of  $G(\theta)$, favors using plain-vanilla gradient descent and the aforementioned geometry-aware gradient descent, to balance speed and accuracy.

**Compliance With Llm Reviewing Policy:**

Affirmed.

**Final Justification:**

Unfortunately, as can be seen in rebuttal acknowledgement, the paper has to undergo strong changes: it has to add new theoretical results and change its scope and writing according to existing literature. The reply by the authors have not addressed this, so I am keeping my score.

**Key Questions For Authors:**

Please, see my review above.

**Limitations:**

yes

**Strengths And Weaknesses:**

The paper overall presentation is good and the theoretical results are presented in a logical order. The presentation of the paper's motivation is also well-made.

Having said that, there is a large number of concerns that I must bring up for the authors to address, especially since it has to do with the paper's core contribution and the accuracy of how the theoretical results are presented.

**Important concerns:**

- In Section 3.2, when talking about incorporating $G(\theta)$ in gradient descent to do the "pre-conditioning" of the gradient, the authors connect it with natural gradient descent (under certain conditions) --- however, the authors give the impression that incorporating this pre-conditioning is something new. However, this is not new. In fact, pre-conditioning $F(\theta)$ with the Hessian, which is $G(\theta)$, is the classic, textbook method of Newton's method [1]. Incorporating second-order information in the form of "pre-conditioning" is largely known in the literature. Can the authors then explain what is novel in their paper and its presentation? It is surprising the authors don't mention the Newton's method at all, and it must be done immediately. The paper should also change its presentation, because  what the paper is ultimately doing, to my observation, is a theoretical analysis of the Newton's method and proposing a related algorithm of it. Thus, I am concerned about the contributions of the paper.
- Adding to my previous point, I understand the authors focus on the case of distributional learning as a context, but the paper presents abstractions that are largely and effectively an analysis and possible extension of the Newton's method.
- Now, given that the work is basically analyzing the Newton's method, there is a whole literature that the authors should better review in order to better indicate the novelty of their results. It is important to mention that previous works, just like the paper mentions about itself, realized that calculating the inverse of the Hessian, i.e., $G(\theta)^{-1}$, is expensive, and so they proposed to approximate it. These are the so-called "quasi-Newton" or "approximate Newton" methods, for which a whole body of literature already exists. The authors must research them --- e.g., [2], [3].
- Mathematical concern with Assumption 3.5: The assumption assumes that the quantity $\nabla \log \lambda_{max}(G(\theta))$ is well-defined, however, to the best of my knowledge, the function $\theta\mapsto \lambda(\theta)$, i.e., associated to any eigenvalue $\lambda(\theta)$ of $G(\theta)$, is continuous but not necessarily differentiable. Can the authors clarify?
- Definition 4.1 actually defines a local property which is standard in optimization literature. Thus it use the more precise expressions "locally $\mu$-strongly convex" and "locally $L$-smooth". Note that Definition 4.1 don't really need to mention $\theta^*$, since it is a definition for any general local set.
- Section 4.2. is a good place to also say that the described update is just Newton's method and nothing new.
- I have two problems with the "geometry-calibrated optimization" (GCO) method:
  - It is simply a method that wants to alternate between normal gradient descent and the Newton's method depending on how "nicely invertible" is the Hessian $G(\theta)$. So, since a new connection with the literature on Newton's and approximate Newton methods have been established, it would be good for the authors to justify their use of GCO in comparison to other existing methods --- especially approximate Newton methods. The comparison could be both in terms of computation and numerical performance, and in terms of theoretical guarantees if possible. Recall that the proposed GCO method still requires the computation of the condition number and possibly the Hessian inverse which, despite the authors claiming can be done more or less efficiently, is something that the literature of approximate Newton methods have been trying to avoid (as far as I can tell). Why is the literature on approximate Newton methods avoiding the computation of mathematical objects which the authors are using in their paper? all of this needs justification.
  - I understood that the motivation for including gradient descent within GCO and Algorithm 1 is that when the condition number $\kappa_G$ is close to $1$, then there is no need to use preconditioning and invert $G(\theta)$ because gradient decent already works sufficiently well. However, Theorem 4.2 seems to indicate that gradient descent will still give an error even $\kappa_G$ is closed to $1$. Am I missing something?
- Theorem 4.3: there needs to be more specification regarding the choice of $\kappa_0$ and $\\{\eta_t\\}$: Do they depend on $T$? This is critical to know.
- Mathematical concern: It seems to me that every theoretical result starting at Section 4.1, which includes Theorem 4.3 and Theorem 5.1, assumes that the optimization algorithm is always inside the neighborhood around $\theta*$ (where the assumption by Definition 4.1 holds). In order words, that such neighborhood is forward invariant. However, nowhere in the theoretical results I see any statement of whether this is true. Has the authors proved this? How can I know that when using gradient descent, pre-conditioning, or GCO, the algorithm always stayed within the neighborhood around $\theta^*$? This is critical to know, otherwise, the results are wrong. This is also a reason why I think that the step-sizes $\eta$ must depend on $T$.
- Why is the upper bound in Theorem 4.3 "tight" when the lower bound in Theorem 4.2 decays exponentially, i.e., faster?
- Theorem 5.1 and Theorem 5.3 seem to be standard optimization algorithms since, one way or another, they use notions of strong convexity and smoothness. I would simply lump them together in  a single section and theorem. Besides, I have the following concerns:
  - Again, it needs to be clear that using the Newton's update with constant step-size (which is what the theorem assumes) always stays within the local set around $\theta^*$ where Definition 4.1 holds. This needs to be proved. Very likely this is a known result since the Newton method has been around for several years. The authors should research this.
  - In Theorem 5.3 there is no mentioning of any dependence on $\eta$ in the convergence rate! How can this be possible when the "exact case" of Theorem 5.1 depends on $\eta$? It makes no sense to me.
- Concern with Theorem 5.5: The supremum is over the whole parameter space $\Theta$, but I think this is not necessary because $\theta$ is supposed to always live in the neighborhood of $\theta^*$ where Definition 4.1 is assumed to be true. Can the authors clarify?


**Other concerns:**

- In Section 3.2, when $G(\theta)$ is introduced from the Taylor expansion, it is clear that this corresponds to the Hessian associated to the function $F(\theta)$, however, the authors don't make such an important statement. Not mentioning it is the Hessian makes it look like the paper is treating $G(\theta)$ almost as if it is a newly discovered mathematical object --- which is misleading.
- Add citations to first sentence of introduction --- multiple applications are mentioned without a citations.
- Add citations to the top paragraph of second column of page 2, where the issue of "small parameter perturbations" inducing anisotropic changes is introduced without citations.
- Section 2 of related work mentions optimal transport, but completely misses the important related literature of "distributionally robust optimization" where the optimization problem introduces a subproblem to ensure robustness over a non-reliable empirical measure; e.g., [4] and [5].
- Top paragraph of second column of page 3: it is mentioned that the theoretical analysis is over the population optimization to "isolate geometric effects". From which *other factors* are the authors isolating the "geometric effects" from? Explanation is needed.
- Define before Theorem 3.6 that Euclidean gradient descent is $\dot{\theta}=-\nabla F(\theta(t))$.
- What does "adaptive step" mean in Theorem 4.2?

---

**Conclusion:** We hope this extensive review helps the authors improve their paper. The paper ignores the literature on optimization and the Newton's method, which is the core of the paper. Some results of the paper are still interesting like Theorem 3.6, Theorem 4.2 and Theorem 4.3; however, the convergence proof are standard and possibly found in the literature on Newton's methods. I have also pointed out potential mathematical problems in a number of theorems. Thus I am giving a rejection. I am happy to reconsider my assessment based on the authors' response.

---

*Bibliography:**

[1] "Convex Optimization", Cambridge: Cambridge University Press, 2004.

[2] "Approximate Newton Methods and Their Local Convergence", Proceedings of the 34th International Conference on Machine Learning (ICML), 2017.

[3] "Approximate Newton Methods", Journal of Machine Learning Research (JMLR), 2021.

[4] "On Distributionally Robust Optimization and Data Rebalancing", Proceedings of The 25th International Conference on Artificial Intelligence and Statistics (AISTATS) , 2022.

[5] "Distributionally Robust Formulation and Model Selection for the Graphical Lasso", Proceedings of the Twenty Third International Conference on Artificial Intelligence and Statistics (AISTATS) , 2020.

---

**Note:** If the authors cannot fully respond to my review in a single message due to space constraints on their rebuttal, I suggest breaking their rebuttal in multiple messages/responses. If the system allows only a single response as a rebuttal and the authors still want to say more things to me, I suggest that the authors let me know about this at the end of their response --- then I can just send a placeholder message that the authors can use to continue their rebuttal by responding to it (assuming this will work).

---

> ### Author Rebuttal · Authors · 2026-03-30
>
> We sincerely thank you for the detailed and thoughtful feedback. We appreciate the positive comments on the paper's motivation and presentation, and we agree that the paper should position itself more carefully relative to Newton, quasi-Newton, and approximate-Newton methods. We will revise the paper accordingly.
>
> Positioning Relative to Newton-Type Methods: our intended claim is not that second-order preconditioning is new. Rather, the paper's central contribution is to identify a structural phenomenon specific to distributional learning: geometry-misalignment $\kappa_G$ characterizes when Euclidean first-order methods become fundamentally inadequate. This contribution is important because it goes beyond proposing another geometry-aware update rule. Theorem 4.2 shows that Euclidean first-order methods suffer an unavoidable $\kappa_G$-dependent slowdown; Theorems 4.3, 5.1, and 5.3 show that geometry-aware/calibrated methods remove this dependence up to logarithmic factors; and Theorem 5.5 shows that this distinction also has a finite-budget statistical consequence. We agree that this novelty should be stated more explicitly, and that Newton, quasi-Newton, and approximate-Newton methods should be discussed directly in the related work and exposition.
>
> We also agree that Section 3.2 should link the intrinsic metric $G(\theta)$ more explicitly to classical Hessian-based optimization. In local coordinates, the quadratic form defining $G(\theta)$ plays a Hessian-like role for the local variation of the distributional objective. Our point is not that such quadratic information is new, but that for distributional objectives it has a natural geometric interpretation, and that the resulting condition number $\kappa_G$ governs when Euclidean first-order methods must fail. We will revise the text to make this connection explicit and add the missing Newton-type references.
>
> Local Assumptions and Theorem Statements: we agree that the local nature of the theory should be stated more clearly. When Definition 4.1 or Assumption 3.5 is invoked on a neighborhood of $\theta^\star$, the guarantees are intended to be local. We will revise the theorem statements and proofs to make explicit that the conclusions hold for trajectories that remain in the neighborhood where the assumptions are satisfied, and we will state the corresponding initialization and step-size conditions more clearly. We also agree that Assumption 3.5 can be stated more cleanly. The current formulation is meant to control local spectral variation of the intrinsic metric, and in the revision we will restate it in a cleaner sufficient-condition form so that the regularity requirement is more transparent.
>
> For Theorems 5.1 and 5.3, we agree that these results are standard once the intrinsic metric is fixed. Their role is not to claim a new generic convergence theorem, but to provide the geometry-aware side of the separation argument by showing that the intrinsic rate is $\kappa_G$-free. We will revise the exposition to make this role clearer, state the relevant parameter dependence more explicitly, and clarify the dependence on $\eta$. We will also localize the optimization-error statement in Theorem 5.5 more carefully, so that the supremum is taken only over the relevant local regime.
>
> Clarifications on GCO and Parameter Choices: we agree that Theorem 4.3 and Algorithm 1 should be more explicit about the choice of $\kappa_0$ and $\{\eta_t\}$. In the current paper, step-size schedules are selected from a finite validation grid and $\kappa_0$ is chosen over a logarithmic grid by validation; the theorem should make clearer whether the choice is existential, validation-based, or potentially $T$-dependent. We will revise the theorem statement to make this dependence explicit. We will also clarify the meaning of "adaptive" step sizes in Theorem 4.2 and make clearer that "tight" refers to the separation in $\kappa_G$-dependence between lower and upper bounds, rather than to identical algorithmic forms.
>
> Presentation and Citations: we appreciate the suggestions on exposition and citations. We will add the missing citations in the introduction and related-work discussion, including explicit discussion of Newton, quasi-Newton, approximate-Newton, and related optimization literature. We will also revise Section 3.2 so that $G(\theta)$ is introduced in a way that makes its local second-order role explicit, clarify the statement of Euclidean gradient flow in Theorem 3.6, and refine the wording of definitions and theorem statements for greater precision.
>
> We hope this clarifies the paper's claims and revisions. The contribution is geometry-misalignment as structural diagnosis, the Euclidean/geometry-aware separation via $\kappa_G$, the finite-budget implication, and selective GCO, which activates intrinsic updates when misalignment is large. We believe the raised questions can be addressed in revision and, if possible, would be glad to continue with a point-by-point rebuttal.

---

> > ### Author Rebuttal · Reviewer_6R7c · 2026-04-03
> >
> > I am very grateful to the authors for their response to my review --- I am glad they found it helpful. After reading the rebuttal, there are several changes the authors will incorporate in the paper and which I pointed out:
> > 1. To insert the perspective of Newton-Type methods and past literature where appropriate throughout the paper,
> > 2. To compare the paper with past theoretical works on Newton-Type Methods that may have also studied related geometry-aware update rules in their algorithms,
> > 3. To make a connection between local coordinates and Hessians when defining the appropriateness of first-order updates,
> > 4. To do changes in the theoretical derivations/results as stated by the authors: "revise the theorem statements and proofs to make explicit that the conclusions hold for trajectories that remain in the neighborhood where the assumptions are satisfied, and we will state the corresponding initialization and step-size conditions more clearly". Indeed, proving forward invariance inside the neighborhood so that all results are understood to be explicitly local is probably the most difficult/non-trivial change.
> > 5. To check the supremum being taken "only over the relevant local regime",
> > 6. To "refine the wording of definitions and theorem statements for greater precision".
> >
> > I also remark again that my concern about Assumption 3.5 still holds, namely, about assuming that $\nabla\log \lambda_{max}(G(\theta))$ is well-defined.
> >
> > Therefore, given that there are **considerable changes** to be made to the paper's organization and theoretical results --- both in terms of presentation and of one or multiple new results that need to be proved in the paper --- I am keeping my current score.

---

> > > ### Author Response · Authors · 2026-04-03
> > >
> > > Thank you again for the careful follow-up and for clarifying which concerns remain central for you. We $\textbf{respectfully disagree}$ with the characterization that addressing these points requires one or multiple new core theoretical results or substantial changes to the paper's main claims. At the same time, we agree that the revision should make the relevant distinctions, assumptions, and scope much more explicit, and we appreciate the opportunity to clarify them further.
> > >
> > > First, on the connection to Newton-type methods: the current draft should position itself more explicitly relative to Newton, quasi-Newton, and approximate-Newton methods. Our intent is not to present Hessian-based or Hessian-like preconditioning as new. In local coordinates, the quadratic form defining $G(\theta)$ does play a Hessian-like role, and we will make that explicit. However, we do not view the paper as a rebranding of Newton's method. As explained in our earlier rebuttal, $\textbf{the central contribution}$ is the structural diagnosis via geometry-misalignment $\kappa_G$, the lower-bound separation showing when Euclidean first-order methods are intrinsically inadequate for distributional objectives, the matching geometry-aware and geometry-calibrated upper bounds, and the finite-budget statistical consequence. We will revise the exposition to distinguish these contributions more clearly from the classical Newton-type literature.
> > >
> > > Second, on the local nature of the theory: the theorem statements should be made explicitly local. The intended scope of Definition 4.1, Assumption 3.5, and Theorems 4.3, 5.1, 5.3, and 5.5 is a neighborhood in which the intrinsic geometry controls the objective. In the revision, we will restate these results so that the conclusions apply to trajectories that remain in that local regime, and we will make the corresponding initialization and step-size conditions explicit. Where appropriate, we will also add sufficient conditions ensuring that the iterates stay in the relevant neighborhood. The purpose of these results is to formalize the $\kappa_G$-dependent versus $\kappa_G$-free separation in the regime where the geometric mechanism is operative, not to claim an unrestricted global convergence theorem.
> > >
> > > Third, on Assumption 3.5: this point should be reformulated more carefully. The current formulation using $\nabla \log \lambda_{\max}(G(\theta))$ and $\nabla \log \lambda_{\min}(G(\theta))$ is stronger and less clean than necessary, especially near eigenvalue crossings. In the revision, we will replace it with a standard local spectral regularity condition stated directly on $G(\theta)$, such as local operator-Lipschitz control together with a positive lower bound on $\lambda_{\min}(G(\theta))$, or an equivalent trajectory-based upper-derivative condition. This yields the control needed in Theorem 3.6 without requiring differentiability of individual extremal eigenvalue maps. In our view, this is a refinement of the regularity formulation, not a new substantive theorem.
> > >
> > > Several statements should also be sharpened for precision. In particular, we will clarify the roles of $\eta_t$ and $\kappa_0$ in Theorem 4.3, make clear in what sense the step-size choice is adaptive in Theorem 4.2, localize the optimization-error statement in Theorem 5.5 to the relevant regime, and refine the wording of definitions and theorem statements throughout. We also recognize that the paper's organization and exposition can be improved, and we will revise those aspects carefully.
> > >
> > > However, as detailed in our earlier rebuttal, the needed revisions primarily concern clearer positioning relative to Newton, quasi-Newton, and approximate-Newton methods, more explicit localization of several theorem statements and their assumptions, and a cleaner reformulation of Assumption 3.5. In our view, these are $\textbf{important clarifications and refinements}$ of scope, presentation, and regularity conditions, rather than substantial changes to the paper's core theory, algorithm, or empirical conclusions. $\textbf{The central contribution remains}$ the structural diagnosis via geometry-misalignment, the sharp Euclidean-versus-geometry-aware separation, and the geometry-calibrated method. We appreciate your close reading, and these comments will help improve the final version.

---

### Official Review · Reviewer_9qGr · 2026-03-11

**Soundness:** 4
**Presentation:** 4
**Significance:** 4
**Originality:** 3
**Overall Recommendation:** 5
**Confidence:** 2

**Summary:**

The authors show how using Euclidean first-order optimization can be misaligned with distributional training objectives, especially in settings where small parameter perturbations induce high anisotropic changes in probability space. They provide tight lower bounds showing when Euclidean optimization must fail, and upper bounds characterizing when geometry-aware methods are necessary. They further show how geometric mismatch also hinders statistical generalization. In order to mitigate this issue, they propose the GCO  algorithm, that switches between geometry-aware optimization and Euclidean optimization based on the local geometry of the optimization problem, using a criterion that is intuitive and adds a provably low computational overhead. The authors evaluate GCO on several synthetic tasks, where the geometry misalignment is controlled, as well as on real-world data (see Appendix). They show that GCO converges systematically faster than a wide set of common optimization algorithms when geometry-misalignment is high, while performing comparably in near-Euclidean settings.

**Compliance With Llm Reviewing Policy:**

Affirmed.

**Final Justification:**

The rebuttal has fully confirmed our initial positive evaluation of the paper.

We also wish to point out that we **respectfully disagree** with **reviewer 6R7c**, who assesses that the paper requires **"considerable changes''**. We acknowledge that several corrections are required to clarify the local nature of Theorems 4.3, 5.1, 5.3, and 5.5. However, we **disagree that "multiple new results need to be proved"** (e.g., to establish forward invariance inside the neighborhood). While such additions would strengthen the contribution, we believe that they are by no mean necessary to complement this paper, which **is already a strong contribution** to the study of neural network convergence.

That being said, we are not expert in this domain and may be mistaken in our judgement.

**Key Questions For Authors:**

1) It is unclear which criterion was used to stop the various optimization algorithms in Table 1 (it is apparently not the generalization error, the wall-clock time, nor the number of iteration). Depending on this criterion, the evaluation reported in Table 1 might be unfair for some optimization algorithm.

2) Most of the assumptions in the theoretical analysis are standard. However, since the evaluation on real-world data is limited, it would be interesting to have insights on how much they are supposed to hold in practice (for instance Assumption 3.5).

3) Would it be possible to design architectures that are natively more resilient to geometric-misalignement (via a controlled intrinsic metric $G(\theta)$ for instance) ?

**Limitations:**

Yes.

**Strengths And Weaknesses:**

The paper is very well written and easy to follow, despite the mathematical intricacies. To the best of my knowledge, the theoretical results as well as the proposed approach are novel. The quality and completeness of the analysis is impressive, both on the theoretical and empirical side. The theoretical results cover all the main questions that we could ask about the proposed algorithm (when it is advantageous to use geometry-aware optimization, what performance improvement it is supposed to bring, and what is the expected computational overhead). Likewise, the experimental section validates the theoretical results in both synthetic and real-world settings.

**Minor Observations**

- Table 1 and Figure 3 could benefit from a more detailed legend.
- Although it is not the main goal of the article, it seems interesting to mention the experiments performed on non-synthetic datasets in the main body of the paper.
- It is not clear to me what Figure 3 brings to the discussion of the empirical results.

---

> ### Author Rebuttal · Authors · 2026-03-30
>
> We sincerely thank you for the positive assessment and for recognizing the novelty of the theory and the completeness of the empirical study. We are especially encouraged that you found the paper clear, technically solid, and persuasive on the main questions the method addresses: when geometry-aware optimization helps, what gains it offers, and what computational overhead it incurs.
>
> For Key Question 1, we agree that the evaluation protocol behind Table 1 should be stated more explicitly. The intent of Table 1 is to compare optimization efficiency on a common objective, rather than compare methods using a separate generalization-based stopping rule. In our experiments, all methods optimize the same Sinkhorn divergence objective; the Sinkhorn subproblem is run for a fixed number of iterations to ensure consistent computational cost; methods are initialized from the same starting point within each replication; the same family of step-size schedules is used across methods, with tuning selected by validation; and results are averaged over 500 independent replications. In Table 1, $\theta^\star$ denotes the best solution attained across all methods within a given DGP/regime, and the reported quantities summarize progress relative to that shared reference. We will revise the caption and empirical section to define more clearly what "Iter", "Time", and "Gap" mean, and explain the protocol in a way that makes the comparison fair and transparent.
>
> For Key Question 2, we agree that it is helpful to clarify how the assumptions in the theory should be interpreted in practice. Most play an analytical role: they isolate the geometric mechanism and permit a clean separation between Euclidean and geometry-aware optimization, rather than serving as conditions that practitioners must verify before using the algorithm. For instance, Assumption 3.5 controls local variation of the intrinsic metric and prevents rapid growth of geometry-misalignment along geometry-aware trajectories, which is needed for Theorem 3.6 and the subsequent separation results. At the implementation level, however, GCO only requires operator access to $v \mapsto G(\theta)v$, a small number of power/Lanczos iterations to estimate $\hat{\kappa}_G(\theta)$, and approximate intrinsic solves via conjugate gradient. Thus, the practical method is less restrictive than the local theory used to analyze it. We agree, however, that the real-data scope should be more visible in the main paper. Appendix I already includes benchmark evaluations on Office-Home, VisDA-2017, and DomainNet, where all methods share the same backbone and differ only in the optimization rule. These experiments show the same qualitative pattern as the synthetic results; GCO matches or exceeds the accuracy of Euclidean and strong domain-adaptation baselines, while invoking geometry-aware updates on only a subset of iterations and keeping runtime far below always-on geometry-aware optimization. We will state this more explicitly in the main text and clarify that the assumptions justify the mechanism theoretically, not that practical usefulness depends on exact verification of the full set of local regularity conditions.
>
> For Key Question 3, we think this is an excellent and exciting direction. The paper suggests that optimization difficulty depends on how the representation map induces anisotropy in the pullback metric. This raises the possibility of designing architectures or parameterizations that are intrinsically more geometry-aligned, for example by controlling spectral anisotropy, feature sensitivity, or the conditioning of the induced metric $G(\theta)$. Such architecture design is beyond the scope of the present paper, whose focus is on diagnosis and adaptive optimization, but we agree that it is a natural next step and we will mention it explicitly in the discussion. This perspective suggests that optimization geometry may serve not only as an algorithmic tool, but also as a useful principle for model design and representation learning.
>
> We also appreciate the minor observations. We will improve the legends and captions for Table 1 and Figure 3, explain more directly what Figure 3 adds beyond the aggregated results in Table 1, and mention the non-synthetic experiments in the main text rather than only in Appendix I. In particular, Table 1 aggregates performance across DGPs and misalignment regimes, whereas Figure 3 visualizes the regime-dependent dynamics predicted by the theory: as misalignment increases, Euclidean methods deteriorate sharply, while GCO remains close to always-on geometry-aware optimization by activating intrinsic updates selectively. We will make this distinction clearer in the revision.
>
> We thank you again for the supportive evaluation. The points raised are very helpful, and we can address them by clarifying the empirical protocol, strengthening the discussion of the assumptions, and highlighting the non-synthetic results and implications of geometry-misalignment.

---

> > ### Author Rebuttal · Reviewer_9qGr · 2026-04-02
> >
> > We sincerely thank the authors for their clear and detailed answer to our concerns. We are convinced by their arguments, as far as our comments are concerned. For this reason we propose to keep the current evaluation of the paper.
> >
> > We wish however to point out that Reviewer 6R7c has raised important questions regarding the relation to related work and the validity of the proofs given the current assumptions (differentiability of the eigenvalues, trajectory staying within the neighborhood or $\theta^*$, dependency on the step size). They should be properly answered. However, we do not have the expertise to evaluate if they can be easily addressed in revision or require more fundamental changes in the paper. That being said, we trust the judgement of the authors to make this decision.

---

> > > ### Author Response · Authors · 2026-04-02
> > >
> > > Thank you very much for the thoughtful follow-up and for confirming that our rebuttal addressed your concerns. We are grateful for your positive assessment of the paper.
> > >
> > > We also appreciate your note regarding the related-work and theorem-scope questions raised by Reviewer 6R7c. We are addressing those points directly by clarifying the paper's positioning relative to Newton, quasi-Newton, and approximate-Newton methods, and by making the local scope, assumptions, and parameter dependencies of the theoretical statements more explicit. We believe these issues primarily call for clarification and sharper positioning, rather than changes to the paper's core contribution, theoretical separation, or empirical findings. The central contribution remains the structural diagnosis via geometry-misalignment, the sharp Euclidean-versus-geometry-aware separation, and the geometry-calibrated method. We will make these clarifications explicit in the revision. Thank you again for your careful reading and support.

---

### Official Review · Reviewer_QZAS · 2026-03-13

**Soundness:** 3
**Presentation:** 3
**Significance:** 3
**Originality:** 2
**Overall Recommendation:** 4
**Confidence:** 3

**Summary:**

This paper argues that typical optimization failures in distributional learning objectives often come from a structural mismatch between the geometry of the objective and the Euclidean geometry assumed by standard optimizers like SGD/Adam. To measure the distortion between Euclidean geometry and the intrinsic geometry, the authors proposed "geometry-misalignment". A main theoretical contribution is: establishing lower bounds showing that for a broad class of distributional learning problems, the convergence slow down induced by the geometry-misalignment is unavoidable. To solve this problem, the authors proposed a practical geometry-calibrated optimization approach. Empirical evidence that the proposed method helps most in regimes where misalignment is high.

**Compliance With Llm Reviewing Policy:**

Affirmed.

**Final Justification:**

I keep my score unchanged.

**Key Questions For Authors:**

1. Practical computation of $G(\theta)$ and solves: In the main experimental setup(s), what concrete procedure is used to obtain (i) $G(\theta)v$ products and (ii) approximate $G(\theta)^{-1} \nabla F(\theta)$? How do runtime and performance change as solver tolerance / iteration count varies?

2. How sensitive is GCO to misalignment estimation error in practice? Can you include ablations showing performance vs. noisy $\hat{\kappa}_G$ and vs. different gating thresholds $\kappa_0$.

3. The theory is local and uses intrinsic strong convexity/smoothness assumptions. Empirically, do you see that misalignment remains predictive of optimization difficulty in highly nonconvex deep models? Any negative cases where misalignment is high but Euclidean methods still work well?

**Limitations:**

The paper does discuss limitations in terms of access to the intrinsic metric and approximation requirements, but it would benefit from a more explicit limitations paragraph covering: 1. when $G(\theta)$ is hard to compute/ill-conditioned 2. overhead and memory constraints for large-scale deep nets.

**Strengths And Weaknesses:**

Strengths:

1. The paper makes a clear, technically grounded point: distributional discrepancies induce an intrinsic metric, and Euclidean optimization can be fundamentally mismatched to it.

2. The definition of misalignment as an eigenvalue ratio is straightforward and helps the reader build intuition.

3. Distributional objectives are widely used, and optimization instability is an important problem; giving a principled explanation and a condition number that predicts when SGD-like methods slow down is practically meaningful.

Weaknesses:


1. Related work positioning could be sharpened around geometry-aware optimization: the paper mentions natural gradient connections, but a clearer "what is new vs. existing geometry-aware optimization theory" summary would help.

2. The impact hinges on whether the proposed geometry-calibrated optimization can be deployed at scale with acceptable overhead and beyond numerical experiments (e.g., experiments on modern LLMs).

---

> ### Author Rebuttal · Authors · 2026-03-30
>
> We sincerely thank you for the careful reading and constructive feedback. We are encouraged that you found the paper technically interesting and recognized its central point: distributional objectives induce an intrinsic geometry, and optimization difficulty can arise from a mismatch between this geometry and the Euclidean geometry underlying standard first-order methods. We also appreciate the questions on novelty, scalability, robustness to estimation error in $\hat{\kappa}_G$, and the scope of the theoretical guarantees.
>
> Our contribution is not the general observation that geometry-aware optimization can help. Rather, the main novelty is a structural diagnosis for distributional learning through geometry-misalignment $\kappa_G$, together with a sharp separation between Euclidean and geometry-aware optimization. In particular, Theorem 4.2 shows that Euclidean first-order methods incur an unavoidable $\kappa_G$-dependent slowdown even when the objective is strongly convex and smooth. Theorems 4.3, 5.1, and 5.3 then show that geometry-aware and geometry-calibrated methods remove this dependence up to logarithmic factors. Theorem 5.5 further shows that this effect is not only optimization-theoretic but also statistical, since finite optimization budgets induce an excess-risk penalty that scales with misalignment. We agree that this distinction from prior natural-gradient, mirror-descent, and geometry-aware optimization work should be stated more explicitly, and in the revision we will sharpen the related-work discussion to emphasize that our main claim is a provable characterization of when Euclidean optimization is fundamentally inadequate for distributional objectives.
>
> Regarding practical computation, our implementation does not form $G(\theta)$. Instead, it relies on operator access $v \mapsto G(\theta)v$, estimates $\hat{\kappa}_G(\theta)$ from a few power/Lanczos iterations, and computes geometry-aware directions by approximately solving $G(\theta)v=\nabla F(\theta)$ with conjugate gradient. These solves use warm starts, a tolerance schedule proportional to the step size, and an iteration cap; if the tolerance is not met, the method falls back to a Euclidean step for stability. Thus, the extra cost is incurred only when misalignment is large enough to trigger geometry-aware updates. This selective activation is exactly why GCO achieves low overhead in well-aligned regimes while still obtaining the benefits of intrinsic optimization in highly anisotropic regimes. We agree that these implementation details should be stated more clearly in the main text, and we will revise the paper accordingly.
>
> On sensitivity to $\hat{\kappa}_G$ and the threshold $\kappa_0$, Appendix F already includes ablations varying both gating thresholds and solve tolerances. The results show a gradual trade-off rather than abrupt deterioration. Smaller $\kappa_0$ triggers geometry-aware updates more often and increases runtime, while larger $\kappa_0$ reduces overhead but gradually recovers Euclidean-like behavior in medium- and high-misalignment regimes. Likewise, tighter solve tolerances mainly affect runtime, with little effect on final optimization gaps. This matters because it shows that the practical value of GCO does not depend on an overly precise estimate of $\kappa_G$. We will make these robustness findings more explicit in the revision.
>
> We also agree that the main theory is local and does not claim a global theorem for arbitrary deep nonconvex models. This is intentional: the theory is designed to isolate the geometric mechanism from nonconvexity and stochastic noise, so that the effect of geometry-misalignment can be identified cleanly. To complement this theory, Appendix G includes deep distribution-matching experiments where $\hat{\kappa}_G$ is estimated using autodiff/Hessian-vector-product access. Our claim in that setting is empirical rather than a global theoretical guarantee: along realistic training trajectories, geometry-misalignment remains predictive of optimization difficulty, and GCO helps primarily when anisotropy is high. We do not claim that $\kappa_G$ is the sole determinant of optimization behavior in every nonconvex setting; rather, it is a predictive structural signal whose importance grows in anisotropic regimes. We agree that evaluation at modern-LLM scale is an important future direction, but it is beyond the scope of this submission.
>
> Finally, we agree that the limitations should be stated more explicitly. In the revision, we will add a clearer paragraph noting that the method is most attractive when matrix-vector access to $G(\theta)$ is reasonably available, and that memory and solver overhead may limit use in very large deep models.
>
> We hope these clarifications address your concerns and clarify the paper's contribution and scope. Our contribution is a structural theory, optimization guarantees, and a practical adaptive method whose gains concentrate in the regimes predicted by the theory.

---

> > ### Author Rebuttal · Reviewer_QZAS · 2026-04-01
> >
> > The rebuttal has addressed most of my concerns. In particular, it clarifies the intended novelty of the paper relative to prior geometry-aware optimization work, provides a concrete practical picture of how the method is implemented, and makes the scope of the theory in nonconvex settings clearer. While I still think the large-scale practical impact remains to be further validated, the rebuttal is overall satisfactory and does not change my original assessment. I therefore choose to keep my score unchanged.

---

> > > ### Author Response · Authors · 2026-04-01
> > >
> > > Thank you very much for the thoughtful follow-up. We appreciate that the rebuttal clarified the paper's novelty, implementation, and theoretical scope. We understand your remaining reservation about large-scale validation, and we agree this is an important next step. Our current claim is that the paper contributes a structural theory, a sharp Euclidean-vs-geometry-aware separation, and a practical adaptive method whose gains appear in the high-misalignment regimes predicted by the theory. We will further clarify the practical scope of the method and the associated scalability considerations in the revision. Thank you again for your careful assessment.

---

### Decision · Program_Chairs · 2026-04-30

**Decision:**

Accept (regular)

**Comment:**

There was some disagreement among reviewers as to the novelty of the core contribution of this work, its clarity, and how it is situated within the context of existing work.

During rebuttal, the authors emphasized that their proposed contribution is not the use of gradient preconditioning, but rather the observation that non-geometry-aware optimization methods can be seriously mismatched for distributional learning. The authors also acknowledged that the setting in which the theory holds could be made more precise, and that the relationship to existing work on second-order optimization methods should be fleshed out. In my opinion these are valid criticisms, but they can be addressed with relatively minor revisions.

We encourage the authors to incorporate all reviewers’ feedback for the final version to ensure it is clear what are the limits of the theory and how this work contributes on top of the extensive existing literature on optimization.